# `TokSuite`: Measuring the Impact of Tokenizer Choice on Language Model Behavior

Gül Sena Altıntaş [* 1 2]  Malikeh Ehghaghi [* 1 2]  Brian Lester [1 2 3]  Fengyuan Liu [4 5]  Wanru Zhao [6]
Marco Ciccone [2]  Colin Raffel [1 2 7]

## Abstract

Tokenizers provide the fundamental basis through which text is represented and processed by language models (LMs). Despite the importance of tokenization, its role in LM performance and behavior is poorly understood due to the challenge of measuring the impact of tokenization in isolation. To address this need, we present `TokSuite`, a collection of models and a benchmark that supports research into tokenization's influence on LMs. Specifically, we release fourteen pre-trained models that use different off-the-shelf tokenizers but are otherwise identical, using the same architecture, dataset, training budget, and initialization. We also release a multilingual robustness benchmark that measures model performance under real-world perturbations in English, Chinese, Farsi, Italian, and Turkish, curated by native annotators. Together, `TokSuite` allows robust decoupling of the influence of a model's tokenizer, supporting a series of novel findings that elucidate the respective benefits and shortcomings of a wide range of popular tokenizers.

🙂 **Models & Benchmark:** toksuite
🐙 **Code:** r-three/TokSuite  r-three/lingua.

## 1. Introduction

Language models (LMs) generally do not process "raw text" directly; instead, they operate on a sequence of "tokens" that represent words, sub-words, or characters. As a result, tokenization fundamentally influences the representation learned by LMs and, consequently, affects the downstream model capabilities (Mielke et al., 2021). For example, the

*Equal contribution  [1]University of Toronto [2]Vector Institute [3]Google DeepMind [4]McGill University [5]Mila - Quebec AI Institute [6]University of Cambridge [7]Hugging Face. Correspondence to: Gül Sena Altıntaş <gsaltintas@cs.toronto.edu>.

*Proceedings of the 43rd International Conference on Machine Learning*, Seoul, South Korea. PMLR 306, 2026. Copyright 2026 by the author(s).

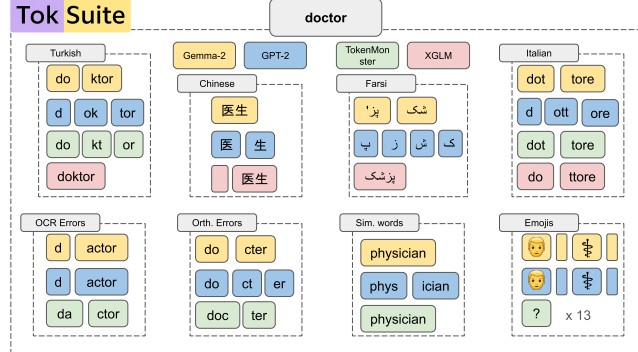

*Figure 1.* `TokSuite` is a comprehensive benchmark covering real-world perturbations that change tokenization, and 14 models that share the same initialization, architecture, and data but differ only in their tokenizers. The figure illustrates how different tokenizers fragment the concept "doctor" when subjected to OCR errors, orthographic mistakes, semantic equivalents, emoji substitution, and multilingual translations. Each colored box represents one token across Gemma-2 (yellow), GPT-2 (blue), TokenMonster (green), and XGLM (red) tokenizers.

tokenizer used in T5 (Raffel et al., 2020) cannot represent curly brace tokens, making the T5 models poorly suited to processing many coding languages (Wang et al., 2021c). The importance of tokenization naturally motivates not only understanding the impact of different tokenization strategies but also the design of better tokenizers. However, tokenization is a relatively understudied aspect of language model development compared to model architectures, training recipes, and dataset curation. In fact, the design of the tokenizer is often treated as an afterthought, with many open models simply using a preexisting tokenizer off the shelf. For instance, the GPT-2 tokenizer was directly reused for Meta's Open Pretrained Transformers (OPT) (Zhang et al., 2022), and EleutherAI's GPT-NeoX-20B tokenizer was directly used for the MPT-7B-8k model (Team, 2023) and Pythia models (Biderman et al., 2023).

One reason research on the impact of tokenization remains limited is that, with existing artifacts, it is difficult to disentangle the tokenizer's effects from confounding factors such as model architecture and training data. For example, it would be fraught to compare the Qwen 3 (Yang et al., 2025) and Llama 3 (Dubey et al., 2024) tokenizers by studying the

respective models because differences in training data, training duration, and architecture make it difficult to attribute performance specifically to tokenization. Understanding the downstream effects of tokenizer design choices is further complicated by the multifaceted nature of tokenization itself, involving various interrelated factors including the underlying segmentation algorithm (e.g., BPE (Gage, 1994; Sennrich et al., 2016), Unigram (Kudo, 2018), WordPiece (Wu et al., 2016)), granularity level (e.g., byte-level (Xue et al., 2022), character-level, word-level), vocabulary size constraints, and the composition of training data used to learn the vocabulary.

*How can we reliably measure the impact of tokenization on model behavior?* Isolating the effect of tokenization necessitates the use of models that are completely identical apart from the tokenizer used. To the best of our knowledge, there is no open collection of models that are identical in architecture and training setup and differ only in their choice of off-the-shelf, industry-standard tokenizers.

Our first contribution is therefore to train and release 14 LMs with identical initialization, architecture, and training data composition, varying only in the tokenizer used. Our suite of models covers a wide range of tokenizer types, selected among popular *pretrained* tokenizers as representatives of their main distinctive features, from byte-level tokenization to subword-based approaches including BPE, SentencePiece (Kudo & Richardson, 2018), and WordPiece variants. This collection encompasses both English-only tokenizers trained on monolingual corpora and multilingual tokenizers designed to handle diverse language families and scripts. The tokenizers additionally exhibit varying approaches to out-of-vocabulary (OOV) handling, Unicode normalization, whitespace treatment protocols, continuation token markers for subword boundaries, and pretokenization splitting rules. Our chosen tokenizers also have diverse vocabulary sizes ranging from compact, efficient lexicons to comprehensive multilingual vocabularies, each with distinct trade-offs between compression efficiency and linguistic coverage. Noting that different vocabularies might share tokens, we develop a novel vocabulary unification framework that creates bijective mappings between tokenizer-specific and unified token spaces. This allows us to use a unified parameter initialization where embeddings for shared tokens are initialized to the same value across models.

*To what extent does linguistic variety and the resulting changes in tokenization affect the consistency of model performance?* Our second contribution is TokSuite benchmark, a novel benchmark of approximately 5,000 samples specifically designed to target use cases where real-world linguistic variations alter tokenization despite identical semantic meaning. Since the effect of different tokenizers can vary across languages (Ali et al., 2024; Dang et al., 2025; Seo et al., 2025), our benchmark includes five orthographically and morphologically diverse languages: English (EN), Turkish (TR), Italian (IT), Farsi (FA), and Mandarin Chinese (ZH). Specifically, Farsi uses Arabic script and presents unique challenges in which the same text can be represented by optional diacritics. Mandarin Chinese is a logographic and isolating language. TokSuite also covers its romanization through Pinyin, the Chinese Phonetic Alphabet, and errors relating to it, which is rarely found in the training data but is an essential part of daily communication. Turkish is an agglutinative language with six additional letters in its alphabet and rich in grammar that severely impacts word form and tokenization. Italian is representative of fusional Latin languages with complex inflectional patterns and accents.

Our benchmark includes 40 "canonical" multiple-choice text completion questions translated into all five languages. Each question has different perturbed versions manually curated by native speakers that reflect real-world changes users might make. For example, we test what happens when visually identical characters have different Unicode values (e.g., replacing Latin "a" with Cyrillic "a"), when users type Turkish text with English keyboards (causing "ş" to become "s"), when Farsi text includes or omits optional accent marks, and when regular text uses special Unicode formatting such as enclosed characters. We also add two specialized benchmarks: an elementary school math dataset and a science, technology, engineering, and mathematics (STEM) dataset, respectively, with 20 and 44 "canonical" technical questions alongside targeted perturbations. This multi-domain approach allows us to assess tokenizer performance across general, mathematical, and scientific content.

Our evaluations confirm prior intuitions and uncover new insights into how tokenizer design shapes model behavior. Unconventional tokenizers (e.g., TokenMonster (Forsythe, 2025) and ByT5 (Xue et al., 2022)) are substantially more robust than standard subword methods despite smaller vocabularies. Noise-based perturbations disproportionately affect non-English languages, and all tokenizers struggle with Unicode formatting and technical text such as LaTeX and STEM notation. These robustness trends persist across model scales, indicating that tokenizer design rather than parameter count or training duration dominates robustness. Together, our models, dataset, and findings provide a foundation for future tokenization research.

## 2. Background

Before focusing on how tokenization can affect downstream LM performance, we first explain how tokenizers are created and how design decisions can affect the final tokenizer.

**Tokenizers** Tokenization is the process of converting a sequence of input symbols into meaningful lexical tokens

from some vocabulary $\mathcal{V}$. Each entry in the vocabulary corresponds to a particular string, and tokenizing an input string can be seen as segmenting it into strings from the vocabulary. The vocabulary defines a mapping for each token to an integer ID, $V : S \mapsto \{0, 1, \ldots, |V| - 1\}$. These IDs are then used to look up a vector representation of the token in an LM's embedding table, thus creating a real-valued vector input for each token in an input sequence. While $\mathcal{V}$ can be manually enumerated for languages with restrictive grammars (e.g. programming languages), the ambiguity and open-endedness of natural language necessitate estimating an optimal set of tokens from data.

Consequently, differences in tokenizers can result in different token sequences for the same string. These differences can affect both learnability and how information is processed in downstream models. For example, a tokenizer that maps the string "dogs" to two tokens "dog" and "s" allows the model to "reuse" its understanding of the token for "dog", but requires composing with the meaning of the "s" token as pluralization. In contrast, a tokenizer that includes "dogs" as its own token packs both dog and its pluralization into a single token. These differences generally arise in the three main components involved in tokenizer training: data, learning algorithm, and preprocessing decisions.

**Training Data**    In order to determine the collection of substrings in the vocabulary, tokenizers are generally trained on a text dataset. While the training process for different approaches to tokenization can vary (see the following subsection), one straightforward effect of the training data is that if the training dataset does not include a given word or symbol, it will not be in the vocabulary. Similarly, differences in tokenizer training datasets can result in different choices for tokens included in $\mathcal{V}$ by different tokenizer learning algorithms. For example, if one tokenizer is trained on web data that includes many examples of the typo "teh", it is more likely to represent it as a single token in its vocabulary compared to a tokenizer that is only trained on highly edited text where this typo is rare.

The inclusion of multilingual data in the tokenizer training data can also greatly impact the final vocabulary, especially when scripts that do not share an alphabet are included. Generally a much larger vocabulary is required, for example, the increase from 32,000 to 256,000 when moving from T5 (Raffel et al., 2020) to mT5 (Xue et al., 2021).

**Learning Algorithm**    When training a tokenizer, a learning algorithm produces a vocabulary $\mathcal{V}$ that "fits" the training data, with inclusion primarily determined by frequency. Most tokenizers function as compressors (Lester et al., 2024), assigning common words to single tokens while splitting rarer ones. Common algorithms include Byte-Pair Encoding (BPE) (Gage, 1994), which iteratively merges

the most frequent symbol bigrams until reaching vocabulary size $|\mathcal{V}|$; WordPiece (Wu et al., 2016), which merges symbols by maximizing training data likelihood; and Unigram (Kudo, 2018), which starts with all possible segmentations and removes symbols causing minimal unigram loss increase. TokenMonster (Forsythe, 2025) uses an unusual approach, building a global vocabulary from all possible tokens and employing an "ungreedy" algorithm that revises tokenization by lookahead. Byte-level models like ByT5 (Xue et al., 2022) use predefined Unicode vocabularies of bytes rather than learned ones (Mielke et al., 2021).

Vocabulary size $|\mathcal{V}|$ significantly affects composition, as larger vocabularies include more rare words as individual tokens. While most tokenizer training algorithms ensure that every string in the training set can be tokenized, "byte-fallback" forces $\mathcal{V}$ to include the 256 bytes needed to represent any character in Unicode. This allows tokenization of symbols that do not appear in the training dataset.

For a more in-depth discussion of various tokenization approaches, refer to Mielke et al. (2021).

**Preprocessing**    Tokenization pipelines often use some form of pre-tokenization, which segments the input text into "intuitive" pre-tokens, such as whitespace-separated words, before the learning algorithm is applied. This segmentation can limit which strings can be added to $\mathcal{V}$ as the learning algorithms do not consider bigrams that cross pre-tokenization boundaries. This means that very common bigrams such as "New York" *cannot* be represented as a single token. While some work (Schmidt et al., 2025; Liu et al., 2025, *et alia*) explores methods that allow cross-boundary merges, most commonly used tokenizers do not.

As another example of pre-tokenization, the GPT-2 tokenizer (Radford et al., 2019) splits contractions—e.g., "we'll" → "we", "'ll"—meaning that "we'll" cannot be a token in $\mathcal{V}$. In contrast, BLOOM's (Workshop et al., 2022) pre-tokenization process does not force contractions to a new token, thus allowing for "we'll" $\in \mathcal{V}$.

Similar differences exist in the handling of numbers. The pre-tokenization used in some models, like GPT-4 (Achiam et al., 2023), breaks contiguous digits into groups of three ("1337" → "133", "7") while other models split numbers into their digits. There are also models that rely exclusively on the learning algorithm to decide how to segment numbers into digits. Each approach has trade-offs; for example, splitting numbers into thousands might be natural for math but is less natural for dates. Similar considerations exist for how repeated whitespace is handled, especially in domains like code where whitespace can be especially meaningful.

## 3. The `TokSuite` Models

### 3.1. Tokenizer Selection and Characteristics

To systematically investigate how different tokenization design choices affect model performance and robustness, we began by selecting a diverse set of 14 preexisting tokenizers, specifically ByT5 (Xue et al., 2022), TokenMonster (Forsythe, 2025), Phi-3 (Abdin et al., 2024), GPT-2 (Radford et al., 2019), Comma (Kandpal et al., 2026), mBERT (Devlin et al., 2019), Llama-3.2 (Dubey et al., 2024), Tekken (AI, 2024), Qwen-3 (Yang et al., 2025), GPT-4o (Hurst et al., 2024), BLOOM (Workshop et al., 2022), Aya (Dang et al., 2024), Gemma-2 (Team et al., 2024), and XGLM (Lin et al., 2022).

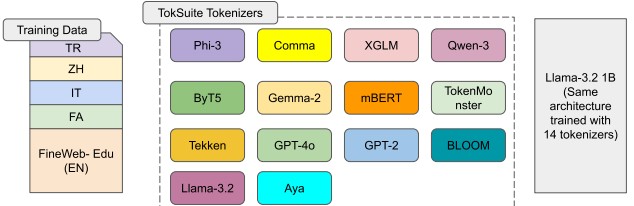

*Figure 2.* `TokSuite` consists of 14 models that share the same initialization, architecture, and data but differ only in their tokenizers.

Our selection provides comprehensive coverage across vocabulary sizes (ranging from 259 tokens in byte-level tokenizers like ByT5 to over 256,000 tokens in models such as Aya or XGLM), tokenization algorithms (BPE, Word-Piece, Unigram, TokenMonster, and byte-level approaches). This diversity enables systematic analysis of how different tokenizers handle out-of-vocabulary words, morphological variations, and adversarial inputs. The selected tokenizers also encompass notable variation in preprocessing strategies that affect robustness, including different approaches to numerical content handling (digit splitting vs. grouping), contraction processing (rule-based vs. learned), Unicode normalization schemes, and multilingual support ranging from monolingual to 100+ languages. Additionally, the tokenizers vary in their out-of-vocabulary handling mechanisms, with some incorporating byte-fallback and others relying on unknown tokens, providing insight into how these design choices propagate to model robustness under various challenges. Detailed technical specifications for each tokenizer are provided in Table 2 and Table 3 in the Appendix.

### 3.2. Cross-Tokenizer Vocabulary Alignment

To align vocabularies across tokenizers, we first create a unified "*super vocabulary*". For each tokenizer $i \in T$, where $T$ is the set of all tokenizers, we extract its vocabulary $\mathcal{V}_i$, accounting for tokenizer-specific quirks (like WordPiece's "##" prefixes or Unigram's "_" whitespace markers). We also unify the strings that denote the beginning of a sequence—

``, `<|beginoftext|>`, etc. Then, we create a super vocabulary, $\mathcal{SV}$, by taking the union of all vocabularies $\mathcal{SV} = \bigcup_i \mathcal{V}_i$. Note that this unification is based on the UTF-8 byte representation of each element in the vocabularies.

Finally, for each tokenizer, we create a mapping, $SV : V(X) \mapsto SV(X)$ that translates a tokenizer's original token IDs to the corresponding positions in the unified super vocabulary. This ensures a token string always maps to the same index, regardless of the tokenizer used, that is, $\forall i, j \in T, \ SV(V_i(S)) = SV(V_j(S))$, if $S \in \mathcal{V}_i \cap \mathcal{V}_j$. The use of the super vocabulary allows us to use the same initialization for the embeddings for shared tokens across models. This shared starting point alleviates the variation in initialization across models, allowing more rigorous attribution of downstream performance to tokenizer characteristics.

### 3.3. Model Architecture and Training Configuration

We trained fourteen LMs (one for each tokenizer in Figure 2) using Meta's Lingua framework (Videau et al., 2024). Our model architecture and training hyperparameters follow Lingua's Llama-3.2-1B with untied embeddings configuration with approximately one billion non-embedding parameters, following the Llama model family (Dubey et al., 2024). All models use a shared initialization based on the super vocabulary. See Appendix C.1 for more information. All models were trained for 100,000 steps with batches of 256 length-4096 sequences. We use the AdamW (Loshchilov & Hutter, 2019) with a weight decay of 0.1 and a peak learning rate of 0.001 with cosine annealing and 2000 warm-up steps.

We train all models on a multilingual corpus totaling approximately 100 billion tokens. For English content, we use FineWeb-Edu (Penedo et al., 2024b; Lozhkov et al., 2024), which provides high-quality content filtered from Common Crawl data. For the multilingual components, we use the Chinese, Turkish, Italian, and Farsi subsets of the FineWeb2-HQ Dataset (Messmer et al., 2025), which is a pre-training dataset derived from FineWeb2 (Penedo et al., 2025) by selecting the top-quality documents across languages. The final corpus composition consists of 40B English tokens and 60B multilingual tokens equally distributed across the four target languages (15B each).

For training, we use a fixed token budget following current practice in LLM training and reporting. This means that each model sees different amounts of raw information (in bytes/documents), see Appendix C.3. For example, 100B tokens correspond to approximately 100GB (ByT5), 278GB (Comma), and 471GB (Gemma-2) of UTF-8 bytes. However, we consider the alternative—training each model on the same text, but for a different number of training steps—to be more problematic, because training duration heavily influences performance and some models would be relatively under- or over-trained. We validate this choice by training

four models with the same text-budget in Appendix G.2. Additionally, a tokenizer's efficiency in compressing the training data is a relevant factor in tokenizer selection.

As a sanity check to ensure that our trained models behave as expected, we evaluated them on standard benchmarks commonly used to assess the base LMs: HellaSwag (Zellers et al., 2019), ARC (Clark et al., 2018), PIQA (Bisk et al., 2020), and XNLI (Conneau et al., 2018). Results are shown in Figure 7. In general, we find that our models perform reasonably well given their parameters and training budget. However, we do find notable differences in performance across different models. Since our models are otherwise equivalent, this performance difference can be attributed directly to tokenization, which we discuss further in Section 5.

## 4. The TokSuite Benchmarks

To systematically study the impact of tokenizers on model performance, we develop a new benchmark that captures various input variations models may encounter in real-world deployment. Unlike existing evaluations focusing on clean, canonical text, our benchmark specifically targets naturally occurring perturbations that expose tokenization-dependent issues across our target languages (Chinese (ZH), English (EN), Farsi (FA), Italian (IT), and Turkish (TR)) and domains including general knowledge, basic arithmetic, and STEM. Since the benchmark aims to assess robustness to variations in tokenization, we deliberately select simple, canonical questions designed to attain a strong baseline performance across all models. The selection of canonical questions follows a model-in-the-loop process in which we iteratively test question candidates across our model suite to ensure high baseline accuracy, allowing us to cleanly measure performance degradation under perturbations. For each question in the canonical benchmarks, over 70% of the models responded correctly. As shown in Figure 8, model performance consistently exceeds 70–75% accuracy on canonical tasks, both in English and non-English settings.

The TokSuite benchmark is integrated into lm-eval-harness (Gao et al., 2024) where any model-tokenizer pair can be evaluated on TokSuite tasks by running the following command:

```
$ lm-eval --model=hf \
--model_args pretrained=toksuite/..,
tokenizer=..   --tasks=toksuite
```

### 4.1. Multi-lingual Parallel Dataset

We select 40 *canonical* questions in multiple-choice text completion format in English that almost all of the fourteen models answer correctly, such as "The capital of France is," "The chemical formula for water is," and "The number of

continents on Earth is". We aim for canonical questions that our base models get correct so that we can study cases where perturbations flip the answer to incorrect. Native speakers then translate each canonical question into FA, IT, TR, and ZH. Subsequently, each example undergoes targeted *perturbations* designed to reflect the morphological and orthographic characteristics of each language. Canonical questions in English are provided in Appendix D.1, and further examples of each category with detailed case studies on tokenization differences are presented in Appendix E.

**Orthographic Perturbations** include input medium challenges, diacritics perturbations, orthographic errors, and variations in writing systems, linguistic register, and stylistic conventions. *Writing System Variations* include script variations such as traditional vs. simplified Chinese characters, and romanization, i.e. writing text in Latin script like Pinyin for Chinese or Finglish for Farsi. *Input medium challenges* capture typing scenarios where users employ non-native keyboards, leading to systematic character substitutions. This category also includes spacing irregularities with zero-width characters, and homoglyphs (visually similar characters with different Unicode values). *Diacritics* perturbations include presence of optional diacritics, where text remains valid with or without marks, fatha for /a/, kasra for /e/ in FA, and common accent errors (è → é) in IT.

*Orthographic errors* represent spelling mistakes and character-level variations encountered in real-world text like vowel substitutions, consonant errors, phonetic spelling variants, common misspellings, and punctuation errors.

*Register & Style* captures variations in linguistic register and stylistic conventions across different contexts. This includes web search query formatting with shortened keyword expressions, standard and domain-specific abbreviations, and word reordering that reflect old orthographic conventions. This category encompasses informal digital communication patterns such as colloquial language, emoji or character substitution, and letter repetition for emphasis.

**Morphological Challenges** cover contractions, compound words, inflectional variations in IT, case marking, and derivations that may fragment or alter token boundaries. These challenges are particularly pronounced in agglutinative languages such as Turkish.

**Noise** perturbations introduce realistic types of textual noise encountered in practice, including typos, character or space deletion, character permutation, and formatting inconsistencies arising from sources such as OCR or other data processing pipelines. These variations test the robustness of the tokenizer under imperfect input conditions.

**Grammatical Errors** cover typical mistakes made by non-expert speakers like subject-verb agreement, article omission or misuse, wrong preposition, incorrect verb tenses,

and structural errors.

**Linguistic Variety** covers variations in expressing the same semantic content across different linguistic contexts. It includes equivalent expressions with different syntactic structures, code-switching, similar words, historical spelling variations, and dialects representing regional language varieties with different vocabulary and spelling conventions.

**Structural Text Elements** includes Unicode-based formatting (see Figure 9) and stylistic variations that preserve semantic content while altering visual presentation.

### 4.2. Math & STEM Datasets

Beyond testing simple world knowledge, a subset of our benchmark tests basic arithmetic and STEM, allowing `TokSuite` to include domain-specific perturbations.

**LaTeX and Formatting** variations include straightforward examples such as `$6$` and `$N_2$`, as well as more complex formatted expressions like `$\frac{\text{kg} \cdot \text{m}^2}{\text{s}^2}$`. We also include ASCII-based structural representations such as molecular diagrams, tree structures, and flowcharts.

**Multilingual Basic Arithmetic** is tested by translating canonical math questions to ZH, FA, TR, and IT.

### 4.3. The `TokSuite` Evaluation Framework

**Robustness** We evaluate models with `lm-eval`'s (Gao et al., 2024) byte-length normalized log-likelihood. For fair comparison among models with different baseline capabilities, we report relative accuracy drop for each model against its canonical performance within each category, computed as $\frac{\text{Acc}_{\text{can}} - \text{Acc}_{\text{pert}}}{\text{Acc}_{\text{can}}}$, where $\text{Acc}_{\text{can}}$ is the canonical accuracy and lower values indicate greater robustness.

**Intrinsic Tokenization Efficiency** We evaluate tokenizers' efficiency in compressing text from the five target languages using 10,000 parallel Flores200 (Team et al., 2022) samples with three metrics: (1) *Subword fertility (SF)*: mean number of tokens per word, where lower values indicate less segmentation; (2) *Parity*: cross-lingual fairness measured as the ratio of tokenized lengths $\frac{|T(s_A)|}{|T(s_B)|}$ for parallel sentences (Ali et al., 2024); (3) *Proportion of continued words (PCW)*: fraction of words requiring multiple tokens (Rust et al., 2021). See Appendix B for detailed results.

**Statistical Validation:** We estimate the variability of the robustness metrics through a 10,000-trial bootstrap resampling, computing 95% confidence intervals for all reported metrics. We also evaluate the statistical significance of pairwise performance differences via Wilcoxon signed-rank tests (Wilcoxon, 1945) with $\alpha = 0.05$ in Appendix H.

## 5. Findings

We present our core robustness results in Table 1, showing relative performance drops across `TokSuite` perturbation categories and models. Values are averaged across 10,000 bootstrap samples. All performance differences discussed exceed one standard deviation, confirming these observations are unlikely due to random variation. Additionally, most of the differences reported are statistically significant per paired Wilcoxon signed-rank tests (statistical analysis in Appendix H). The following findings validate our benchmark's effectiveness and demonstrate our model collection's utility for tokenizer research, confirming prior observations under controlled conditions while revealing previously unmeasured phenomena. Our key findings are as follows.

**Impact of Tokenization Algorithm Design on Multilingual Robustness** While orthographic and morphological diversities present universal difficulties across tokenizers, TokenMonster's performance is particularly striking given its architectural constraints. Despite having a 32,000-token vocabulary trained exclusively on English text, which is roughly one-eight the size of multilingual competitors like Aya or XGLM, it achieves the best average robustness score across all multilingual perturbations, with the lowest average relative performance drop of 17% (see Table 1). This effectiveness stems not from its vocabulary size, but from its unique "ungreedy" tokenization algorithm and consistency constraints, which limit the number of distinct token sequences representing the same word, presenting the model with more consistent inputs regardless of surface-level variation such as capitalization or spacing.

ByT5 also demonstrates strong multilingual robustness, outperforming 9 models (see Table 1) despite using only a 259-token vocabulary. Its byte-level design achieves minimal performance degradation across diverse perturbations: 0.04/0.06 drops for English/non-English orthographic errors (see Table 1), 0.00 drop for English grammatical errors (see Table 11), and top average 0.18 drop for multilingual noise (e.g., typos, OCR errors, etc.) (see Table 16). The model shows particular strength in Turkish and Chinese scenarios, including romanized Pinyin handling and even performance improvements (-0.11) with zero-width characters (see Table 9). However, this robustness comes at an efficiency cost, with the highest subword fertility and PCW scores across all languages (see Appendix B), reflecting a robustness-efficiency trade-off for the byte-level model. These findings suggest that hybrid tokenization strategies that combine subword efficiency with byte-level robustness, such as BPE dropout (Provilkov et al., 2020) or subword regularization (Kudo, 2018), warrant further investigation.

*Table 1.* Tokenization robustness under multilingual text perturbations. Values represent relative performance drop ($\frac{\text{Acc}_{\text{can}} - \text{Acc}_{\text{pert}}}{\text{Acc}_{\text{can}}}$); lower values indicate greater robustness. Perturbation types: Input: non-native keyboard/romanization; Diacr.: optional diacritics; Orth. & Gram.: orthographic and grammatical errors; Morph.: derivations/inflections/contractions; Noise: homoglyphs/OCR/typos/spacing; LaTeX: LaTeX-style math formatting; STEM: scientific diagrams and notations; Unic.: Unicode styling characters. NEN: non-English. Breakdown of each category and detailed case studies are presented in Appendix E. **Green** and red entries highlight notable robustness and fragility, respectively.

| Model | Input NEN | Diacr. NEN | Orth. & Gram. EN | Orth. & Gram. NEN | Morph EN | Morph NEN | Noise EN | Noise NEN | LaTeX EN | STEM EN | Unic EN | Avg |
|---|---|---|---|---|---|---|---|---|---|---|---|---|
| TokenMonster | **0.23** | **0.33** | 0.08 | **0.01** | 0.23 | **-0.07** | **0.10** | **0.18** | 0.21 | **0.10** | 0.51 | **0.17** |
| XGLM | 0.34 | 0.49 | 0.10 | 0.11 | 0.25 | 0.07 | 0.12 | 0.22 | 0.29 | 0.29 | **0.11** | 0.22 |
| BLOOM | 0.30 | 0.34 | 0.13 | 0.07 | **0.18** | 0.11 | 0.18 | **0.18** | 0.24 | 0.11 | 0.57 | 0.22 |
| ByT5 | 0.30 | 0.44 | **0.04** | 0.06 | 0.27 | 0.04 | 0.14 | **0.18** | 0.17 | 0.29 | 0.53 | 0.22 |
| Comma | 0.28 | 0.43 | 0.05 | 0.07 | **0.18** | 0.00 | 0.11 | 0.20 | 0.23 | 0.29 | 0.61 | 0.22 |
| mBERT | 0.33 | 0.44 | 0.11 | 0.11 | 0.23 | 0.06 | 0.18 | 0.22 | **0.14** | 0.22 | 0.61 | 0.24 |
| GPT-4o | 0.30 | 0.51 | 0.08 | 0.05 | 0.21 | 0.05 | 0.16 | 0.19 | 0.24 | 0.33 | 0.55 | 0.24 |
| GPT-2 | 0.34 | 0.46 | 0.07 | 0.10 | 0.25 | 0.06 | 0.14 | 0.21 | 0.24 | 0.35 | 0.53 | 0.25 |
| Phi-3 | 0.33 | 0.46 | 0.16 | 0.09 | 0.27 | 0.08 | 0.17 | 0.21 | 0.24 | 0.22 | 0.55 | 0.25 |
| Gemma-2 | 0.32 | 0.42 | 0.14 | 0.15 | 0.24 | 0.03 | 0.16 | 0.25 | 0.22 | 0.36 | 0.57 | 0.26 |
| Qwen-3 | 0.36 | 0.42 | 0.14 | 0.11 | 0.25 | 0.06 | 0.16 | 0.23 | 0.26 | 0.29 | 0.57 | 0.26 |
| Llama-3.2 | 0.33 | 0.55 | 0.11 | 0.10 | 0.25 | 0.08 | 0.15 | 0.24 | 0.17 | 0.30 | 0.59 | 0.26 |
| Aya | 0.31 | 0.46 | 0.14 | 0.10 | 0.22 | 0.03 | 0.19 | 0.25 | 0.21 | 0.38 | 0.58 | 0.26 |
| Tekken | 0.33 | 0.47 | 0.18 | 0.03 | 0.31 | 0.10 | 0.14 | 0.21 | 0.27 | 0.43 | 0.54 | 0.27 |
| Avg | 0.31 | 0.44 | 0.11 | 0.08 | 0.24 | **0.04** | 0.15 | 0.21 | 0.22 | 0.28 | 0.53 | 0.24 |

**Amplification of Tokenization Vulnerabilities under Multilingual Noise** Noise-based perturbations create systematic degradation across all tokenizers, but the average performance drop due to noise is markedly more severe for non-English languages (0.21) compared to English (0.15) (see Table 1). This degradation can stem from the core mechanics of subword tokenization: when noise corrupts a familiar word, the tokenizer fragments it into unfamiliar or non-sensical subword units. This effect is particularly damaging in morphologically complex languages. For instance, a simple spacing error in the Turkish phrase "gün sayısı" (day count) causes it to be re-tokenized into chaotic and less meaningful sequences like gün, ##s, ay, ##ısı by mBERT or gü, ns, ay, ısı by Llama-3.2. In contrast, the byte-level tokenizer ByT5 proves more resilient, as character-level errors result in a predictably altered sequence of known bytes rather than catastrophic fragmentation. This suggests that the reliance on a fixed vocabulary in subword models creates an inherent brittleness that is significantly exacerbated by noise in multilingual contexts. See Section E.3 for a detailed case study of this fragmentation phenomenon.

**Structural Limitations in Mathematical and STEM Content** Technical content presents unique tokenization challenges extending beyond vocabulary coverage. Analysis of mathematical and STEM content reveals critical tokenizer dependencies, with models showing significant performance degradation (average drops of 0.22 for LaTeX and 0.28 for STEM content (see Table 1)). Even in simplified text completion format with mild technical notation, models exhibit vulnerability to descriptive STEM content. These domains rely heavily on precise whitespace treatment, symbol placement, and structural conventions parallel to challenges in coding tasks where spacing and formatting carry semantic meaning. See Appendix E.4 for a detailed case study.

**Universal Challenges Across Tokenizers** Formatting presents a universal challenge. Unicode styling and character transformations degrade performance consistently across nearly all models, with an average drop of 0.53, the highest drop observed (see Tables 1, 18, 19). XGLM shows strong robustness to these perturbations thanks to its NFKC normalization during preprocessing. While this mitigates performance degradation from styled characters, it also means that the tokenizer cannot faithfully represent or generate the diverse Unicode formatting present in real-world text.

**Scaling Effects on Tokenization Robustness** TokSuite remains a challenging benchmark across different model capacities. In a controlled experiment with identically trained Llama-3.2 models of varying sizes (300M, 1B, and 7B), we found minimal variation in robustness across model scales, with average performance drops of 0.26, 0.26, and 0.22, respectively (Table 22). While canonical performance (Table 23) notably improves, robustness remains mostly similar across all perturbations except those related to noise.

Evaluation of larger, industry-scale models (Table 28), trained for orders of magnitude longer than the models in TokSuite, shows only modest improvements in robustness (details in Appendix G.4). These findings demonstrate that tokenization design is the dominant factor influencing these robustness characteristics, more so than simply increasing

parameter size, training duration, and data.

# 6. Related Work

While tokenization is relatively understudied compared to other aspects of LM development, prior work has studied tokenization's impact on model performance and cost.

**Tokenization Design Factors:** Ali et al. (2024) and Schmidt et al. (2024) both trained models with different tokenization settings, finding that intrinsic metrics like compression efficiency do not reliably predict downstream performance. In TokSuite we evaluate pretrained tokenizers with finergrained control (including controlled initialization) across five typologically diverse languages.

*Multilingual Coverage:* Ali et al. (2024) demonstrated that using English-centric tokenizers in a multilingual setting leads to severe downstream degradation across five European languages. Rust et al. (2021) found that monolingual tokenizers play an equally important role as pretraining data size in downstream performance. Islam et al. (2022) showed vocabulary-free neural tokenizers yielded substantial improvements for low-resource languages in multilingual natural language inference.

*Algorithm Choice:* Hou et al. (2023) showed that morphological segmentation outperformed BPE in morphologically rich languages, while Richburg et al. (2020) found Unigram models exhibit superior recall for rare words compared to BPE in languages like Swahili and Turkish in machine translation. Kudo & Richardson (2018) demonstrated 380x faster processing speeds with comparable performance.

*Vocabulary Size:* Huang et al. (2025) showed log-linear benefits from scaling input vocabulary size, while Tao et al. (2024) argued that most LLMs use insufficient vocabulary sizes suggesting Llama2-70B's optimal size should be 216K rather than 32K tokens.

*Byte-level Models:* ByT5 (Xue et al., 2022) showed that byte-level tokenizers can match or outperform subword tokenizers on both discriminative and generative tasks. Dang et al. (2025) compared mT5 and ByT5, which share architecture and data but differ in tokenization, finding that ByT5 requires more layers to encode morphological information and performs differently across languages. Recent work has explored latent tokenizer language models, which learn to dynamically segment byte sequences into variable-length latent tokens during training (Nawrot et al., 2023; Pagnoni et al., 2025; Minixhofer et al., 2025). As these methods couple tokenization with architectural design, they are not included in TokSuite to maintain controlled comparison.

**Tokenization Robustness and Vulnerabilities:** Chai et al. (2024) studied LM sensitivity to typographical errors and ambiguities caused by internal token structure, finding that

scaling model parameters mitigates but does not eliminate this sensitivity. Wang et al. (2025) developed adversarial examples that exploit tokenization vulnerabilities through "trap words", concatenating two vocabulary tokens to form a different existing token, successfully degrading state-of-the-art LM performance, particularly in Chinese. Geh et al. (2025) demonstrated that non-canonical segmentations can preserve semantic meaning while evading safety alignment, bypassing defenses like LlamaGuard and ShieldGemma. Prior work has also evaluated LM vulnerability to noise (Dhole et al., 2023; Wang et al., 2021a;b). While prior work focuses on adversarial examples and synthetic perturbations, TokSuite covers real-world, language-specific variations curated by native speakers.

*Decoding Algorithms:* Token Healing[1] and byte-level probability computation (Phan et al., 2025) address tokenization issues during generation. However, they provide limited improvement on TokSuite (details in Appendix F).

# 7. Discussion

A key limitation of studying off-the-shelf tokenizers is that real-world tokenizers differ along multiple dimensions simultaneously across vocabulary size, algorithm, normalization strategy, pre-tokenization rules, and training corpus making it difficult to attribute performance differences to any single design choice. Unlike prior work (Ali et al., 2024; Schmidt et al., 2024) that train tokenizers from scratch to isolate individual properties, we study tokenizers in the realistic settings practitioners actually encounter. Despite these entangled factors, our study uncovers statistically significant and meaningful differences in robustness across tokenizers, providing a useful reference for practitioners selecting among existing options. A fully controlled factorial study, training tokenizers from scratch while varying one design factor at a time, is beyond the scope of our work.

Despite spanning a diverse set of tokenizers, robustness scores converge across several perturbation categories, particularly in English, suggesting that the dominant design choices in the field have reached a certain equilibrium. The differences that emerge are not explained by standard intrinsic metrics that focus on compression, like fertility, vocabulary size, parity, PCW in Appendix B.

The most striking evidence is that the best robustness is achieved by a tokenizer with one of the smallest vocabularies (32,000 tokens) trained exclusively on English text, albeit at the cost of lower canonical performance. We speculate this is driven by TokenMonster's tokenization consistency, the degree to which the same surface form maps to the

---

[1] https://github.com/guidance-ai/guidance/blob/main/notebooks/art_of_prompt_design/prompt_boundaries_and_token_healing.ipynb

same token sequence, achieved through a distillation-based vocabulary construction and an ungreedy algorithm that explicitly limits token sequence variation for the same word. For example, many modern tokenizers treat `hello` and `_hello` (with a leading space) as distinct tokens, whereas TokenMonster unifies such variations.

Our findings suggest several practical takeaways for practitioners working on multilingual language model development. First, tokenizer choice has a measurable and statistically significant impact on model robustness that is independent of model scale as we observe similar robustness scores across 300M, 1B, and 7B parameter models trained with the same tokenizer (Table 22). Second, while the text budget consumed during training affects canonical performance, it does not explain robustness differences, as our text-matched experiments in Appendix G.2 confirm. Third, robustness to surface-level variations is a meaningful criterion to consider alongside compression-oriented metrics when selecting or evaluating a tokenizer for multilingual deployment.

## 8. Future Work & Limitations

`TokSuite` models are trained exclusively on five languages with higher mixing rates than massive multilingual models (for example, the highest mixing rate across *all* languages in mT5 (Xue et al., 2021)'s training was less than 5%). This setup may underestimate multilingual interference effects present in more realistic settings, where cross-lingual interference could degrade performance.

While additional training data may alleviate some vulnerabilities, tokenizers provide a cost-free inductive bias that fundamentally shapes robustness and efficiency. Critically, intrinsic properties like compression rates directly constrain information processing within token budgets, leaving certain languages systematically under-served within the same token budget.

While coding and math tasks present interesting challenges related to non-natural text and whitespace handling, we excluded them from our benchmark due to inconsistent model performance at the scale we considered. More broadly, `TokSuite` is designed for *base models* using multiple-choice questions that yield reliable baseline performance across all model sizes considered. Extending `TokSuite` to generation-focused evaluations, instruction-following settings, code domains, and larger model scales remains an important direction for future work.

## 9. Conclusion

Despite tokenization's fundamental role in language model behavior, practitioners commonly adopt off-the-shelf tokenizers without a systematic understanding of their impact.

To address this, we release `TokSuite`: 14 identical language models differing only in their tokenizer, plus a benchmark curated by native speakers probing natural variations that capture orthographic and morphological challenges across 5 languages and technical domains. Our results show that tokenizer design can matter more than vocabulary size. For example, an English-only tokenizer (TokenMonster) outperformed larger multilingual ones on certain perturbations, while byte-level models proved more robust to multilingual noise and subword fragmentation. Technical content analysis revealed critical vulnerabilities where trivial formatting differences caused catastrophic performance degradation. Our work provides clear evidence that tokenizer choice directly impacts model robustness and capability across diverse contexts and will support future work on understanding the impact of tokenization on LM performance.

## Acknowledgements

We thank Haokun Liu for annotating the Chinese benchmarks. We are also grateful to all contributors who supported the collection, validation, and annotation of the Farsi dialectal variations dataset, including Mohammad Azimi (Shirazi), Mohammad Farahani (Araki), Negin Memar (Isfahani), Mobina Salavati (Sorkheyi), Hosain Zaman (Dezfouli), Abolfazl Aghdaee (Kashani), Mahda Hekmatshoar (Sabzevari), Parsa Zahedi (Mazandarani), Amir Kabirian (Kermanshahi), and Donya Esfandiarpour (Kermani). Their expertise across diverse regional dialects was essential to ensuring the linguistic diversity and quality of the benchmark. We also thank Laura De Santis for contributions to the Italian subset, and Timur Cinay and Hazal Çintaş for their support with Turkish dialects.

Resources used in preparing this research were provided, in part, by the Province of Ontario, the Government of Canada through CIFAR, the Digital Research Alliance of Canada, and companies sponsoring the Vector Institute.

## Impact Statement

In this work, we present `TokSuite`, a benchmark and model collection designed to isolate the influence of tokenization on language model behavior in multilingual real-life settings. By providing a controlled experimental setup and open-sourcing our models, we enable a more rigorous study of how "off-the-shelf" tokenizer design choices impact downstream performance. Moreover, `TokSuite` covers five diverse languages, English, Chinese, Farsi, Italian, and Turkish, to highlight tokenization dependencies. We intend for this study to serve as an initial effort in establishing a standardized methodology for decoupling tokenizer effects, encouraging the development of future multilingual and language-specific benchmarks that target the unique

linguistic nuances and character-level perturbations of individual scripts.

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

**Appendix Contents**

*Table 2.* Comprehensive Overview of Selected Tokenizers. Part A: Basic Properties

| Tokenizer | Method | Vocab. Size | OOV Handling | Language(s) | Pretokenization |
|---|---|---|---|---|---|
| ByT5 | Bytes | 259 (XS) | Bytes | LA. | None (raw bytes) |
| TokenMonster | Custom | 32,000 (S) | Ignores Unknowns | English-Only | None (boundaries are learned) |
| Phi-3 | BPE | 32,064 (S) | Byte-fallback | Multilingual | SentencePiece |
| GPT-2 | BPE | 50,257 (M) | Byte-fallback | English-Only | GPT-2 |
| Comma | BPE | 64,000 (M) | Byte-fallback | Multilingual | GPT-4 |
| mBERT | WordPiece | 110,000 (M) | [UNK] | Multilingual | BERT |
| Llama-3.2 | BPE | 128,256 (M) | Byte-fallback | Multilingual | GPT-4 |
| Tekken | BPE | 130,000 (M) | Byte-fallback | Multilingual | GPT-4o* |
| Qwen-3 | BPE | 151,646 (L) | Byte-fallback | Multilingual | GPT-4* |
| GPT-4o | BPE | 200,000 (L) | Byte-fallback | Multilingual | GPT-4o |
| BLOOM | BPE | 250,680 (L) | Byte-fallback | Multilingual | BLOOM |
| Aya | BPE | 255,029 (L) | Byte-fallback | Multilingual | GPT-2 |
| Gemma-2 | Unigram△ | 256,128 (L) | Byte-fallback | Multilingual | SentencePiece |
| XGLM | Unigram | 256,008 (L) | Byte-fallback | Multilingual | SentencePiece |

[1] Vocabulary bucket is indicated in ( ).
[2] OOV = Out-of-vocabulary
[3] LA. = Language-agnostic
△ While the paper (Team et al., 2024) describes the tokenizer as Unigram, the HuggingFace implementation specifies BPE.

## A. Tokenizer Processing Glossary

**Pretokenization**

**BERT**          Pre-tokenization splits are based on whitespace and punctuation.

**GPT-2**          Pre-tokenization splits are done on whitespace and transitions between letters, numbers, and punctuation.

**GPT-4**          GPT-4 pre-tokenization follows GPT-2's approach, but it also creates a new token after 3 contiguous digits.

*Table 3.* Comprehensive Overview of Selected Tokenizers. Part B: Processing Details. See Appendix A for detailed explanations of tokenization processing terminologies and methodologies.

| Tokenizer Name | Numbers | Contractions | Unicode Norm. | Whitespace | Zerowidth chars |
|---|---|---|---|---|---|
| ByT5 | N/A | N/A | None | N/A | 3 Bytes |
| TokenMonster | Learned | Learned | NFD | Learned | Token |
| Phi-3 | Split | Learned | None | Manual | Token |
| GPT-2 | Group | GPT-2 | None | Individual | Token |
| Comma | Group by 3 | GPT-4 | None | Learned | Token |
| mBERT | Learned | Composed | None | Normalized | Normalized/Removed |
| Llama-3.2 | Group by 3 | GPT-4 | None | Learned | Token |
| Tekken | Split | GPT-4* | None | Learned | Token |
| Qwen-3 | Split | GPT-4 | NFC | Learned | Token |
| GPT-4o | Group by 3 | Learned | None | Learned | Token |
| BLOOM | Learned | Learned | None | Learned | Token |
| Aya | Split | GPT-2 | NFC | Learned | Token |
| Gemma-2 | Split | Learned | None | Manual | Token |
| XGLM | Learned | Learned | NFKC | Normalized | Normalized/Removed |

Note that Qwen 3 uses the same pretokenization as GPT-4, but does not split numbers into groups of three.

**GPT-4o**     GPT-4o pre-tokenization follows that of GPT-4, but specific contractions ('s, 'd, 'm, 't, 'll, 've, 're) are not split from the preceding word. Note that Tekken uses the same pre-tokenization methods as GPT-4o, but without special case handling of the specific english contractions.

**BLOOM**     Pre-tokenization splits are done based on whitespace and punctuation like commas and periods.

**SentencePiece**     Pre-tokenization splits are done on whitespace, and at transitions between letters, numbers and punctuation.

## Numbers Processing

**Split**     Numbers are deterministically broken down into individual digits which are each treated as single tokens.

**Group**     Numbers are deterministically split from adjoining text during pre-tokenization. The learning algorithm then determines which numbers become single tokens and which are further tokenized.

**Group by 3**     Similar to **Group**, but contiguous digits are split into groups of 3 during pre-tokenization. Again, the learning algorithm then determines which numbers are single tokens. For example, "username12345" is pre-tokenized into "username", "123", and "45", but "123" is not a token in $\mathcal{V}$ yielding a final token stream of "username", "1", "23", "45".

**Learned**     Numbers are not automatically segmented from surrounding text. Thus, the learning algorithm determines token boundaries for letters and numbers jointly. This can result in tokens that include both characters and digits.

## Contractions Processing

**GPT-2**     A selected number of English contractions ('s, 'd, 'm, 't, 'll, 've, 're) are manually split into their own tokens. The learning algorithm then decides if they should be their own token or if it should be broken down further. This makes it impossible to have a token like "I'll".

**GPT-4**     Uses GPT-4's contraction processing method. The name set of contractions are explicitly handled, but the regex is implemented differently. Note that Tekken uses the GPT-4 regex without special casing english contractions; however, it still results in splitting contractions from the base during pre-tokenization.

**Learned**     Contractions are not manually split from the base word; the learning algorithm decides if the contraction should be its own token or a composition.

**Composed**     The pre-tokenization splits all contractions into multiple tokens (base, apostrophe, and contraction, e.g., he'll → "he", "'", "ll"), which cannot be merged back together in the learning algorithm.

### Unicode Normalization

**None**
No Unicode normalization is applied; characters are processed exactly as they appear in the input. Note that this can result in $\mathcal{V}$ containing multiple tokens that are visually the same, but differ in their underlying bytes, for example two "é" tokens, but one is represented by a single code point while the other is represented as the composition of "e" and "´".

**NFD**
*Normalization Form Decomposed*: Unicode characters are decomposed into their constituent parts (base characters + combining marks separately).

**NFC**
*Normalization Form Composed*: Unicode characters are composed into their canonical combined form (base characters + combining marks merged when possible).

**NFKC**
*Normalization Form Compatibility Composed*: Similar to NFC but also applies compatibility mappings, converting visually similar characters to their canonical equivalents before composition. Note that this can result in lossy detokenization as characters like "$^2$" are mapped to "2".

### Whitespace Treatment

**Normalized**
Whitespace like tabs, newlines, and contiguous spaces are normalized to a single space. This results in lossy detokenization and often stops the downstream model from understanding domains with meaningful whitespace such as code.

**Learned**
Each piece of contiguous whitespace is segmented into a single token during pre-tokenization, then the learning algorithm decides how to subdivide them into individual tokens. This results in whitespace being preserved and allows for lossless detokenization.

**Manual**
The handling of whitespaces during pre-tokenization matches **Learned**, but pre-defined whitespace tokens of various sizes are used instead of learning them from the data. This results in whitespace being preserved and allows for lossless detokenization.

**Individual**
Whitespace is preserved, but each individual whitespace character is represented as its own token. This yields long token sequences for whitespace heavy inputs. This results in whitespace being preserved and allows for lossless detokenization.

### Zero-width Characters

**3 Bytes**
Zero-width characters are maintained in their original 3-byte representation.

**Token**
Zero-width characters are preserved and assigned as new tokens in the vocabulary.

**Normalized/Removed** Zero-width characters are either normalized to standard equivalents or completely removed.

## B. Intrinsic Tokenization Efficiency Metrics

Tokenizers exhibit varying degrees of compactness when segmenting text into tokens, resulting in notable disparities in model performance across languages and domains. To systematically evaluate these differences, we analyze several metrics across our selected pretrained tokenizers, focusing on our five languages.

We compute three primary intrinsic efficiency metrics using 10,000 parallel random samples from Flores200 (Team et al., 2022), split into "real" words via language-specific word-level tokenizers from the DataTrove library (Penedo et al., 2024a):

- Subword fertility (SF): is the mean number of tokens used to represent each "real" text word. This reflects how aggressively a tokenizer segments words. The theoretical minimum is 1, implying that the tokenizer's vocabulary encompasses every word in the reference text (Penedo et al., 2025).

- Parity: evaluates whether a tokenizer processes equivalent sentences fairly across languages. Achieved when the ratio of tokenized lengths $\frac{|T(s_A)|}{|T(s_B)|} \approx 1$ for parallel sentence sets $s_A$ and $s_B$ from languages A and B (Ali et al., 2024).

- Proportion of continued words (PCW): the proportion of "real" text words that require two or more tokens for encoding. This metric indicates how frequently a tokenizer splits words. A score of 0 means no splitting occurs, while a score of 1 means every word is split (Rust et al., 2021).

The intrinsic metrics reflect a tokenizer's efficiency in processing a language and are critical factors in tokenizer selection, as they directly impact an LM's computational cost, context window utilization, and representation quality. Table 4 reveals substantial disparities in how our tokenizers handle our target languages. ByT5 and tokenizers with smaller vocabularies (TokenMonster and Phi-3) exhibit significantly higher subword fertility and PCW scores, particularly for non-English languages. ByT5 requires 7.72 tokens per word in Farsi compared to 4.40 in English. Multilingual-specialized tokenizers (mBERT, XGLM) demonstrate superior language parity, with XGLM achieving near-optimal parity scores (1.18 average) and mBERT showing the lowest average subword fertility (1.54). We don't observe significant correlations between intrinsic metrics and the downstream performance on TokSuite.

Notably, vocabulary size alone does not guarantee efficiency; Qwen-3 and Gemma-2, despite having large vocabularies (>150K), show comparable or worse performance than smaller vocabulary tokenizers like mBERT on certain metrics. We also observe higher fertility and PCW scores for morphologically rich languages (e.g., Turkish, Farsi) compared to English.

*Table 4.* Multilingual Tokenizers Comparison on Flores200 Using Intrinsic Tokenizer Efficiency Metrics. sf denotes subword fertility, pcw denotes proportion of continued words, and parity is measured against English parallel samples. Summary statistics report average values across all languages. Lower is better for all metrics. Bold font highlights the best performance in each row. Models are ordered from smallest to largest vocabulary size, left to right. Vocabulary size is categorized as XS, S, M, and L for $< 1K$, $1K–50K$, $50K–150K$, and $> 150K$ tokens, respectively.

| Tokenizer Vocab. Size | ByT5 XS | TokenMonster S | Phi-3 S | GPT-2 M | Comma M | mBERT M | Llama-3.2 M | Tekken M | Qwen-3 L | GPT-4o L | BLOOM L | Aya L | Gemma-2 L | XGLM L |
|---|---|---|---|---|---|---|---|---|---|---|---|---|---|---|
| English sf | 4.40 | 1.75 | 1.24 | 1.30 | 1.44 | 1.15 | 1.26 | 1.35 | 1.28 | 1.24 | 1.31 | 1.19 | **1.14** | 1.23 |
| English pcw | 0.87 | 0.56 | 0.16 | 0.23 | 0.34 | **0.10** | 0.20 | 0.27 | 0.21 | 0.20 | 0.25 | 0.15 | 0.11 | 0.21 |
| | | | | | | | | | | | | | | |
| Chinese sf | 5.00 | 4.92 | 3.44 | 3.54 | 2.45 | 1.68 | 1.49 | 1.64 | 1.21 | 1.44 | **1.16** | 1.23 | 1.28 | 2.19 |
| Chinese pcw | 0.98 | 0.97 | 0.97 | 0.82 | 0.58 | 0.55 | 0.35 | 0.41 | 0.16 | 0.32 | **0.13** | 0.18 | 0.21 | 0.87 |
| Chinese parity | 0.94 | 4.99 | 2.03 | 3.21 | 1.94 | 1.40 | 1.29 | 1.43 | 1.02 | 1.27 | **0.93** | 1.05 | 1.09 | 1.15 |
| | | | | | | | | | | | | | | |
| Turkish sf | 6.49 | 4.31 | 3.20 | 3.20 | 3.29 | 1.99 | 2.38 | 2.44 | 2.58 | 2.33 | 2.71 | 2.17 | 2.23 | **1.69** |
| Turkish pcw | 0.87 | 0.80 | 0.76 | 0.76 | 0.78 | **0.52** | 0.72 | 0.73 | 0.74 | 0.71 | 0.72 | 0.68 | 0.69 | **0.52** |
| Turkish parity | **1.12** | 3.34 | 2.11 | 2.45 | 2.21 | 1.37 | 1.39 | 1.50 | 1.63 | 1.43 | 1.98 | 1.21 | 1.39 | **1.12** |
| | | | | | | | | | | | | | | |
| Farsi sf | 7.72 | 7.74 | 4.77 | 4.91 | 4.43 | 1.53 | 1.94 | 1.92 | 2.45 | 1.93 | 2.01 | 1.85 | 1.83 | **1.36** |
| Farsi pcw | 0.95 | 0.94 | 0.93 | 0.90 | 0.90 | 0.31 | 0.58 | 0.58 | 0.67 | 0.57 | 0.58 | 0.53 | 0.53 | **0.28** |
| Farsi parity | 1.72 | 9.45 | 4.08 | 5.35 | 4.31 | 1.38 | 1.52 | 1.47 | 2.63 | 1.55 | 1.80 | 1.48 | 1.45 | **1.21** |
| | | | | | | | | | | | | | | |
| Italian sf | 4.78 | 2.50 | 1.64 | 1.99 | 2.05 | **1.34** | 1.81 | 1.77 | 1.83 | 1.71 | 1.75 | 1.61 | 1.54 | 1.36 |
| Italian pcw | 0.84 | 0.63 | 0.42 | 0.57 | 0.59 | **0.23** | 0.55 | 0.53 | 0.55 | 0.52 | 0.51 | 0.47 | 0.41 | 0.32 |
| Italian parity | **1.19** | 2.30 | 1.48 | 2.02 | 1.87 | 1.28 | 1.62 | 1.40 | 1.64 | 1.47 | 1.63 | 1.31 | 1.33 | 1.24 |
| | | | | | | | | | | | | | | |
| Avg sf | 5.79 | 4.39 | 2.90 | 3.19 | 2.93 | **1.54** | 1.78 | 1.82 | 1.87 | 1.73 | 1.79 | 1.61 | 1.60 | 1.56 |
| Avg pcw | 0.90 | 0.78 | 0.62 | 0.66 | 0.64 | **0.34** | 0.48 | 0.50 | 0.47 | 0.46 | 0.44 | 0.40 | 0.39 | 0.46 |
| Avg parity | 1.27 | 5.31 | 2.54 | 3.44 | 2.74 | 1.36 | 1.46 | 1.45 | 1.73 | 1.43 | 1.59 | 1.26 | 1.32 | **1.18** |

## C. Model Training

Each model is trained on a single 8×H100 GPU node using the Lingua framework (Videau et al., 2024). Training takes 3 to 5 days depending on the tokenizer's embedding table size, which varies with vocabulary size. Our training code and configs for all fourteen models and ablations in the following sections are available at https://github.com/r-three/lingua/tree/toksuite.

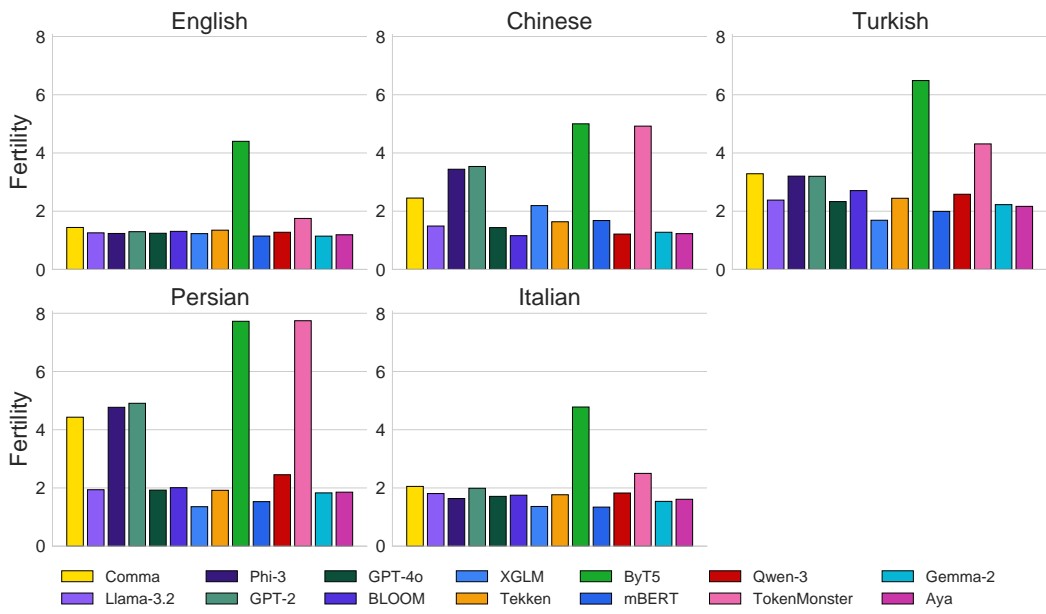

*Figure 3.* Tokenizer performance comparison across languages using Flores200 dataset measured by fertility. Lower is better.

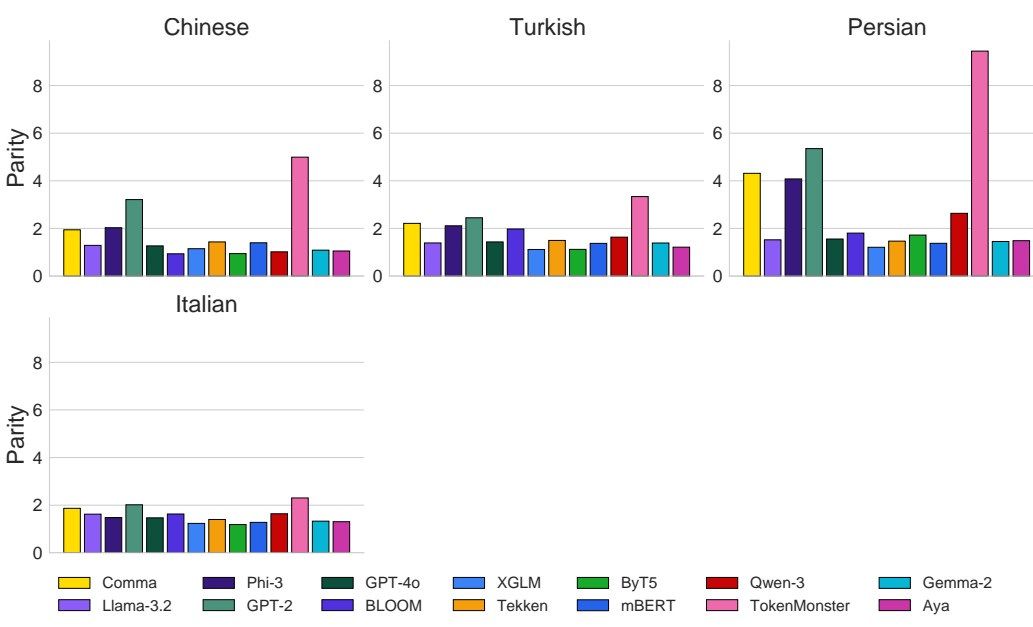

*Figure 4.* Tokenizer performance comparison across languages using Flores200 dataset measured by parity. Lower is better.

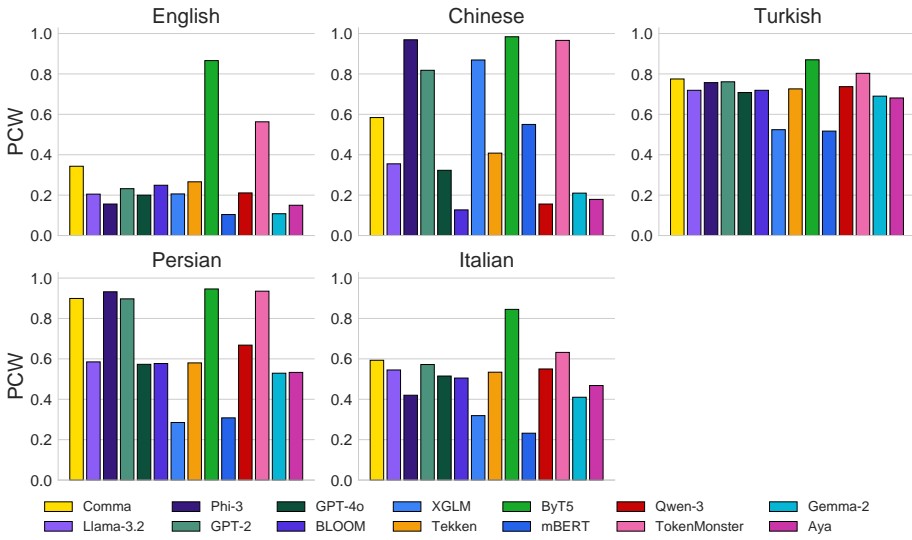

*Figure 5.* Tokenizer performance comparison across languages using Flores200 dataset measured by the proportion of continued words. Lower is better.

### C.1. Model Initialization

We use the same initialization strategy as the Llama-1B configuration, however, we first create a shared initialization where the size of the embedding table and the final output layer is the size of the *super vocabulary*, $|E_{sv}| = |\mathcal{SV}|$. Each model then uses the parameter values from this shared initialization for most layers. The embedding table for an individual model, $E$, is initialized by selecting the appropriate rows from the super vocabulary embedding table. Thus after initialization, $E(x) = E_{sv}(sv(X))$. This results in a shared initialization for all models, including the initial embedding value for any shared tokens.

### C.2. Model Performance

We report the bits-per-byte (BPB) loss curves for `TokSuite` models throughout training in Figure 6. BPB normalizes the cross-entropy loss by $\log_2 |V|$ to account for differences in vocabulary size, enabling a fair comparison. All models converge to similar BPB values, with no loss spikes or divergent behavior observed during training. This suggests that all models are comparably well-trained, and that observed differences in downstream performance and robustness are attributable to tokenizer choice rather than differences in training stability or capabilities.

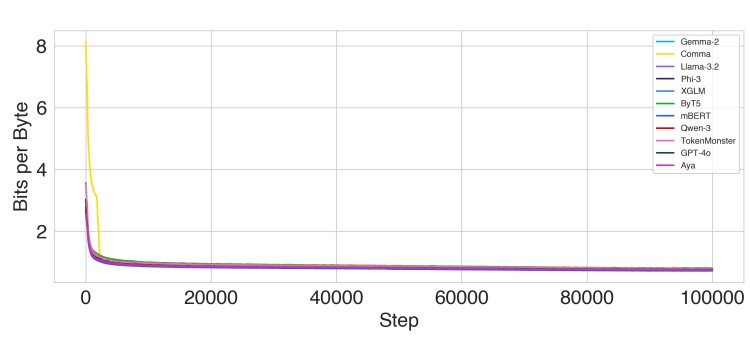

*Figure 6.* Bit-per-Byte (BPB) during training

We evaluate all models on standard English reasoning tasks (HellaSwag (Zellers et al., 2019), ARC Easy/Challenge (Clark et al., 2018), PIQA (Bisk et al., 2020)), multilingual natural language inference (XNLI (Conneau et al., 2018) in English,

Turkish, and Chinese), reading comprehension (Belebele (Bandarkar et al., 2024) in English, Italian, Farsi, Turkish, and Chinese), and a multilingual reasoning benchmark (INCLUDE Base 44 (Romanou et al., 2025) in Chinese, Italian, and Turkish) in Figure 7. Although models achieve sufficient performance on easier English reasoning tasks, their performance on more advanced multilingual reading comprehension and reasoning benchmarks hardly exceeds the random baseline. Results for Belebele and INCLUDE are omitted from the figures for visual clarity, as their performance trends were consistent with this pattern, slightly above random but not competitive across languages. Note that models with larger vocabulary (Aya, XGLM, mBERT, Gemma-2, GPT-4o, and Llama-3.2) tend to perform better on the downstream tasks, with TokenMonster and Tekken falling slightly behind.

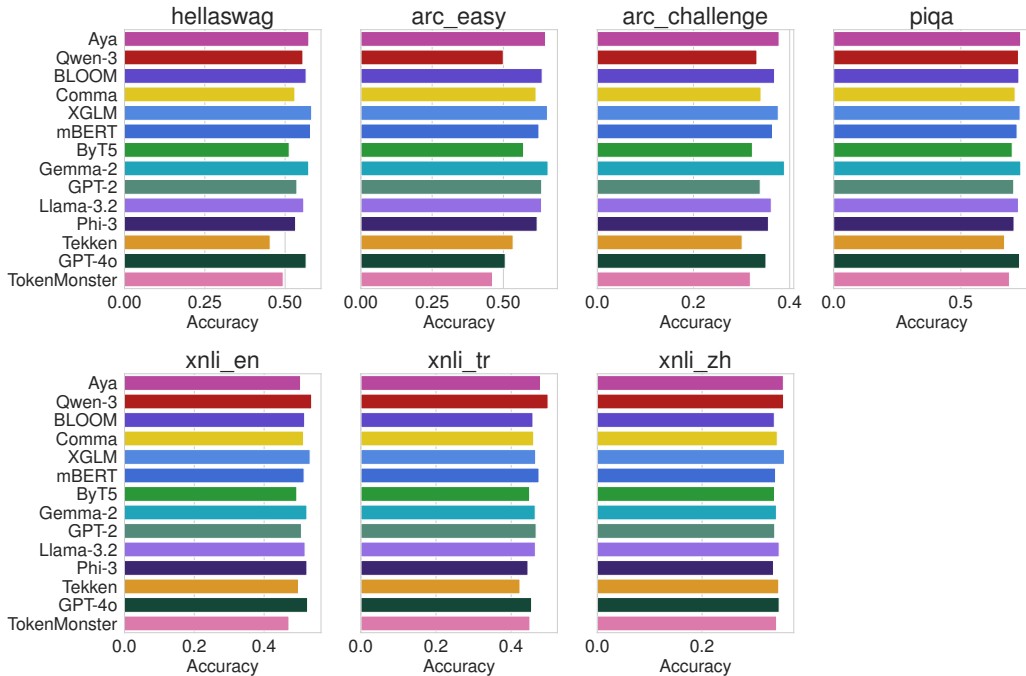

*Figure 7.* Model Performance on Multilingual Benchmarks

## C.3. Training Data Consumption and Fairness

*Table 5.* Data consumed during training across different tokenizers

| Model | Data Consumed (GB) |
|---|---|
| ByT5 | 100.00 |
| TokenMonster | 215.61 |
| GPT-2 | 263.81 |
| Comma | 278.59 |
| Phi-3 | 287.38 |
| Qwen-3 | 411.23 |
| Tekken | 437.00 |
| BLOOM | 437.66 |
| mBERT | 445.80 |
| Llama-3.2 | 452.26 |
| GPT-4o | 467.10 |
| Aya | 468.44 |
| Gemma-2 | 471.38 |
| XGLM | 477.22 |

The training process utilizes a deterministic data loader, sampling documents in the same order for all models. However,

the varying compression efficiency of each tokenizer results in variation in the tokenized batch streams, which leads to different total numbers of actual UTF-8 bytes consumed for a fixed token budget. This consumption difference is an inherent consequence of tokenizer design and is unavoidable when comparing tokenizers under current LLM training practice (fixed token budget). To quantify this trade-off, we reconstructed the entirety of the text data consumed by each model [2], detokenized each batch, and computed the total UTF-8 bytes seen. ByT5 consumed 100 GB, while others ranged from $\sim 215$ GB to $\sim 477$ GB, with the exact numbers provided in Table 5. Crucially, models that consumed a greater total byte count were not necessarily the best performers (Table 1), suggesting that the tokenization strategy plays a larger role than the sheer volume of raw input.

To further validate that our findings are not artifacts of differential data consumption, we conducted a controlled experiment where models train on identical content measured in bytes. In Appendix G.2, we train four models using a fixed text budget rather than a token budget, which results in varying token counts but ensures models consume identical raw information. This alternative training regime confirms that tokenization robustness patterns remain consistent regardless of whether training is normalized by tokens or text, reinforcing that robustness is primarily determined by tokenizer architecture rather than data volume.

## D. `TokSuite` Benchmark Details

### D.1. Question Style and Difficulty

The `TokSuite` benchmark comprises straightforward multiple-choice text completion questions. Below we present the canonical English questions that form our English subset, which are subsequently translated into Farsi (FA), Italian (IT), Turkish (TR), and Chinese (ZH). The fourteen models demonstrate strong performance on the canonical questions in English and Italian (Figure 8), while the canonical accuracy on Farsi, Turkish, and Chinese is slightly behind. Higher subword fertility, PCW, and parity scores in these three languages (see Table 4) suggest that the models are likely to consume less information measured in raw bytes in these languages.

300 Dr Smith is a doctor. Occupation of Dr Smith is: **doctor**, teacher, judge, lawyer

301 The color of the sky is: **blue**, red, green, yellow

302 The price of this house is 1,028,415 dollars. The cost of this house is: **1,028,415 dollars**, 1.028.415 dollars, 1,028,411 dollars, 1,028.415 dollars

303 Today's date is 29/08/2025. Today is: **29/08/2025**, 19/08/2025, 26/08/2025, 29/09/2025

304 The number of continents on Earth is: **7**, 5, 6, 8

305 The capital city of Iran is: **Tehran**, Mashhad, Baghdad, Isfahan

306 The number of days in a week is: **7**, 5, 6, 8

307 The number of hours in a day is: **24**, 20, 25, 30

308 The number of legs a cow has is: **4**, 8, 3, 5

309 The number of minutes in 2 hours is: **120**, 100, 140, 90

310 The number of months in a year is: **12**, 10, 11, 13

311 The number of seconds in a minute is: **60**, 50, 100, 30

312 The number of sides a hexagon has is: **6**, 5, 7, 8

313 The number of sides a triangle has is: **3**, 2, 4, 5

314 In "I work at Apple", Apple is: **company**, person, city, fruit

315 In "I work at Google", Google is: **company**, person, city, fruit

---

[2]https://hf.co/collections/toksuite/training-data-detokenized

316 In "Microsoft released a new update", Microsoft is: **company**, person, place, date

317 In "The cat sat on the mat", the subject is: **the cat**, sat, the mat, on

322 The gas humans need to breathe to live is: **oxygen**, methane, helium, hydrogen

323 10% of 100 is: **10**, 5, 15, 20

324 25% of 80 is: **20**, 15, 25, 30

326 Chad's capital is: **N'Djamena**, Moundou, Abéché, Ngama

327 The capital of France is: **Paris**, London, Berlin, Rome

328 The capital of Japan is: **Tokyo**, Kyoto, Osaka, Hiroshima

329 The capital of Turkey is: **Ankara**, İstanbul, İzmir, Bursa

330 The chemical formula for water is: **H2O**, CO2, NaCl, O2

331 The intent in "What time does the store close?" is: **get information**, make purchase, book appointment, file complaint

332 The largest mammal in the world is: **blue whale**, dolphin, giraffe, bear

333 The unit of measurement for temperature in the International System is: **Kelvin**, Celsius, meter, Rankine

334 The country whose space agency is NASA is: **United States**, Russia, China, Japan

335 The language spoken in Brazil is: **Portuguese**, Spanish, French, Italian

336 The metal with chemical symbol 'Fe' is: **iron**, lead, zinc, gold

337 The organ in the human body that pumps blood is: **heart**, liver, lungs, kidneys

338 The planet closest to the Sun in our solar system is: **Mercury**, Venus, Mars, Earth

339 The largest planet in the Solar System is: **Jupiter**, Earth, Saturn, Mars

340 The process that allows plants to produce their own food using sunlight is: **photosynthesis**, respiration, digestion, fermentation

341 The author who wrote the play "Romeo and Juliet" is: **William Shakespeare**, Charles Dickens, Mark Twain, Jane Austen

342 What bees produce is: **honey**, milk, silk, wax

343 What plants need from the air to make food is: **carbon dioxide**, nitrogen, hydrogen, helium

344 In "Can you please book a flight to Paris?", the person wants to: **make a booking**, go shopping, file a complaint, cancel reservation

### D.2. Benchmark Composition

In Table 7, we list the composition of the categories and perturbations in TokSuite. The multilingual parallel dataset comprises 80% of the dataset, while the remaining part covers math, and STEM.

## E. Detailed Benchmark Results

In this section, we provide case studies for each category in Section 4.1.

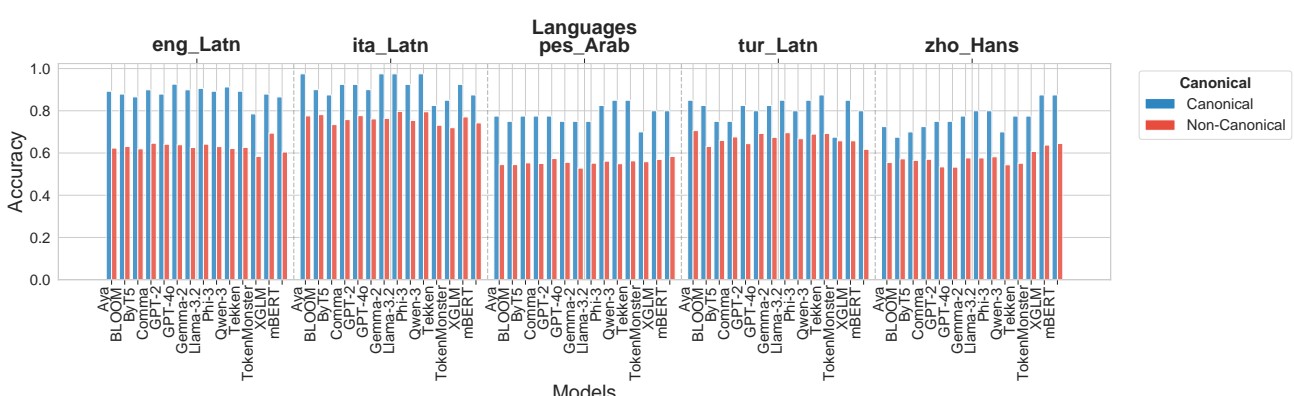

*Figure 8.* Accuracies of models on canonical versus perturbed questions across the English (eng_Latn), Italian (ita_Latn), Farsi (pes_Arab), Turkish (tur_Latn), and Chinese (zho_Hans) questions in `TokSuite`. Note that eng_Latn here is averaged over canonical questions in English, Math and STEM subsets, refer to Table 6 for more fine-grained numbers.

*Table 6.* Canonical accuracy on `TokSuite` (higher better).

| Model | EN | FA | ZH | IT | TR | STEM | Math | Avg |
|---|---|---|---|---|---|---|---|---|
| Qwen-3 | 0.95 | **0.85** | **0.97** | 0.85 | 0.70 | **0.95** | 0.89 | **0.88** |
| Llama-3.2 | **1.00** | 0.75 | **0.97** | 0.85 | 0.80 | 0.81 | 0.89 | 0.87 |
| XGLM | 0.95 | 0.80 | 0.93 | 0.85 | **0.88** | 0.71 | 0.89 | 0.86 |
| Gemma-2 | 0.97 | 0.75 | **0.97** | 0.82 | 0.78 | 0.81 | 0.89 | 0.86 |
| Phi-3 | 0.95 | 0.82 | 0.93 | 0.80 | 0.80 | 0.81 | 0.89 | 0.86 |
| Tekken | 0.95 | **0.85** | 0.82 | **0.88** | 0.78 | 0.81 | 0.89 | 0.85 |
| GPT-4o | 0.93 | 0.75 | 0.90 | 0.80 | 0.75 | 0.90 | **0.93** | 0.85 |
| GPT-2 | 0.95 | 0.78 | 0.93 | 0.82 | 0.75 | 0.81 | 0.86 | 0.84 |
| Aya | 0.97 | 0.78 | **0.97** | 0.85 | 0.72 | 0.67 | 0.91 | 0.84 |
| Comma | 0.95 | 0.78 | 0.93 | 0.75 | 0.72 | 0.86 | 0.89 | 0.84 |
| mBERT | 0.93 | 0.80 | 0.88 | 0.80 | **0.88** | 0.67 | 0.89 | 0.83 |
| BLOOM | **1.00** | 0.75 | 0.90 | 0.82 | 0.68 | 0.81 | 0.84 | 0.83 |
| ByT5 | 0.90 | 0.78 | 0.88 | 0.75 | 0.70 | 0.71 | 0.89 | 0.80 |
| TokenMonster | 0.88 | 0.70 | 0.85 | 0.68 | 0.78 | 0.67 | 0.77 | 0.76 |
| Avg | 0.95 | 0.78 | 0.92 | 0.81 | **0.76** | 0.79 | 0.88 | 0.84 |

*Table 7.* Benchmark statistics by language and domain

| Language/Domain | Total Examples | Perturbations |
|---|---|---|
| English | 1,180 | 42 types |
| Chinese | 485 | 18 types |
| Turkish | 638 | 21 types |
| Italian | 1,088 | 19 types |
| Farsi | 747 | 15 types |
| Math | 189 | 5 types |
| STEM | 614 | 25 types |
| Total | 4941 | |

*Table 8.* Tokenization robustness under different input mediums or writing systems, granular version of **Input** in Table 1. Values represent relative performance drop ($\frac{\text{Acc}_{\text{can}} - \text{Acc}_{\text{pert}}}{\text{Acc}_{\text{can}}}$); lower values indicate greater robustness. 'Traditional' refers to traditional Chinese characters instead of simplified.

| Model | Romanization | | Number Romanization | English Keyboard | | Arabic Keyboard | Traditional | Avg |
|---|---|---|---|---|---|---|---|---|
| | FA | ZH | FA | TR | IT | FA | ZH | |
| TokenMonster | 0.46 | 0.58 | -0.10 | **-0.04** | 0.21 | **0.25** | **0.02** | **0.20** |
| Comma | 0.42 | 0.59 | 0.21 | 0.03 | 0.24 | 0.42 | 0.04 | 0.28 |
| GPT-4o | 0.57 | 0.67 | -0.03 | 0.22 | **0.09** | 0.43 | 0.03 | 0.28 |
| Llama-3.2 | 0.60 | 0.66 | **-0.23** | 0.24 | 0.14 | 0.53 | 0.09 | 0.29 |
| BLOOM | 0.63 | 0.48 | 0.08 | 0.21 | 0.15 | 0.40 | 0.10 | 0.29 |
| Aya | 0.55 | 0.62 | 0.01 | 0.06 | 0.16 | 0.55 | 0.12 | 0.29 |
| ByT5 | 0.61 | **0.46** | 0.21 | 0.13 | 0.15 | 0.39 | 0.18 | 0.30 |
| Tekken | 0.59 | 0.61 | 0.00 | 0.17 | 0.20 | 0.44 | 0.18 | 0.31 |
| Gemma-2 | **0.40** | 0.52 | 0.28 | 0.24 | 0.19 | 0.47 | 0.18 | 0.32 |
| Phi-3 | 0.58 | 0.66 | 0.25 | 0.06 | 0.24 | 0.39 | 0.09 | 0.33 |
| XGLM | 0.59 | 0.63 | 0.13 | 0.29 | 0.19 | 0.41 | 0.10 | 0.34 |
| mBERT | 0.44 | 0.60 | 0.42 | 0.22 | 0.18 | 0.50 | 0.10 | 0.35 |
| GPT-2 | 0.61 | 0.67 | 0.31 | 0.30 | 0.16 | 0.32 | 0.11 | 0.35 |
| Qwen-3 | 0.68 | 0.64 | 0.19 | 0.15 | 0.19 | 0.47 | 0.18 | 0.36 |
| Avg | 0.55 | 0.60 | 0.12 | 0.16 | 0.18 | 0.43 | 0.11 | 0.31 |

*Table 9.* Tokenization robustness under errors from input mediums. Values represent relative performance drop ($\frac{\text{Acc}_{\text{can}} - \text{Acc}_{\text{pert}}}{\text{Acc}_{\text{can}}}$); lower values indicate greater robustness.

| Model | Homoglyphs | Zero-width chars. | | Avg |
|---|---|---|---|---|
| | EN | FA | ZH | |
| mBERT | 0.08 | **0.09** | 0.00 | **0.06** |
| Phi-3 | **0.03** | 0.21 | -0.06 | **0.06** |
| TokenMonster | 0.09 | 0.18 | -0.06 | 0.07 |
| BLOOM | 0.12 | 0.17 | -0.07 | 0.07 |
| XGLM | **0.03** | 0.19 | 0.03 | 0.08 |
| ByT5 | 0.06 | 0.32 | -0.11 | 0.09 |
| Comma | 0.05 | 0.32 | -0.07 | 0.10 |
| GPT-4o | 0.14 | 0.23 | -0.03 | 0.11 |
| Aya | 0.28 | 0.23 | **-0.14** | 0.12 |
| Gemma-2 | 0.15 | 0.27 | 0.03 | 0.15 |
| Llama-3.2 | 0.12 | 0.30 | 0.03 | 0.15 |
| GPT-2 | 0.13 | 0.23 | 0.13 | 0.16 |
| Tekken | 0.13 | 0.29 | 0.10 | 0.17 |
| Qwen-3 | 0.11 | 0.38 | 0.11 | 0.20 |
| Avg | 0.11 | 0.24 | -0.01 | 0.11 |

### E.1. Orthographic & Script Challenges

**Variations in Writing Systems or Input Mediums**    Table 8 examines tokenization robustness under orthographic and script challenges, focusing on variations in writing systems or input mediums where users employ non-native keyboards. For Chinese romanization, we write the full question and choices in Pinyin without tone markers as if the user only has access to an English keyboard with spaces between individual groups that constitute a character for easy segmentation. However, this segmentation aid does not improve tokenization robustness, as models still exhibit substantial performance degradation (0.60 relative accuracy drop on average) when processing romanized Chinese text compared to native scripts. For Farsi, we examine two romanization approaches: (1) Finglish-style romanization (FA column), where Farsi text is written using Latin characters following common transliteration practices used by native speakers on English keyboards, and (2) number-based romanization, where Farsi numerals replace corresponding characters (e.g., using digits like 2, 3, 7 as phonetic substitutes). We also evaluate cross-script keyboard constraints: Latin-script languages (Italian and Turkish) are tested with English keyboard layouts (TR, IT columns), while Farsi is tested with Arabic keyboard input (Arabic Keyboard column), reflecting common scenarios where users lack access to their native keyboard. Finally, the Traditional column assesses Chinese model performance when presented with Traditional Chinese characters instead of the standard Simplified Chinese characters used in training. Across these input medium variations, models show varying degrees of robustness, with average relative performance drops ranging from 0.11 (Traditional Chinese) to 0.60 (Chinese romanization).

**Homoglyphs and Zero-width Characters**    In Table 9, the errors due to input systems (like homoglyphs and zero-width characters) are presented. This category examines tokenization robustness under typographic irregularities: (1) homoglyphs in English, where visually identical characters from different Unicode scripts (e.g., Cyrillic 'o' vs. Latin 'o') replace their Latin counterparts, and (2) zero-width characters (invisible Unicode characters like zero-width spaces) inserted into Farsi and Chinese text. This category tests whether tokenizers can handle Unicode irregularities and visually deceptive characters-issues that arise from copy-pasting text across different systems, malicious input, or encoding errors. Models demonstrate relatively good robustness to homoglyphs (0.11 average drop) and Chinese zero-width characters (-0.01 average drop), but show moderate degradation with Farsi zero-width characters (0.24 average drop), likely because of its dual reliance on both white-space boundaries for word segmentation and contextual letter joining rules (where zero-width joiners/non-joiners are legitimately used), making tokenizers particularly sensitive to incorrectly placed invisible characters that can simultaneously disrupt both spacing patterns and character connectivity.

*Table 10.* Tokenization robustness to diacritics, granular version of Diacr. in Table 1 and wrong accents in Italian. Values represent relative performance drop ($\frac{\text{Acc}_{\text{can}} - \text{Acc}_{\text{pert}}}{\text{Acc}_{\text{can}}}$); lower values indicate greater robustness.

| Model | Diacritics | | Wrong accents | Avg |
|---|---|---|---|---|
| | FA | ZH | IT | |
| BLOOM | 0.33 | **0.37** | 0.08 | **0.26** |
| TokenMonster | **0.21** | 0.45 | 0.17 | 0.28 |
| GPT-2 | 0.42 | 0.50 | **-0.02** | 0.30 |
| Qwen-3 | 0.41 | 0.43 | 0.10 | 0.31 |
| ByT5 | 0.42 | 0.46 | 0.06 | 0.31 |
| mBERT | 0.31 | 0.57 | 0.06 | 0.31 |
| Gemma-2 | 0.43 | 0.42 | 0.10 | 0.32 |
| Phi-3 | 0.39 | 0.53 | 0.05 | 0.32 |
| Tekken | 0.47 | 0.48 | 0.07 | 0.34 |
| Aya | 0.45 | 0.48 | 0.10 | 0.34 |
| XGLM | 0.44 | 0.54 | 0.11 | 0.36 |
| GPT-4o | 0.47 | 0.57 | 0.08 | 0.37 |
| Comma | 0.39 | 0.48 | 0.30 | 0.39 |
| Llama-3.2 | 0.60 | 0.50 | 0.16 | 0.42 |
| Avg | 0.41 | 0.49 | 0.10 | 0.33 |

**Diacritics Perturbations**    Table 10 expands on diacritics perturbations, examining how tokenizers handle optional Farsi diacritics that are used to clarify pronunciation and phonetic details, Chinese tonal variations in the Pinyin format, and incorrect accent placement in Italian text. We test how tokenizers handle optional diacritics, where text remains valid with or without marks (e.g., marks placed above or below letters to clarify pronunciation and phonetic details such as short vowels (fatḥa for /a/, kasra for /e/, ḍamma for /o/), or sukūn for the absence of vowels in Farsi), wrong accents such as using é instead of è in Italian. Models experience substantial performance degradation when diacritics are added to Chinese (0.49 average drop) and Farsi (0.41 average drop), languages that typically lack such markers. This indicates that tokenizers

trained on undiacritized text struggle when these marks are introduced, despite their disambiguating potential. In contrast, models show much higher robustness to incorrect Italian accents (0.10 average drop). Among models, BLOOM performs best overall (0.26 average drop) due to its multilingual design; TokenMonster excels on Farsi (0.21 drop); GPT-2 slightly improves on Italian wrong accents (-0.02 drop); while Llama-3.2 exhibits severe degradation on Farsi (0.60 drop).

**Orthographic and Grammatical Errors**   Table 11 reveals that orthographic and grammatical errors create varying challenges depending on the morphological complexity of the language. TokenMonster and ByT5, a character-level approach, demonstrate the strongest performance.

- **Orthographic Errors:**  Orthographic errors represent spelling mistakes and character-level variations commonly

*Table 11.* Tokenization robustness under orthographic and grammatical errors. Values represent relative performance drop ($\frac{\text{Acc}_{can} - \text{Acc}_{pert}}{\text{Acc}_{can}}$); lower values indicate greater robustness.

| Model | Orthographic Errors | | | Grammatical Errors | | | Phonetic | Avg |
|---|---|---|---|---|---|---|---|---|
| | EN | TR | IT | EN | TR | IT | IT | |
| TokenMonster | 0.10 | **0.04** | 0.04 | 0.06 | 0.03 | -0.03 | 0.04 | **0.04** |
| ByT5 | **0.06** | 0.10 | 0.08 | 0.00 | **-0.01** | 0.04 | 0.02 | **0.04** |
| GPT-4o | 0.12 | 0.13 | 0.08 | 0.00 | 0.05 | -0.01 | 0.02 | 0.06 |
| Comma | 0.09 | 0.20 | 0.06 | **-0.03** | 0.13 | 0.01 | 0.04 | 0.07 |
| Llama-3.2 | 0.14 | 0.18 | 0.13 | 0.05 | 0.07 | 0.03 | 0.02 | 0.09 |
| Tekken | 0.24 | 0.23 | **-0.01** | 0.08 | 0.21 | **-0.07** | -0.01 | 0.09 |
| GPT-2 | 0.08 | 0.30 | 0.10 | 0.05 | 0.12 | 0.01 | 0.09 | 0.11 |
| BLOOM | 0.18 | 0.24 | 0.05 | 0.03 | 0.21 | -0.01 | 0.07 | 0.11 |
| Qwen-3 | 0.17 | 0.18 | 0.12 | 0.08 | 0.15 | 0.05 | 0.02 | 0.11 |
| Phi-3 | 0.18 | 0.22 | 0.13 | 0.11 | 0.09 | -0.02 | 0.07 | 0.11 |
| Aya | 0.21 | 0.21 | 0.13 | 0.03 | 0.07 | 0.02 | 0.14 | 0.11 |
| mBERT | 0.15 | 0.41 | 0.08 | 0.03 | 0.22 | -0.02 | 0.04 | 0.13 |
| XGLM | 0.13 | 0.32 | 0.12 | 0.03 | 0.23 | -0.02 | 0.15 | 0.14 |
| Gemma-2 | 0.18 | 0.30 | 0.12 | 0.05 | 0.29 | 0.07 | 0.09 | 0.16 |
| Avg | 0.14 | 0.22 | 0.09 | 0.04 | 0.13 | 0.00 | 0.06 | 0.10 |

encountered in real-world text, including vowel substitutions, consonant errors, phonetic spelling variants, common misspellings, and punctuation errors. Imagine perturbing the word "week" to "weak" in the question, "The number of days in a week is". This change breaks 6/14 models despite both words existing as distinct tokens with separate embeddings. This suggests that tokenization robustness depends not merely on vocabulary coverage but on the semantic stability of token representations.

- **Grammatical Errors:** Consider the Turkish locative suffix variants "*saat*teki" for the root saat (in the *hour*) versus the incorrect "saatdeki" as part of the canonical question "2 saatteki dakika sayısı" (Translation in English: "The number of minutes in 2 hours is").

This example demonstrates how agglutinative languages amplify tokenization brittleness: a single phoneme change (/t/ to /d/) can completely restructure token boundaries. This reflects the curse of multilinguality, where tokenizers trained predominantly on English struggle with morphologically complex languages, sometimes producing cleaner segmentation with meaningful morphemes for incorrect forms than correct ones (as Gemma-2 and BLOOM below). English grammatical errors, on the other hand like wrong prepositions and subject-verb agreement tend to change token boundaries less, and we observe a less striking performance degradation in Table 11.

*Assimilation error ("saatteki" vs. "saatdeki"):*

- **BLOOM, Gemma-2:** `sa, atte, ki` vs. `saat, de, ki` (meaningful morphemes after error)

- **XGLM:** `saat, teki` vs. `saat, deki` (clean morpheme separation)

- **Llama-3.2:** `sa, atte, ki` vs. `sa, at, deki` (inconsistent segmentation)

- **mBERT:** `saat, ##tek, ##i` vs. `saat, ##deki` (subword fragmentation changes)

- **Qwen-3:** `sa, atte, ki` vs. `sa, at, de, ki` (boundary reorganization)

- **TokenMonster:** sa, at, tek, i vs. sa, a, td, ek, i (severe fragmentation)

- **GPT-4o:** s, aat, te, ki vs. s, aat, de, ki (character-level consistency)

- **Tekken:** sa, atte, ki vs. sa, at, deki (partial boundary preservation)

- **GPT-2:** sa, at, te, ki vs. sa, at, d, eki (fine-grained segmentation)

*Turkish final-obstruent devoicing error ("ineğin" → "inekin")* in the word cow's (possesive)

- **BLOOM:** ine, Ç, Ş, in vs. in, ekin

- **XGLM:** in, e, ğ, in vs. in, ekin

- **Llama-3.2:** ine, Ç, Ş, in vs. ine, kin

- **mBERT:** [UNK] vs. in, ##ekin (unknown token fallback)

- **Qwen-3:** ine, ÇŞ, in vs. ine, kin

- **TokenMonster:** ine, g, ˇi, n vs ine, kin (diacritic decomposition)

- **Gemma-2:** ine, ğ, in vs. ine, kin

- **GPT-4o:** ine, ğ, in vs. ine, kin

- **Tekken:** ine, ğ, in vs. ine, kin

- **GPT-2:** ine, ğ, in vs. ine, kin

**Register and Style Variations**   Consider using emoji substitution in "The capital of Japan is" by replacing "Japan" with the emoji for Japan's flag.

*Table 12.* Tokenization robustness under different register and style variations. Values represent relative performance drop ($\frac{\text{Acc}_{\text{can}} - \text{Acc}_{\text{pert}}}{\text{Acc}_{\text{can}}}$); lower values indicate greater robustness. Abb.: abbreviations, Word Ord.: word reordering, Emoji: emoji substitution, Char. Subs.: character substitution, Repet.: letter repetition for emphasis

| Model | Web Search | | | Abb. | | Word Ord. | | Phonetic | Colloquial | | | | Emoji | Char. Subs. | Repet. | Avg |
|---|---|---|---|---|---|---|---|---|---|---|---|---|---|---|---|---|
| | EN | TR | IT | EN | IT | EN | TR | IT | EN | FA | TR | ZH | EN | EN | EN | |
| TokenMonster | 0.26 | 0.07 | 0.38 | 0.32 | 0.04 | 0.06 | -0.01 | 0.04 | 0.11 | 0.00 | -0.00 | 0.04 | 0.25 | -0.07 | 0.22 | 0.11 |
| mBERT | 0.33 | 0.25 | 0.23 | 0.27 | 0.07 | 0.08 | 0.18 | 0.04 | 0.15 | 0.09 | 0.12 | 0.18 | 0.29 | -0.08 | 0.18 | 0.16 |
| GPT-4o | 0.36 | 0.34 | 0.53 | 0.18 | 0.09 | 0.05 | 0.03 | 0.02 | 0.20 | 0.10 | 0.12 | 0.15 | 0.16 | -0.01 | 0.21 | 0.17 |
| ByT5 | 0.40 | 0.30 | 0.29 | 0.28 | 0.11 | 0.06 | 0.12 | 0.02 | 0.15 | 0.19 | 0.14 | 0.16 | 0.32 | -0.04 | 0.11 | 0.17 |
| Comma | 0.43 | 0.33 | 0.43 | 0.32 | 0.08 | -0.03 | 0.03 | 0.04 | 0.12 | 0.13 | 0.14 | 0.19 | 0.23 | 0.01 | 0.13 | 0.17 |
| BLOOM | 0.41 | 0.36 | 0.31 | 0.24 | 0.09 | 0.12 | 0.20 | 0.07 | 0.17 | 0.20 | 0.15 | 0.01 | 0.20 | 0.00 | 0.17 | 0.18 |
| GPT-2 | 0.29 | 0.36 | 0.38 | 0.20 | 0.16 | 0.13 | 0.15 | 0.09 | 0.10 | 0.06 | 0.18 | 0.21 | 0.26 | -0.05 | 0.28 | 0.19 |
| XGLM | 0.29 | 0.32 | 0.30 | 0.29 | 0.16 | 0.03 | 0.17 | 0.15 | 0.20 | 0.22 | 0.17 | 0.15 | 0.33 | 0.01 | 0.08 | 0.19 |
| Llama-3.2 | 0.38 | 0.32 | 0.36 | 0.30 | 0.13 | 0.10 | 0.14 | 0.02 | 0.19 | 0.17 | 0.08 | 0.17 | 0.25 | 0.06 | 0.27 | 0.20 |
| Tekken | 0.49 | 0.34 | 0.42 | 0.29 | 0.01 | 0.05 | 0.19 | -0.01 | 0.16 | 0.26 | 0.07 | 0.24 | 0.26 | 0.01 | 0.20 | 0.20 |
| Aya | 0.42 | 0.38 | 0.33 | 0.28 | 0.24 | 0.08 | 0.20 | 0.14 | 0.17 | 0.13 | 0.11 | 0.15 | 0.11 | -0.03 | 0.32 | 0.20 |
| Qwen-3 | 0.32 | 0.41 | 0.49 | 0.26 | -0.03 | 0.08 | 0.17 | 0.02 | 0.14 | 0.32 | 0.17 | 0.16 | 0.14 | 0.08 | 0.36 | 0.21 |
| Gemma-2 | 0.50 | 0.36 | 0.54 | 0.25 | 0.28 | 0.08 | 0.15 | 0.09 | 0.18 | 0.07 | 0.12 | 0.24 | 0.18 | 0.04 | 0.20 | 0.22 |
| Phi-3 | 0.43 | 0.31 | 0.62 | 0.20 | 0.04 | 0.11 | 0.15 | 0.07 | 0.24 | 0.21 | 0.19 | 0.23 | 0.33 | -0.05 | 0.28 | 0.22 |
| Avg | 0.38 | 0.32 | 0.40 | 0.26 | 0.11 | 0.07 | 0.13 | 0.06 | 0.16 | 0.15 | 0.13 | 0.16 | 0.24 | -0.01 | 0.21 | 0.19 |

*Emoji handling reveals differences:* Most modern tokenizers like Gemma-2, GPT-4o, Tekken, GPT-2, and Qwen-3 have emojis in their vocabulary, correctly parse the Japanese flag emoji into two tokens as the corresponding regional indicators ([J] and [P]). Aya on the other hand has a stand-alone token for the flag emoji. BLOOM, Llama-3.2, and TokenMonster use byte-fallback, XGLM and mBERT resort to unknown tokens. The coverage of emojis translate into good performance in the Emoji substitution perturbations (see Table 12).

**Linguistic Variety**    Table 13 examines how tokenizers handle linguistic diversity including historical spellings, code-switching, dialects, and colloquial expressions. TokenMonster demonstrates remarkable consistency across varied linguistic phenomena (0.08 average drop), while most models struggle significantly with certain types of variation. In Table 14, we group the models based on their vocabulary size (see Table 2) to investigate potential correlations with vocabulary size, as larger vocabularies theoretically provide more comprehensive dictionaries.

Counterintuitively, vocabulary size shows little to no correlation with linguistic robustness. Byte-level model (ByT5) demonstrates superior consistency despite operating without traditional vocabulary constraints, while some large-vocabulary tokenizers exhibit significant brittleness. We observe that larger vocabulary size doesn't always produce a lexically-rich vocabulary. Modern tokenizers may actually compound the problem by learning multiple variants of common words (Gemma-2 has distinct tokens for "hello", " hello", "Hello", and " Hello"), reducing the effective vocabulary. While this multiplicity has efficiency gains it could make models sensitive to stylistic variations that should be semantically equivalent.

Historical spelling variants ("capitall"[3], "Japane") demonstrate systematic fragmentation patterns where tokenizers often segment archaic or non-standard spellings along morphological boundaries:

- **Most tokenizers:** `capit, all` and `Jap, ane` (consistent morpheme-like splitting)

- **mBERT:** `capital, ##l` and `Japan, ##e` (subword suffix handling)

- **XGLM:** `capital, l` and `Japan, e` (clean separation)

Colloquial expressions reveal deeper challenges in world knowledge representation. The question "Turkey's capital turns out to be" with the correct answer "Ankara" illustrates how informal phrasing can disrupt factual recall: as it breaks 3 models. This suggests that tokenizers' handling of casual discourse markers and words ("turns out to be") may interfere with models' access to factual knowledge. The pattern indicates that linguistic variety challenges extend beyond mere tokenization to fundamental issues of how models integrate linguistic style with semantic content.

### E.2. Morphological Challenges

Table 15 examines how tokenizers handle morphological variations including derivations, inflections, and contractions across English, Turkish, and Italian. Morphological perturbations reveal fundamental inconsistencies in how tokenizers segment related word forms—contractions like "Google's" versus decomposed forms, or Italian elision patterns where "dell'Italia" and "d'Italia" receive dramatically different tokenization despite identical meaning. These inconsistencies suggest that current tokenization approaches lack coherent strategies for handling morphologically related forms, potentially leading models to develop disparate semantic representations for linguistically equivalent expressions. For example while BLOOM learns contractions, GPT-2 and GPT-4o use a regex-based search.

**English Contractions: "Google is"→ "Google's"**

- **BLOOM, Llama-3.2, Qwen-3, Gemma-2, GPT-2, GPT-4o, Tekken,:** `Google, 's` (separate marker)

- **XGLM, mBERT:** `Google, ', s` (fragmentation)

- **TokenMonster:** `google, 's` (lowercase normalization)

**Italian Ellisions**    The Italian contraction "L'intento" (the intent) demonstrates varying approaches to handling elided articles:

- **BLOOM:** `L', int, ento`

- **XGLM:** `L, ', inten, to`

- **Llama-3.2:** `L, 'int, ento`

- **mBERT:** `L, ', intento`

---

[3]https://www.oed.com/search/dictionary/?scope=Entries&q=capitall

- **Qwen-3:** `L, 'int, ento`

- **TokenMonster:** `l', intent, o`

- **Gemma-2:** `L, ', int, ento`

- **GPT-4o:** `L, 'int, ento`

- **Tekken:** `L, 'int, ento`

- **GPT-2:** `L, ', intent, o`

"dell'Italia" vs. "d'Italia":

- **BLOOM:** `d, ell, ', Italia` vs. `d', Italia`

- **XGLM:** `dell, ', Italia` vs. `d, ', Italia`

- **Llama-3.2, Qwen-3:** `d, ell, 'It, alia` vs. `d, 'It, alia` (fragments "Italia")

- **mBERT:** `dell, ', Italia` vs. `d, ', Italia` (length-dependent)

- **TokenMonster:** `dell, ', ita, lia` vs. `d, ', ita, lia` (lowercase + fragmentation)

- **Gemma-2:** `dell, ', Italia` vs. `d, ', Italia` (clean separation)

- **GPT-4o:** `d, ell, ', Italia` vs. `d, ', Italia` (inconsistent decomposition)

- **Tekken:** `d, ell, 'Italia` vs. `d, 'Italia` (treats apostrophe differently)

- **GPT-2:** `d, ell, ', It, alia` vs. `d, ', It, alia` (fragments country name)

### E.3. Noise

Table 16 shows robustness against common noise in digital text, such as keyboard proximity errors (e.g., s→(a,w,d,x), j→(k,u,h,m), ب → (ق, ر, ي, ل), 价→(加,们,份,什)) , OCR misrecognition (O→0, I→l), character deletion, space removal, and typographical errors (doctor→ doctro). These perturbations reflect authentic user input scenarios where models must maintain performance despite noisy text across multiple languages and writing systems.

We observe that tokenizers that segment text into complete word tokens tend to exhibit greater vulnerability to noise errors, as single character perturbations can cause familiar words to fragment into unfamiliar subword combinations, whereas tokenizers using smaller subword units maintain more consistent segmentation patterns.

**Noise in Chinese subset** For keyboard proximity errors in Chinese, characters are replaced with phonetically or positionally similar alternatives on the keyboard layout. For space removal, we use the Pinyin input without any spaces.

**Typos** Typographical errors demonstrate how different tokenization approaches handle character-level perturbations. For example, the word "doctor" with a typo becomes "doctro":

- **mBERT:** `doctor, doc, ##tro`

- **Comma:** `do, ctor, ⎵doc, tro`

- **Llama-3.2:** `doctor, ⎵do, ct, ro`

- **Tekken:** `doctor, doct, ro`

- **Aya:** `doctor, ⎵doct, ro`

- **GPT-4o:** `doctor, doct, ro`

*Table 13.* Tokenization robustness under linguistic variety. Values represent relative performance drop ($\frac{\text{Acc}_{\text{can}} - \text{Acc}_{\text{pert}}}{\text{Acc}_{\text{can}}}$); lower values indicate greater robustness. Hist.: historical spelling, equiv. exp.: equivalent expressions, sim. words: similar words

| Model | Hist. | Code switch | | | | Dialects | | | Equiv. exp. | | | | Sim. words | | | Avg |
|---|---|---|---|---|---|---|---|---|---|---|---|---|---|---|---|---|
| | EN | FA | TR | IT | ZH | FA | TR | IT | EN | FA | TR | ZH | EN | TR | IT | |
| TokenMonster | 0.09 | 0.07 | 0.00 | 0.00 | 0.03 | 0.22 | 0.09 | 0.17 | 0.14 | 0.07 | 0.04 | 0.03 | 0.03 | -0.06 | 0.22 | 0.08 |
| ByT5 | 0.06 | 0.03 | 0.04 | 0.06 | -0.04 | 0.29 | 0.15 | 0.15 | 0.02 | 0.13 | 0.06 | 0.04 | 0.08 | -0.08 | 0.24 | 0.08 |
| Comma | 0.21 | 0.10 | 0.13 | 0.06 | 0.03 | 0.30 | 0.04 | 0.06 | -0.05 | 0.10 | 0.06 | 0.03 | 0.08 | -0.02 | 0.28 | 0.09 |
| BLOOM | 0.25 | -0.07 | 0.16 | -0.03 | -0.04 | 0.31 | 0.19 | 0.14 | 0.05 | 0.07 | 0.14 | -0.07 | 0.09 | 0.13 | 0.26 | 0.11 |
| mBERT | 0.11 | 0.09 | 0.16 | 0.03 | 0.09 | 0.30 | 0.31 | 0.12 | -0.05 | 0.06 | 0.04 | 0.06 | 0.02 | 0.23 | 0.05 | 0.11 |
| Tekken | 0.21 | 0.12 | 0.16 | -0.03 | 0.03 | 0.37 | 0.14 | -0.02 | 0.17 | 0.15 | 0.06 | 0.03 | 0.05 | 0.18 | -0.01 | 0.11 |
| GPT-4o | 0.08 | -0.03 | 0.10 | -0.08 | 0.07 | 0.29 | 0.10 | 0.14 | 0.14 | -0.03 | -0.03 | 0.13 | 0.05 | 0.29 | 0.44 | 0.11 |
| XGLM | 0.18 | 0.09 | 0.21 | 0.06 | -0.03 | 0.30 | 0.15 | 0.02 | 0.17 | 0.03 | 0.10 | 0.09 | 0.08 | 0.16 | 0.10 | 0.11 |
| Gemma-2 | 0.31 | 0.17 | 0.05 | 0.05 | 0.10 | 0.33 | 0.23 | 0.07 | 0.17 | 0.00 | 0.07 | -0.10 | 0.04 | 0.08 | 0.40 | 0.13 |
| Aya | 0.21 | 0.03 | 0.13 | 0.08 | 0.03 | 0.30 | 0.18 | 0.14 | 0.27 | 0.16 | 0.10 | 0.00 | 0.07 | 0.10 | 0.23 | 0.14 |
| GPT-2 | 0.18 | 0.10 | 0.18 | 0.06 | 0.20 | 0.28 | 0.23 | 0.23 | 0.07 | 0.10 | 0.14 | 0.03 | 0.09 | 0.08 | 0.10 | 0.14 |
| Llama-3.2 | 0.25 | 0.03 | 0.13 | 0.03 | 0.09 | 0.24 | 0.05 | 0.17 | 0.10 | 0.03 | 0.17 | 0.19 | 0.09 | 0.16 | 0.40 | 0.14 |
| Qwen-3 | 0.32 | 0.21 | 0.18 | 0.05 | 0.04 | 0.34 | 0.18 | 0.11 | 0.02 | 0.24 | 0.17 | -0.07 | 0.09 | 0.22 | 0.15 | 0.15 |
| Phi-3 | 0.32 | 0.12 | 0.16 | 0.09 | 0.13 | 0.35 | 0.10 | 0.23 | -0.05 | 0.15 | 0.34 | 0.09 | 0.09 | 0.29 | 0.19 | 0.17 |
| Avg | 0.20 | 0.08 | 0.13 | 0.03 | 0.05 | 0.30 | 0.15 | 0.12 | 0.08 | 0.09 | 0.11 | 0.03 | 0.07 | 0.13 | 0.22 | 0.12 |

*Table 14.* Tokenization robustness under linguistic variety. Same as Table 13 but grouped under vocabulary size. Values represent relative performance drop ($\frac{\text{Acc}_{\text{can}} - \text{Acc}_{\text{pert}}}{\text{Acc}_{\text{can}}}$); lower values indicate greater robustness. Hist.: historical spelling, equiv. exp.: equivalent expressions, sim. words: similar words

| Vocab Size | Hist. | Code switch | | | | Dialects | | | Equiv. exp. | | | | Sim. words | | | Avg |
|---|---|---|---|---|---|---|---|---|---|---|---|---|---|---|---|---|
| | EN | FA | TR | IT | ZH | FA | TR | IT | EN | FA | TR | ZH | EN | TR | IT | |
| X-Small | 0.06 | 0.03 | 0.04 | 0.06 | -0.04 | 0.29 | 0.15 | 0.15 | 0.02 | 0.13 | 0.06 | 0.04 | 0.08 | -0.08 | 0.24 | 0.08 |
| Medium | 0.19 | 0.09 | 0.15 | 0.03 | 0.09 | 0.30 | 0.15 | 0.12 | 0.05 | 0.09 | 0.10 | 0.07 | 0.07 | 0.13 | 0.17 | 0.12 |
| Large | 0.23 | 0.07 | 0.14 | 0.02 | 0.03 | 0.31 | 0.17 | 0.10 | 0.14 | 0.08 | 0.09 | 0.00 | 0.07 | 0.16 | 0.26 | 0.13 |
| Small | 0.21 | 0.10 | 0.09 | 0.05 | 0.08 | 0.29 | 0.10 | 0.20 | 0.04 | 0.11 | 0.20 | 0.06 | 0.06 | 0.13 | 0.20 | 0.13 |
| Avg | 0.17 | 0.07 | 0.11 | 0.04 | 0.04 | 0.30 | 0.14 | 0.14 | 0.06 | 0.10 | 0.11 | 0.04 | 0.07 | 0.08 | 0.22 | 0.11 |

*Table 15.* Tokenization robustness under morphological challenges, granular version of Morphological in Table 1. Values represent relative performance drop ($\frac{\text{Acc}_{\text{can}} - \text{Acc}_{\text{pert}}}{\text{Acc}_{\text{can}}}$); lower values indicate greater robustness.

| Model | Contractions | | Compounds | Derivations | Inflections | | Avg |
|---|---|---|---|---|---|---|---|
| | EN | IT | EN | TR | EN | TR | |
| Comma | 0.23 | 0.18 | 0.09 | -0.11 | 0.02 | 0.02 | 0.07 |
| TokenMonster | 0.30 | 0.16 | 0.17 | -0.12 | 0.02 | -0.09 | 0.07 |
| GPT-2 | 0.33 | -0.08 | 0.09 | 0.05 | 0.02 | 0.13 | 0.09 |
| Aya | 0.27 | -0.03 | 0.19 | 0.02 | 0.05 | 0.06 | 0.10 |
| Gemma-2 | 0.27 | -0.03 | 0.14 | 0.02 | 0.12 | 0.06 | 0.10 |
| mBERT | 0.26 | -0.14 | 0.09 | 0.18 | 0.15 | 0.06 | 0.10 |
| Qwen-3 | 0.31 | 0.12 | 0.09 | 0.02 | 0.10 | 0.06 | 0.12 |
| GPT-4o | 0.26 | 0.26 | 0.12 | -0.04 | 0.07 | 0.06 | 0.12 |
| ByT5 | 0.30 | -0.03 | 0.15 | 0.09 | 0.21 | 0.05 | 0.13 |
| BLOOM | 0.20 | -0.01 | 0.16 | 0.11 | 0.14 | 0.16 | 0.13 |
| XGLM | 0.26 | 0.02 | 0.07 | 0.11 | 0.25 | 0.06 | 0.13 |
| Llama-3.2 | 0.29 | 0.12 | 0.16 | 0.02 | 0.14 | 0.11 | 0.14 |
| Tekken | 0.36 | -0.04 | 0.14 | 0.08 | 0.17 | 0.18 | 0.15 |
| Phi-3 | 0.28 | 0.07 | 0.14 | 0.09 | 0.25 | 0.08 | 0.15 |
| Avg | 0.28 | 0.04 | 0.13 | 0.04 | 0.12 | 0.07 | 0.11 |

- **GPT-2:** `doctor, doct, ro`

- **ByT5:** `d, o, c, t, o, r, ⎵, d, o, c, t, r, o`

Similarly, for Turkish text "gün sayısı" (day count) with spacing errors becomes "güns ayısı":

- **mBERT:** `gün, sayısı, gün, ##s, ay, ##ısı`

- **Comma:** `g, ün, ⎵say, ı, s, ı, ⎵g, ü, ns, ⎵ay, ı, s, ı`

- **Tekken:** `g, ün, say, ısı, gün, s, ay, ısı`

- **GPT-4o:** `g, ün, say, ısı, gün, s, ay, ısı`

- **Llama-3.2:** `gün, ⎵sayısı, ⎵gü, ns, ⎵ay, ısı`

- **GPT-2:** `g, ü, n, say, ı, s, ı, g, ü, ns, ay, ı, s, ı`

- **Aya:** `gün, ⎵sayısı, ⎵gün, s, ⎵ay, ısı`

- **ByT5:** Character-level segmentation (individual Unicode characters)

### E.4. Mathematical & Scientific Expressions

Table 17 demonstrates that models generally struggle with the formatting and structural challenges inherent in scientific domains. When numerical values are replaced with their spelled-out equivalents (15 → fifteen), we observe a consistent performance degradation even in English. The parallel multilingual basic arithmetic questions reveal that certain tokenizers may exhibit inductive biases favoring specific languages. For instance, Gemma-2's performance on Italian questions matches that of the canonical English questions, whereas it shows a 53% performance degradation in Farsi. Llama-3.2 demonstrates similar behavior with Turkish, while the Aya tokenizer, developed as part of a multilingual language model, exhibits the greatest robustness across languages. It should be noted, however, that this represents one of the few instances in our study where Aya tokenizer demonstrates clear multilingual advantages.

**Tokenization of scientific text:** Consider the unit "cubic meters" expressed as `m^3`, `$m^3$`, `$m^{3}$`, and `$m^{ 3 }$`. Despite semantic equivalence, tokenization patterns reveal increasing fragmentation:

- **BLOOM:**
    - Plain: `m, ^3`
    - LaTeX: `$m, ^3, $`
    - Braced: `$m, ^{3, }$`
    - Spaced: `$m, ^{, 3, }$`

- **XGLM:**
    - Plain: `m, ^, 3`
    - LaTeX: `$, m, ^, 3, $`
    - Braced: `$, m, ^, {, 3, }, $`
    - Spaced: `$, m, ^, {, 3, }, $`

- **Llama-3.2:**
    - Plain: `m, ^, 3`
    - LaTeX: `$m, ^, 3, $`
    - Braced: `$m, ^{, 3, }$`
    - Spaced: `$m, ^{, , 3, }$`

- **mBERT:**

*Table 16.* Tokenization robustness under multi-lingual noise. Values represent relative performance drop ($\frac{Acc_{can}-Acc_{pert}}{Acc_{can}}$); lower values indicate greater robustness.

| Model | Keyboard Errors | | | | | OCR | | Char. Del. | Space Removal | | Typos | | | Avg |
|---|---|---|---|---|---|---|---|---|---|---|---|---|---|---|
| | EN | FA | TR | IT | ZH | EN | ZH | EN | EN | ZH | EN | TR | IT | |
| Comma | 0.05 | 0.29 | 0.15 | 0.18 | 0.17 | 0.12 | 0.10 | 0.10 | 0.14 | 0.55 | 0.20 | 0.04 | 0.25 | 0.18 |
| ByT5 | 0.19 | 0.26 | 0.13 | 0.22 | 0.11 | 0.18 | 0.11 | 0.09 | 0.18 | 0.43 | 0.17 | 0.11 | 0.18 | 0.18 |
| TokenMonster | 0.22 | 0.18 | 0.15 | 0.13 | 0.16 | 0.10 | 0.26 | 0.04 | 0.13 | 0.58 | 0.08 | 0.09 | 0.25 | 0.18 |
| GPT-2 | 0.20 | 0.16 | 0.29 | 0.16 | 0.27 | 0.15 | 0.23 | 0.09 | 0.16 | 0.50 | 0.18 | 0.22 | 0.20 | 0.22 |
| Qwen-3 | 0.20 | 0.32 | 0.25 | 0.19 | 0.11 | 0.15 | 0.25 | 0.12 | 0.17 | 0.43 | 0.23 | 0.16 | 0.26 | 0.22 |
| GPT-4o | 0.13 | 0.20 | 0.13 | 0.13 | 0.23 | 0.15 | 0.40 | 0.18 | 0.16 | 0.53 | 0.24 | 0.13 | 0.22 | 0.22 |
| BLOOM | 0.22 | 0.23 | 0.34 | 0.16 | 0.11 | 0.19 | 0.11 | 0.16 | 0.21 | 0.56 | 0.16 | 0.25 | 0.17 | 0.22 |
| Gemma-2 | 0.17 | 0.23 | 0.21 | 0.22 | 0.19 | 0.17 | 0.29 | 0.16 | 0.15 | 0.52 | 0.14 | 0.13 | 0.30 | 0.22 |
| Llama-3.2 | 0.12 | 0.30 | 0.26 | 0.21 | 0.19 | 0.10 | 0.28 | 0.17 | 0.20 | 0.56 | 0.08 | 0.22 | 0.24 | 0.22 |
| XGLM | 0.18 | 0.25 | 0.29 | 0.19 | 0.23 | 0.15 | 0.29 | 0.13 | 0.13 | 0.60 | 0.11 | 0.22 | 0.21 | 0.23 |
| Tekken | 0.23 | 0.29 | 0.33 | 0.12 | 0.26 | 0.20 | 0.29 | 0.11 | 0.12 | 0.52 | 0.11 | 0.21 | 0.20 | 0.23 |
| Phi-3 | 0.15 | 0.27 | 0.22 | 0.20 | 0.22 | 0.20 | 0.22 | 0.21 | 0.18 | 0.53 | 0.20 | 0.20 | 0.21 | 0.23 |
| mBERT | 0.24 | 0.25 | 0.32 | 0.16 | 0.14 | 0.20 | 0.20 | 0.14 | 0.24 | 0.60 | 0.11 | 0.23 | 0.26 | 0.24 |
| Aya | 0.15 | 0.42 | 0.25 | 0.26 | 0.24 | 0.17 | 0.28 | 0.19 | 0.21 | 0.52 | 0.10 | 0.19 | 0.27 | 0.25 |
| Avg | 0.18 | 0.26 | 0.24 | 0.18 | 0.19 | 0.16 | 0.24 | 0.13 | 0.17 | 0.53 | 0.15 | 0.17 | 0.23 | 0.22 |

*Table 17.* Tokenization robustness under math and STEM related challenges. Values represent relative performance drop ($\frac{Acc_{can}-Acc_{pert}}{Acc_{can}}$); lower values indicate greater robustness. LaTeX: LaTeX-style math formatting; Diag. scientific diagrams and notations; Unic.: Unicode formatted ASCII characters; NEN=non-English.

| Model | LaTeX | Spelled Out | | | | | Diag. | | Multilingual | | | Unicode | Avg |
|---|---|---|---|---|---|---|---|---|---|---|---|---|---|
| | EN | EN | FA | TR | IT | ZH | EN | FA | TR | IT | ZH | EN | |
| TokenMonster | 0.23 | 0.28 | 0.49 | 0.07 | 0.33 | 0.31 | 0.11 | 0.29 | 0.00 | 0.14 | 0.00 | 0.08 | 0.19 |
| Phi-3 | 0.25 | 0.34 | 0.39 | 0.14 | 0.47 | 0.23 | 0.22 | 0.29 | 0.00 | 0.00 | 0.24 | 0.11 | 0.22 |
| Aya | 0.23 | 0.32 | 0.35 | 0.41 | 0.47 | 0.26 | 0.38 | 0.07 | 0.00 | 0.00 | 0.00 | 0.21 | 0.23 |
| mBERT | 0.15 | 0.35 | 0.55 | 0.45 | 0.35 | 0.38 | 0.22 | 0.14 | 0.07 | 0.14 | 0.07 | 0.23 | 0.26 |
| Llama-3.2 | 0.18 | 0.33 | 0.43 | 0.34 | 0.45 | 0.23 | 0.29 | 0.18 | 0.47 | 0.00 | 0.18 | 0.07 | 0.26 |
| GPT-2 | 0.25 | 0.38 | 0.35 | 0.32 | 0.44 | 0.08 | 0.35 | 0.18 | 0.35 | 0.24 | 0.24 | 0.17 | 0.28 |
| Tekken | 0.27 | 0.37 | 0.33 | 0.36 | 0.38 | 0.31 | 0.44 | 0.18 | 0.24 | 0.12 | 0.24 | 0.15 | 0.28 |
| BLOOM | 0.25 | 0.29 | 0.24 | 0.47 | 0.40 | 0.20 | 0.11 | 0.41 | 0.35 | 0.24 | 0.29 | 0.19 | 0.29 |
| Comma | 0.23 | 0.36 | 0.54 | 0.17 | 0.47 | 0.26 | 0.29 | 0.39 | 0.28 | 0.17 | 0.22 | 0.19 | 0.30 |
| ByT5 | 0.18 | 0.37 | 0.54 | 0.42 | 0.54 | 0.23 | 0.29 | 0.07 | 0.20 | 0.27 | 0.27 | 0.23 | 0.30 |
| GPT-4o | 0.25 | 0.38 | 0.33 | 0.45 | 0.52 | 0.28 | 0.33 | 0.37 | 0.32 | 0.05 | 0.16 | 0.20 | 0.30 |
| Gemma-2 | 0.22 | 0.35 | 0.33 | 0.32 | 0.53 | 0.40 | 0.37 | 0.53 | 0.35 | 0.00 | 0.18 | 0.23 | 0.32 |
| Qwen-3 | 0.26 | 0.41 | 0.50 | 0.41 | 0.47 | 0.23 | 0.29 | 0.25 | 0.35 | 0.20 | 0.30 | 0.23 | 0.33 |
| XGLM | 0.30 | 0.35 | 0.46 | 0.41 | 0.53 | 0.30 | 0.29 | 0.27 | 0.33 | 0.20 | 0.20 | 0.27 | 0.33 |
| Avg | 0.23 | 0.35 | 0.42 | 0.34 | 0.45 | 0.26 | 0.29 | 0.26 | 0.24 | 0.13 | 0.18 | 0.19 | 0.28 |

– Plain: `m, ^, 3`
– LaTeX: `$, m, ^, 3, $`
– Braced: `$, m, ^, {, 3, }, $`
– Spaced: `$, m, ^, {, 3, }, $` (identical tokenization)

- **Qwen-3:**

  – Plain: `m, ^, 3`
  – LaTeX: `$m, ^, 3, $`
  – Braced: `$m, ^{, 3, }$`
  – Spaced: `$m, ^{, , 3, }$`

- **TokenMonster:**

  – Plain: `m, ^, 3`
  – LaTeX: `$, m^, 3$`
  – Braced: `$, m^, {3}$`
  – Spaced: `$, m^, {, 3, }$`

Performance drops precipitously with formatting complexity. While all models correctly identified "volume" for plain text, only 8/14 succeeded with basic LaTeX formatting, 2/14 with braces, and just 2/14 with spaced braces. TokenMonster and Qwen-3 showed the highest robustness, maintaining correct answers through the spaced version.

This shows that even trivial whitespace differences in technical notation can cause catastrophic performance degradation, highlighting a critical vulnerability for applications that require strong mathematical reasoning.

**Structural ASCII Art and Chemical Notation**  These examples demonstrate how tokenizers handle structured chemical representations, from simple formulas to ASCII molecular diagrams and systematic nomenclature. The input contains `CH4`, an ASCII diagram of methane, `H2SO4`, and the systematic name "Dihydrogen sulfur tetraoxide":

- **BLOOM:**

  – Simple formulas: `CH, 4` and `H2, SO4`
  – ASCII structure: `H, |, H-C-H, |, H` (preserves structural elements)
  – Systematic name: `D, ih, yd, rogen, sulfur, tet, ra, oxide`

- **XGLM:**

  – Simple formulas: `CH, 4` and `H, 2, SO, 4`
  – ASCII structure: `H, |, H-, C, -, H, |, H` (fragments bonds)
  – Systematic name: `Di, hydro, gen, su, lfur, te, tra, oxide`

- **mBERT:**

  – Simple formulas: `CH, ##4` and `H, ##2, ##S, ##O, ##4`
  – ASCII structure: `H, |, H, -, C, -, H, |, H` (aggressive fragmentation)
  – Systematic name: `Di, ##hy, ##dro, ##gen, sul, ##fur, te, ##tra, ##ox, ##ide`

- **Gemma-2:**

  – Simple formulas: `CH, 4` and `H, 2, SO, 4`
  – ASCII structure: Uses special spacing tokens (____) for whitespace
  – Systematic name: `Di, hydrogen, sulfur, tetra, oxide`

- **GPT-4o:**

  – Simple formulas: `CH, 4` and `H, 2, SO, 4`

- – ASCII structure: `H, |, H-C-H, |, H` (clean structural preservation)
- – Systematic name: `D, ih, yd, rogen, sulfur, tetra, oxide`

- **GPT-2:**

  - – Simple formulas: `CH, 4` and `H, 2, SO, 4`
  - – ASCII structure: `H, |, H-, C, -, H, |, H`
  - – Systematic name: `D, ih, yd, rogen, sulfur, tet, ra, oxide`

- **Tekken:**

  - – Simple formulas: `CH, 4` and `H, 2, SO, 4`
  - – ASCII structure: `H, |, H-C-H, |, H` (preserves structure well)
  - – Systematic name: `D, ihydro, gen, sulfur, tetra, oxide`

- **TokenMonster:**

  - – Simple formulas: `ch, 4` and `h2, so, 4` (lowercase normalization)
  - – ASCII structure: Complex Unicode handling with encoding artifacts
  - – Systematic name: `di, hydrogen, sul, fur, tet, ra, ox, ide`

While all models correctly identified `CH4` as methane, only Llama and GPT-2 models correctly interpreted the ASCII molecular diagram. For `H2SO4`, all models succeeded, while spelled-out systematic nomenclature achieved 65% accuracy. The ASCII diagram failure is particularly revealing—the structured representation that humans easily recognize as methane becomes nearly incomprehensible to models when tokenized, despite containing identical chemical information. XGLM and mBERT normalize the whitespaces in the diagram, however they still fail to identify the molecule, maybe due to — characters. Gemma-2's special whitespace handling (`___`) and GPT-4o's clean structural preservation suggest different approaches to spatial formatting, yet neither prevented the semantic confusion in the ASCII representation.

### E.5. Styling & Unicode Challenges

*Table 18.* Tokenization robustness under Unicode formatting, NFKC normalization used by XGLM strips away all normalizations below. Values represent relative performance drop ($\frac{Acc_{can}-Acc_{pert}}{Acc_{can}}$); lower values indicate greater robustness.

| Model | Decorative Unicode | Fullwidth Characters | Scripted Text | Double Struck | Enclosed Characters | (Sup/sub) script | Avg |
|---|---|---|---|---|---|---|---|
| | EN | EN | EN | EN | EN | EN | |
| XGLM | **0.07** | **0.07** | **0.02** | **0.12** | **0.19** | **0.08** | **0.09** |
| ByT5 | 0.40 | 0.54 | 0.58 | 0.56 | 0.73 | 0.66 | 0.58 |
| GPT-2 | 0.47 | 0.59 | 0.59 | 0.68 | 0.61 | 0.65 | 0.60 |
| TokenMonster | 0.36 | 0.62 | 0.57 | 0.64 | 0.72 | 0.70 | 0.60 |
| Tekken | 0.41 | 0.73 | 0.57 | 0.62 | 0.73 | 0.62 | 0.62 |
| Gemma-2 | 0.53 | 0.54 | 0.67 | 0.62 | 0.68 | 0.66 | 0.62 |
| GPT-4o | 0.47 | 0.62 | 0.61 | 0.70 | 0.67 | 0.67 | 0.62 |
| Phi-3 | 0.47 | 0.54 | 0.59 | 0.75 | 0.73 | 0.67 | 0.62 |
| Aya | 0.36 | 0.68 | 0.71 | 0.63 | 0.69 | 0.69 | 0.63 |
| BLOOM | 0.59 | 0.51 | 0.62 | 0.67 | 0.72 | 0.65 | 0.63 |
| Qwen-3 | 0.60 | 0.67 | 0.69 | 0.62 | 0.57 | 0.64 | 0.63 |
| mBERT | 0.36 | 0.73 | 0.70 | 0.69 | 0.81 | 0.71 | 0.67 |
| Llama-3.2 | 0.59 | 0.60 | 0.70 | 0.69 | 0.76 | 0.68 | 0.67 |
| Comma | 0.67 | 0.60 | 0.67 | 0.81 | 0.70 | 0.58 | 0.67 |
| Avg | 0.45 | 0.57 | 0.59 | 0.63 | **0.67** | **0.62** | **0.59** |

Using Unicode characters and applying styling to the questions (or all choices) causes performance degradation across all models (see Tables 18 and 19). Although some tokenizers maintain distinct tokens for certain styled characters, they nevertheless exhibit significant failure rates. These styling variations could potentially be mitigated through normalization techniques, such as the NFKC normalization employed by XGLM. However, this is not always desirable, as these transformations are irreversible. We include the sample transformations in Figure 9.

*Table 19.* Tokenization robustness under different styling formats. Values represent relative performance drop ($\frac{\mathrm{Acc_{can}} - \mathrm{Acc_{pert}}}{\mathrm{Acc_{can}}}$); lower values indicate greater robustness.

| Model | Diacriticized EN | Lowercase EN | Capitalized EN | Upside Down EN | Spaced EN | Hyphenated EN | Avg |
|---|---|---|---|---|---|---|---|
| TokenMonster | 0.60 | 0.01 | **-0.03** | 0.47 | 0.66 | 0.69 | **0.40** |
| Aya | 0.66 | 0.08 | 0.15 | **0.42** | 0.54 | 0.67 | 0.42 |
| GPT-2 | **0.52** | 0.06 | 0.21 | 0.52 | 0.63 | 0.63 | 0.43 |
| Tekken | 0.57 | 0.03 | 0.16 | 0.60 | 0.63 | **0.61** | 0.43 |
| Gemma-2 | 0.69 | 0.06 | 0.15 | 0.47 | 0.64 | 0.67 | 0.45 |
| GPT-4o | 0.57 | **0.00** | 0.16 | 0.62 | 0.62 | 0.70 | 0.45 |
| Phi-3 | 0.58 | 0.11 | 0.18 | 0.47 | 0.68 | 0.66 | 0.45 |
| Comma | 0.58 | 0.06 | 0.11 | 0.60 | 0.68 | 0.68 | 0.45 |
| Llama-3.2 | 0.60 | 0.11 | 0.05 | 0.54 | 0.68 | 0.75 | 0.45 |
| Qwen-3 | 0.58 | 0.09 | 0.11 | 0.67 | **0.53** | 0.76 | 0.46 |
| ByT5 | 0.61 | 0.06 | 0.06 | 0.73 | 0.69 | 0.67 | 0.47 |
| BLOOM | 0.61 | 0.08 | 0.12 | 0.65 | 0.72 | 0.65 | 0.47 |
| mBERT | 0.64 | 0.09 | 0.16 | 0.80 | 0.59 | 0.65 | 0.49 |
| XGLM | 0.63 | 0.11 | 0.32 | 0.87 | 0.61 | 0.63 | 0.53 |
| Avg | 0.60 | 0.07 | 0.14 | 0.60 | 0.64 | 0.67 | 0.45 |

| Style | Text |
|---|---|
| Full width | Ｐｙｔｈｏｎ |
| Script | 𝒫𝓎𝓉𝒽𝑜𝓃 |
| Enclosed/ Circled | ⓅⓎⓉⒽⓄⓃ |
| Enclosed/ Parenthesized | ⒫⒴⒯⒣⒪⒩ |
| Superscript | ᴾʸᵗʰᵒⁿ |
| Subscript | ₚᵧₜₕₒₙ |
| Double struck | ℙ𝕪𝕥𝕙𝕠𝕟 |

| Style | Text |
|---|---|
| Underline | P̲y̲t̲h̲o̲n̲ |
| Macron | P̄ȳt̄h̄ōn̄ |
| Overline | P̅ȳt̅h̅ōn̅ |
| Upside down | Pʌʇɥou |
| Ring above | P̊ẙt̊h̊o̊n̊ |
| Diacritics | Pÿthøñ |
| Strikethrough | P̶y̶t̶h̶o̶n̶ |
| Strikethrough/ Forward slash | P̸y̸t̸h̸o̸n̸ |
| Strikethrough/ Backward slash | P̷y̷t̷h̷o̷n̷ |

*Figure 9.* **Left:** Styling challenges that are normalized by NFKC, **Right:** Styling challenges that NFKC cannot

# F. Decoding Algorithms

While `TokSuite` evaluates the model's ability to process perturbed variations of the input, we also evaluate tokenization robustness under various decoding algorithms that address tokenization bias. In our benchmark, the choices remain constant in the majority of examples unless applying the perturbation to the input is impossible. We evaluate each choice as a context-continuation log-likelihood normalized by byte-length. Since our perturbations occur at the input (context) level and not the output choices, existing decoding schemes are not directly comparable in our setting.

**Token Alignment and Token Healing** Methods like Token Alignment (Athiwaratkun et al., 2024) and Token Healing[4] are primarily designed to address ambiguity or failure modes during output generation and at token boundaries. A simple form of token healing is already implemented in the log-likelihood evaluation function of the lm-eval-harness framework (Gao et al., 2024) (see _encode_pair function).

**Exact Byte-Level Probabilities** Phan et al. (2025) introduce a method to convert tokenized language models into statistically equivalent byte-level models without retraining. Their approach addresses tokenization bias by computing exact byte-level probabilities from token distributions, enabling models with different tokenizers to be ensembled and improving performance on fill-in-the-middle tasks. We test their approach but note the following challenges:

*SentencePiece constraint:* The publicly available implementation relies heavily on SentencePiece tokenizers. Among our models, only Phi-3, Gemma-2, and XGLM use this tokenizer type. We ran byte-level evaluation specifically for our Gemma-2 model, which performed the worst among our SentencePiece tokenizers.

*Computational cost:* Due to byte-level rolling, the evaluation takes substantially longer (several hours on a subset of 40 examples compared to a few minutes for standard evaluation).

*Multilingual complexity:* The publicly available implementation relies heavily on space-based token delimiters and SentencePiece tokenizers, requiring extensive manual modification to handle non-Latin languages in our benchmark (Chinese and Farsi).

While the algorithm mitigates the performance drops in the Gemma-2 model as shown in Table 20, it fails to fully solve the issue. The robustness performance only brings it close to the top performer in Table 1, which is TokenMonster. This confirms that `TokSuite` remains a challenging, unsolved benchmark that warrants further investigation. Note that the algorithm degrades the canonical performance (Table 21).

*Table 20.* Tokenization robustness under multilingual text perturbations, with Phan et al. (2025) Exact-Byte level algorithm applied to the model trained with Gemma-2 tokenizer.

| Model | Input NEN | Diacr. NEN | Orth. EN | Gram. NEN | Morph EN | NEN | Noise EN | NEN | LaTeX EN | STEM EN | Unic EN | Avg |
|---|---|---|---|---|---|---|---|---|---|---|---|---|
| Gemma-2-EBL | **0.23** | 0.38 | 0.17 | 0.03 | **0.23** | **-0.09** | 0.16 | **0.19** | **0.13** | 0.15 | **0.35** | **0.17** |
| TokenMonster | 0.23 | **0.33** | **0.09** | **0.02** | 0.23 | -0.05 | **0.11** | 0.19 | 0.23 | **0.11** | 0.52 | 0.18 |
| Gemma-2 | 0.32 | 0.43 | 0.14 | 0.15 | 0.24 | 0.03 | 0.16 | 0.25 | 0.22 | 0.37 | 0.57 | 0.26 |
| Avg | 0.26 | 0.38 | 0.13 | 0.07 | 0.23 | **-0.04** | 0.14 | 0.21 | 0.19 | 0.21 | 0.48 | 0.21 |

*Table 21.* Canonical performance of `TokSuite`-Gemma-2 and `TokSuite`-TokenMonster along with `TokSuite`-Gemma-2 after applying Exact-Byte Level conversion from Phan et al. (2025).

| | ZH | EN | FA | IT | Math | STEM | TR | Avg |
|---|---|---|---|---|---|---|---|---|
| Gemma-2 | 0.78 | 0.98 | 0.75 | 0.98 | 0.81 | 0.89 | 0.82 | 0.86 |
| TokenMonster | 0.78 | 0.88 | 0.70 | 0.85 | 0.67 | 0.77 | 0.68 | 0.80 |
| Gemma-2-EBL | 0.68 | 0.78 | 0.70 | 0.68 | 0.52 | 0.66 | 0.58 | 0.65 |

*Table 22.* Tokenization Robustness of the Llama-3.2 Tokenizer across 300M, 1B and 7B model scales under multilingual text perturbations. Values represent relative performance drop ($\frac{\text{Acc}_{\text{can}} - \text{Acc}_{\text{pert}}}{\text{Acc}_{\text{can}}}$); lower values indicate greater robustness, same as Table 1.

| Model | Input | Diacr. | Orth. | Gram. | Morph | | Noise | | LaTeX | STEM | Unic | Avg |
|---|---|---|---|---|---|---|---|---|---|---|---|---|
| | NEN | NEN | EN | NEN | EN | NEN | EN | NEN | EN | EN | EN | |
| Llama-3.2-7B | **0.30** | 0.49 | **0.05** | **0.06** | **0.24** | 0.08 | **0.09** | **0.14** | **0.17** | 0.26 | 0.60 | **0.22** |
| Llama-3.2 | 0.34 | 0.54 | 0.12 | 0.09 | 0.25 | 0.08 | 0.15 | 0.24 | 0.19 | 0.29 | 0.59 | 0.26 |
| Llama-3.2-300M | 0.32 | **0.48** | 0.14 | 0.08 | 0.25 | **0.06** | 0.20 | 0.26 | 0.30 | 0.24 | **0.57** | 0.26 |
| Avg | 0.32 | 0.50 | 0.10 | 0.08 | 0.25 | **0.07** | 0.15 | 0.21 | 0.22 | 0.26 | 0.59 | 0.25 |

# G. Additional Ablations

### G.1. Model Scale

While a comprehensive study across all tokenizers at larger architectural scales remains computationally challenging, we trained Llama-3.2 models at three scales: 300M, 1B, and 7B parameters (excluding embeddings). We compare their performance in Table 22. Despite the 7B model demonstrating superior canonical performance over all fourteen baseline LMs in `TokSuite` (Table 23), the underlying tokenization robustness profile remains largely unchanged across scales. The three models exhibit highly similar robustness metrics across most perturbation categories, with noise showing modest improvement at larger scales, but other categories remaining nearly identical.

*Table 23.* Canonical accuracy of `TokSuite`-Llama models across different scales on `TokSuite` (higher better).

| Model | EN | FA | ZH | IT | TR | STEM | Math | Avg |
|---|---|---|---|---|---|---|---|---|
| Llama-3.2-7B | **1.00** | **0.80** | **0.97** | **0.93** | **0.90** | **1.00** | **0.93** | **0.93** |
| Llama-3.2 | **1.00** | 0.78 | **0.97** | 0.85 | 0.80 | 0.81 | 0.89 | 0.87 |
| Llama-3.2-300M | 0.93 | 0.70 | 0.88 | 0.70 | 0.65 | 0.76 | 0.82 | 0.78 |
| Avg | 0.97 | **0.76** | 0.94 | 0.82 | 0.78 | 0.86 | 0.88 | 0.86 |

### G.2. Text-matched Models

In addition to our primary evaluations, we trained four models using a fixed text budget rather than a token budget, using the Llama training data as baseline. We reconstructed the training data by logging each document sampled during the original Llama-3.2-1B training run and concatenating them into rank-specific corpora that preserve the original sampling order and mixing ratios. Naturally, we don't have control over the number of tokens consumed in each language. Moreover, the stochasticity of sampling introduces a slight difference in the order batches are seen during training between the models in the main text and here, hence slight differences in the baseline performance is expected.

We then trained four models (Llama-3.2, Gemma-2, Comma, Qwen-3) on this identical text sequence, calibrating training duration based on a 100,000-step run for Llama, which corresponds to approximately 104.86B Llama-3.2 tokens in total. See Table 24 for detailed hyper-parameters.

The text-matched models (Table 25) showed robustness patterns highly consistent with their token-budget counterparts (Table 1), with average performance drops of 0.24 in both conditions. Notably, despite Comma training on 70% more tokens (169.80B vs. ∼100B baseline), its robustness remained at a similar level (0.22 → 0.24) with modest improvements in its canonical performance (0.84 to 0.86), while Gemma consuming 5% less tokens showed comparable performance (0.26 → 0.24). This suggests that changing the training token budget does not directly translate into robustness gains, reinforcing that tokenization robustness is primarily determined by the tokenizer rather than training scale or data characteristics.

### G.3. Sensitivity to Random Initialization

To assess the stability of our results against different random initializations, we train two additional models with the LLaMA-3.2-1B tokenizer from different initialization seeds (888 and 222), keeping the training data order fixed relative to the original model. All robustness values reported in the main text fall within the bootstrapped confidence intervals reported in Figure 10, suggesting that our uncertainty estimates are consistent with cross-seed performance variation. As shown in

---

[4] https://github.com/guidance-ai/guidance/blob/main/notebooks/art_of_prompt_design/
prompt_boundaries_and_token_healing.ipynb

*Table 24.* Detailed training hyper-parameters and resulting token budgets for the four models trained with the same text budget. Training steps were adjusted per model to maintain a fixed text-volume budget relative to the Llama tokenizer baseline.

| | Llama | Qwen | Gemma | Comma |
|---|---|---|---|---|
| Local batch size | 4 | 4 | 4 | 4 |
| Gradient accumulation steps | 8 | 8 | 8 | 8 |
| # GPUs | 8 | 8 | 8 | 8 |
| Sequence length | 4096 | 4096 | 4096 | 4096 |
| Total steps | 100000 | 109564 | 95584 | 161938 |
| Total tokens (B) | 104.86 | 114.89 | 100.23 | 169.80 |

*Table 25.* Tokenization Robustness of the models trained with the same text budget (instead of token budget) under multilingual text perturbations. Values represent relative performance drop ($\frac{Acc\_can - Acc\_pert}{Acc\_can}$); lower values indicate greater robustness.

| Model | Input NEN | Diacr. NEN | Orth. EN | Gram. NEN | Morph EN | Morph NEN | Noise EN | Noise NEN | LaTeX EN | STEM EN | Unic EN | Avg |
|---|---|---|---|---|---|---|---|---|---|---|---|---|
| meta-llama-Llama-3.2-1B-textmatched | 0.29 | 0.43 | 0.08 | **0.08** | 0.24 | 0.06 | **0.12** | 0.21 | **0.16** | **0.28** | 0.59 | **0.23** |
| google-gemma-2-2b-textmatched | 0.29 | **0.40** | **-0.00** | 0.09 | **0.19** | 0.05 | 0.15 | **0.18** | 0.17 | 0.51 | 0.59 | 0.24 |
| common-pile-comma-v0.1-textmatched | **0.29** | 0.41 | 0.15 | 0.09 | 0.24 | 0.05 | 0.13 | 0.20 | 0.21 | 0.31 | 0.56 | 0.24 |
| Qwen-Qwen3-8B-textmatched | 0.31 | 0.45 | 0.11 | 0.09 | 0.19 | **0.04** | **0.12** | 0.22 | 0.19 | 0.48 | **0.53** | 0.25 |
| Avg | 0.29 | 0.43 | 0.08 | 0.09 | 0.21 | **0.05** | 0.13 | 0.20 | 0.18 | 0.39 | 0.57 | 0.24 |

*Table 26.* Canonical accuracy of `TokSuite` models trained with fixed text budget (instead of token budget) on `TokSuite` (higher better).

| Model | EN | FA | ZH | IT | TR | STEM | Math | Avg |
|---|---|---|---|---|---|---|---|---|
| meta-llama-Llama-3.2-1B-textmatched | **1.00** | 0.75 | **0.97** | **0.80** | **0.80** | **0.90** | 0.86 | **0.87** |
| common-pile-comma-v0.1-textmatched | 0.97 | 0.70 | 0.95 | **0.80** | 0.75 | **0.90** | 0.91 | 0.86 |
| google-gemma-2-2b-textmatched | 0.88 | 0.75 | 0.88 | 0.78 | 0.68 | 0.86 | 0.89 | 0.81 |
| Qwen-Qwen3-8B-textmatched | 0.93 | **0.78** | **0.97** | 0.75 | 0.65 | 0.76 | 0.84 | 0.81 |
| Avg | 0.94 | 0.74 | 0.94 | 0.78 | **0.72** | 0.86 | 0.88 | 0.84 |

Table 27, individual perturbation category scores also remain stable across seeds, with an average robustness of 0.25 across the three runs.

## G.4. Evaluating Industry-level Models on `TokSuite` Benchmark

To contextualize our findings, we evaluate existing pre-trained models on `TokSuite`, especially those using the same tokenizers as our suite. It is important to note that direct comparisons between our models and their pre-trained counterparts must be interpreted with caution due to fundamental differences in training data, model architecture, and scale. Even within the same model family, models may differ in training duration and other factors. We outline the major confounding factors below:

*Training Data & Multilingual Coverage:* While most of the evaluated models are reported to be "multilingual", their exact training data composition is largely unknown. Since robustness is measured by the drop in canonical accuracy, models must achieve reasonable baseline performance on canonical questions for fair comparison. We therefore exclude models with poor multilingual performance on `TokSuite`, as negative degradation values in such cases reflect low baseline performance rather than genuine robustness. For example, Phi-3 and GPT-2 were not pre-trained on a multilingual corpus; we evaluate such English-only models exclusively on English questions and report results separately in Table 29. For completeness, we report the canonical accuracy of these models on `TokSuite` in Table 31.

*Training Duration and Strategy:* It should be noted that these models are trained significantly longer than our controlled experiments. For example, Gemma-2-2B (Team et al., 2024) is trained on 2 trillion tokens. Moreover, comparing within the same model family might have additional confounding factors, like Qwen-3 (Yang et al., 2025) reporting strong-to-weak distillation for smaller models. Beyond duration, training objectives also confound comparisons: mBERT (Devlin et al., 2019) despite good tokenizer vocabulary coverage shows many negative values because it performs near random chance on our benchmark even in the English subset, which is reasonable as it was not trained with the next token prediction objective, and hence excluded from this section.

*Table 27.* Robustness scores for LLaMA-3.2-1B across three initialization seeds, 777 corresponds to the model evaluated in all other instances in the paper.

| Model | Input NEN | Diacr. NEN | Orth. Gram. EN | Orth. Gram. NEN | Morph EN | Morph NEN | Noise EN | Noise NEN | LaTeX EN | STEM EN | Unic EN | Avg |
|---|---|---|---|---|---|---|---|---|---|---|---|---|
| Llama-3.2-1B (model seed=888) | 0.31 | 0.44 | 0.12 | 0.15 | 0.25 | 0.13 | 0.09 | 0.24 | 0.09 | 0.26 | 0.58 | 0.24 |
| Llama-3.2-1B (model seed=777) | 0.31 | 0.56 | 0.11 | 0.11 | 0.25 | 0.10 | 0.09 | 0.26 | 0.13 | 0.29 | 0.60 | 0.26 |
| Llama-3.2-1B (model seed=222) | 0.29 | 0.54 | 0.09 | 0.12 | 0.20 | 0.13 | 0.14 | 0.29 | 0.19 | 0.37 | 0.59 | 0.27 |
| Avg | 0.30 | 0.51 | 0.11 | 0.12 | 0.24 | 0.12 | 0.10 | 0.26 | 0.13 | 0.31 | 0.59 | 0.25 |

*Table 28.* Tokenization robustness of external pre-trained models under multilingual text perturbations. Values represent relative performance drop ($\frac{Acc\_can - Acc\_pert}{Acc\_can}$); lower values indicate greater robustness. NEN=non-English.

| Model | Input NEN | Diacr. NEN | Orth. Gram. EN | Orth. Gram. NEN | Morph EN | Morph NEN | Noise EN | Noise NEN | LaTeX EN | STEM EN | Unic EN | Avg |
|---|---|---|---|---|---|---|---|---|---|---|---|---|
| xglm-564M | -0.01 | -0.03 | 0.15 | 0.00 | 0.14 | 0.12 | 0.13 | 0.09 | 0.24 | 0.05 | 0.11 | 0.09 |
| Qwen3-8B-Base | 0.23 | 0.32 | -0.06 | 0.04 | 0.23 | 0.03 | -0.03 | 0.14 | -0.01 | 0.04 | 0.30 | 0.11 |
| Qwen3-14B-Base | 0.21 | 0.23 | 0.01 | 0.04 | 0.27 | 0.05 | 0.04 | 0.14 | 0.04 | 0.04 | 0.37 | 0.13 |
| gemma-2-9b | 0.22 | 0.31 | 0.00 | -0.01 | 0.25 | 0.00 | 0.01 | 0.12 | 0.16 | 0.06 | 0.34 | 0.13 |
| Qwen3-30B-A3B-Base | 0.22 | 0.33 | 0.03 | 0.05 | 0.25 | 0.05 | 0.03 | 0.14 | 0.02 | -0.01 | 0.36 | 0.13 |
| Qwen3-4B-Base | 0.26 | 0.36 | -0.04 | 0.04 | 0.22 | 0.10 | 0.01 | 0.17 | 0.02 | -0.01 | 0.38 | 0.14 |
| c4ai-command-r-v01 | 0.18 | 0.30 | 0.07 | 0.01 | 0.27 | -0.12 | 0.09 | 0.12 | 0.14 | 0.08 | 0.49 | 0.15 |
| gemma-2-2b-it | 0.21 | 0.31 | 0.03 | 0.07 | 0.22 | 0.10 | 0.04 | 0.21 | 0.00 | 0.08 | 0.41 | 0.15 |
| aya-expanse-8b | 0.22 | 0.37 | 0.03 | 0.05 | 0.16 | 0.07 | 0.03 | 0.07 | 0.11 | 0.14 | 0.49 | 0.16 |
| Llama-3.2-1B-Instruct | 0.17 | 0.05 | 0.05 | 0.04 | 0.27 | 0.10 | 0.10 | 0.20 | 0.04 | 0.13 | 0.62 | 0.16 |
| Qwen3-1.7B-Base | 0.25 | 0.42 | 0.03 | 0.02 | 0.25 | 0.06 | 0.06 | 0.18 | -0.02 | 0.06 | 0.52 | 0.17 |
| Llama-3.2-3B | 0.21 | 0.36 | 0.01 | 0.02 | 0.25 | 0.04 | 0.04 | 0.17 | 0.02 | 0.18 | 0.53 | 0.17 |
| Qwen3-0.6B-Base | 0.12 | 0.44 | 0.10 | 0.02 | 0.25 | -0.07 | 0.12 | 0.19 | 0.04 | 0.18 | 0.50 | 0.17 |
| gemma-2-2b | 0.24 | 0.40 | -0.02 | 0.09 | 0.23 | 0.12 | 0.02 | 0.20 | 0.16 | 0.08 | 0.37 | 0.17 |
| babbage-002 | 0.09 | 0.10 | 0.10 | 0.03 | 0.27 | 0.05 | 0.10 | 0.13 | 0.22 | 0.26 | 0.56 | 0.17 |
| Llama-3.1-8B | 0.20 | 0.47 | 0.03 | 0.06 | 0.27 | -0.05 | 0.07 | 0.20 | 0.09 | 0.23 | 0.47 | 0.19 |
| Llama-3.2-3B-Instruct | 0.15 | 0.30 | 0.02 | 0.07 | 0.23 | 0.13 | 0.06 | 0.21 | 0.18 | 0.08 | 0.63 | 0.19 |
| Llama-3.1-8B-Instruct | 0.22 | 0.45 | 0.02 | 0.06 | 0.31 | 0.03 | 0.08 | 0.17 | 0.03 | 0.23 | 0.50 | 0.19 |
| BLT-1B | 0.15 | 0.49 | 0.06 | 0.09 | 0.25 | 0.06 | 0.06 | 0.23 | 0.16 | 0.11 | 0.61 | 0.21 |
| Llama-3.2-1B | 0.20 | 0.22 | 0.04 | 0.11 | 0.24 | 0.08 | 0.08 | 0.18 | 0.14 | 0.42 | 0.59 | 0.21 |
| Avg | 0.19 | 0.31 | 0.03 | 0.05 | 0.24 | 0.05 | 0.06 | 0.16 | 0.09 | 0.12 | 0.46 | 0.16 |

*Table 29.* Tokenization robustness of external English-only pre-trained models under English text perturbations. Values represent relative performance drop ($\frac{Acc\_can - Acc\_pert}{Acc\_can}$); lower values indicate greater robustness. eng=English.

| Model | Orth. Gram. EN | Morph EN | Noise EN | LaTeX EN | STEM EN | Unic EN | Avg |
|---|---|---|---|---|---|---|---|
| Bolmo-1B | 0.13 | 0.22 | 0.23 | 0.13 | -0.22 | 0.28 | 0.13 |
| Bolmo-7B | 0.14 | 0.37 | 0.11 | 0.04 | -0.08 | 0.41 | 0.16 |
| GPT-2 | 0.09 | 0.13 | 0.18 | 0.23 | -0.12 | 0.49 | 0.17 |
| Phi-3-mini-4k-instruct | 0.07 | 0.24 | 0.08 | 0.04 | 0.08 | 0.59 | 0.18 |
| phi-1_5 | 0.10 | 0.29 | 0.18 | 0.11 | 0.20 | 0.62 | 0.25 |
| Avg | 0.11 | 0.25 | 0.15 | 0.11 | -0.03 | 0.48 | 0.18 |

*Table 30.* Canonical accuracy of external pre-trained models on TokSuite (higher better).

| Model | EN | FA | ZH | IT | TR | STEM | Math | Avg |
|---|---|---|---|---|---|---|---|---|
| gemma-2-9b | **1.00** | **0.85** | 0.90 | **0.82** | 0.95 | **1.00** | 0.93 | **0.92** |
| Qwen3-14B-Base | **1.00** | 0.78 | **1.00** | 0.80 | 0.90 | **1.00** | **0.98** | **0.92** |
| Qwen3-30B-A3B-Base | **1.00** | 0.75 | **1.00** | 0.78 | **0.97** | **1.00** | 0.93 | **0.92** |
| Llama-3.1-8B | **1.00** | 0.72 | 0.95 | 0.70 | 0.90 | **1.00** | **0.98** | 0.89 |
| Llama-3.1-8B-Instruct | **1.00** | 0.62 | 0.93 | 0.75 | 0.88 | **1.00** | **0.98** | 0.88 |
| Qwen3-8B-Base | 0.90 | 0.72 | **1.00** | 0.78 | 0.88 | 0.90 | 0.91 | 0.87 |
| gemma-2-2b-it | **1.00** | 0.70 | 0.97 | 0.72 | 0.78 | **1.00** | 0.89 | 0.87 |
| Qwen3-4B-Base | 0.95 | 0.72 | 0.90 | 0.70 | 0.88 | 0.95 | 0.86 | 0.85 |
| gemma-2-2b | 0.95 | 0.68 | 0.90 | 0.78 | 0.85 | 0.90 | 0.89 | 0.85 |
| Qwen3-1.7B-Base | 0.97 | 0.70 | 0.88 | 0.60 | 0.90 | 0.95 | 0.86 | 0.84 |
| Llama-3.2-3B | 0.97 | 0.53 | 0.90 | 0.72 | 0.70 | 0.95 | 0.84 | 0.80 |
| Llama-3.2-3B-Instruct | 0.95 | 0.42 | 0.85 | 0.55 | 0.72 | 0.95 | 0.89 | 0.76 |
| Llama-3.2-1B | 0.97 | 0.45 | 0.78 | 0.68 | 0.62 | 0.90 | 0.86 | 0.75 |
| Qwen3-0.6B-Base | 0.97 | 0.50 | 0.75 | 0.42 | 0.78 | 0.95 | 0.84 | 0.75 |
| BLT-1B | 0.97 | 0.47 | 0.72 | 0.55 | 0.65 | 0.86 | 0.91 | 0.73 |
| Llama-3.2-1B-Instruct | 0.97 | 0.40 | 0.65 | 0.53 | 0.65 | 0.95 | 0.86 | 0.72 |
| aya-expanse-8b | 0.80 | 0.55 | 0.88 | 0.50 | 0.53 | 0.81 | 0.80 | 0.69 |
| c4ai-command-r-v01 | 0.88 | 0.68 | 0.70 | 0.50 | 0.55 | 0.76 | 0.75 | 0.69 |
| babbage-002 | 0.97 | 0.33 | 0.72 | 0.53 | 0.40 | 0.95 | 0.84 | 0.68 |
| xglm-564M | 0.68 | 0.25 | 0.53 | 0.50 | 0.50 | 0.48 | 0.59 | 0.50 |
| Avg | 0.95 | 0.59 | 0.84 | 0.65 | 0.75 | 0.91 | 0.87 | 0.79 |

*Table 31.* Canonical accuracy of external English-only pre-trained models on TokSuite (higher better).

| Model | EN | FA | ZH | IT | TR | STEM | Math | Avg |
|---|---|---|---|---|---|---|---|---|
| Phi-3-mini-4k-instruct | **1.00** | **0.38** | **0.95** | 0.30 | **0.72** | **0.95** | **0.89** | **0.74** |
| phi-1_5 | **1.00** | **0.38** | 0.45 | 0.20 | 0.40 | **0.95** | 0.86 | 0.61 |
| Bolmo-7B | 0.90 | 0.20 | 0.55 | 0.38 | 0.38 | 0.57 | 0.64 | 0.52 |
| GPT-2 | 0.62 | 0.20 | 0.33 | **0.40** | 0.35 | 0.43 | 0.61 | 0.42 |
| Bolmo-1B | 0.65 | 0.15 | 0.40 | 0.38 | 0.28 | 0.43 | 0.41 | 0.38 |
| Avg | 0.83 | **0.26** | 0.53 | 0.33 | 0.42 | 0.67 | 0.68 | 0.53 |

While we cannot draw definite conclusions due to these factors, several patterns still emerge (Tables 28 and 29):

*Model size improves canonical performance but not tokenization robustness:* Consistent with our controlled ablation across three Llama-3.2 scales, model size does not appear to be the determining factor for robustness. For ease of comparison, we group model families with the same color in the Table 28. Within families, size differences show minimal impact: Gemma-2-9B and Gemma-2-2B exhibit similar robustness profiles despite their parameter gap. Similarly, Aya-Expanse-8B—significantly larger than our TokSuite models—performs comparably to much smaller models. Instruction-tuned models at $\leq 2$ billion parameters (Gemma-2-2b and Llama-3.2-1B) show marginally better robustness than their base counterparts, though the improvement is modest. Majority of models attain very high canonical accuracy (Table 30), with larger models achieving better canonical performance within model families—yet this canonical improvement does not translate into significant robustness gains.

*Pre-trained models show improved robustness but universal challenges persist:* Pre-trained models demonstrate better overall robustness compared to TokSuite models, particularly on technical content: averaged across all models degradation on LaTeX drops from 0.22 (TokSuite in Table 1) to 0.09 (Table 28), and on STEM from 0.28 to 0.12. Unicode formatting also improves from 0.53 to 0.46, though it remains the most challenging perturbation category for both sets of models. This improvement likely stems from more diverse training data at scale, though persistent Unicode vulnerability suggests that even extensive real-world training inadequately covers such specialized character variations.

*Despite the robustness of ByT5, Latent Transformers struggle on TokSuite:* While ByT5 demonstrates strong robustness in our controlled experiments (Table 1), we don't observe same strengths in the byte-level latent transformer models (Byte-Latent Transformer (BLT) (Phan et al., 2025) and Bolmo (Minixhofer et al., 2025)). Since Bolmo is trained on English data, we report its performance exclusively on English perturbations (Table 29).

Surprisingly, neither model shows strong overall robustness comparable to TokSuite-ByT5. While BLT demonstrates improved robustness to English noise, it remains brittle against non-English noise perturbations. Bolmo-1B performs worse than the weakest TokSuite model on English noise. Although BLT shows improved robustness on input-medium challenges like diacritics, its substantially lower baseline canonical performance on the four non-English languages (Table 30) suggests that

that these robustness gains may partly reflect poor baseline performance.

These results suggest that architectural innovations in byte-level processing do not automatically confer robustness advantages. While latent transformers represent a promising direction for next-generation tokenization-free models, they still learn representations and segmentation patterns from training data—essentially a learned form of tokenization, which may inherit similar weaknesses against perturbations not well-represented during training.

*What TokSuite reveals about tokenization?* While we can only speculate about the relative contributions of tokenization in this section, due to confounding factors in these pre-trained models, our controlled TokSuite experiments provide definitive evidence by isolating tokenization effects through identical initialization, training data, and procedures. This reveals that tokenization choices alone account for performance variations comparable to those observed across vastly different training regimes in industry models. The consistent patterns across both controlled and real-world settings—particularly persistent challenges with Unicode formatting, morphological complexity, and technical content—demonstrate that these robustness issues are fundamental properties of tokenization schemes rather than artifacts of insufficient training. Although increased training scale and data diversity improve performance on clean canonical forms, they do not fundamentally resolve the brittleness addressed in the TokSuite benchmark introduced by tokenization design choices. These findings underscore the critical importance of tokenization research and the need for systematic evaluation frameworks like TokSuite.

## H. Statistical Significance

To ensure the robustness and reliability of our results, we employed two distinct statistical methods: bootstrapping to estimate variability and a non-parametric test to confirm performance differences.

**Estimating Variability (Bootstrapping):** We estimated the distributional statistics for robustness through a 10,000-trial bootstrap procedure. This process yielded reliable standard deviations, which are presented alongside the mean performance scores in Figure 10. We highlight that all of the performance differences discussed in Section 5 exceed one standard deviation, confirming that these observations are unlikely due to random variation.

In Figure 10, we plotted the 95% confidence interval (2.5–97.5 percentile) of the robustness metrics obtained from 10,000 bootstrap samples, with colors indicating the statistical significance (in terms of standard deviation from the mean) of each model's performance. We generate each view grouped by vocabulary size (Figure 11), underlying algorithm (Figure 12), and pre-tokenization splits (Figure 13).

**Significance (Wilcoxon Test):** To formally test the statistical significance of the differences between tokenizer performance, we utilized the Wilcoxon signed-rank test (Wilcoxon, 1945). This non-parametric test is appropriate for comparing two related samples (the performance of two different tokenizers on the same set of tasks). The results of the pairwise Wilcoxon signed-rank tests across all perturbation categories are presented in Table 32. Specifically, a p-value threshold of $\alpha = 0.05$ was adopted, and the results clearly demonstrate that the majority of the observed differences in robustness are statistically significant, further validating the conclusions drawn in our study.

*Table 32.* Statistically Significant Performance Differences (Paired Wilcoxon Signed-Rank Test). **Note:** Results where $P < 0.05$ are shown, we report values p values $< 10^{-4}$ as $< 10^{-4}$. The Median Drop Difference is calculated as $Median(Score_{Better}) - Median(Score_{Worse})$. A negative value indicates that the tested model has a statistically significant lower (better) robustness than the baseline model.

| Perturbation | Baseline Model | Model | Median Drop Diff. | $P$-Value |
|---|---|---|---|---|
| Input (Non-EN) | Gemma-2 | TokenMonster | −0.088 | $< 10^{-4}$ |
| | | Qwen-3 | 0.041 | $< 10^{-4}$ |
| Diacritics (Non-EN) | mBERT | TokenMonster | −0.110 | $< 10^{-4}$ |
| | | BLOOM | −0.093 | $< 10^{-4}$ |
| | | GPT-4o | 0.074 | $< 10^{-4}$ |
| | | Llama-3.2 | 0.109 | $< 10^{-4}$ |
| Orthographic Errors (EN) | Llama-3.2 | ByT5 | −0.069 | $< 10^{-4}$ |
| | | Comma | −0.056 | $< 10^{-4}$ |
| | | Phi-3 | 0.050 | $< 10^{-4}$ |
| | | Tekken | 0.076 | $< 10^{-4}$ |
| Orthographic Errors (Non-EN) | Phi-3 | TokenMonster | −0.075 | $< 10^{-4}$ |
| | | Tekken | −0.057 | $< 10^{-4}$ |
| | | Gemma-2 | 0.064 | $< 10^{-4}$ |
| Morphological (EN) | Gemma-2 | Comma | −0.058 | $< 10^{-4}$ |
| | | BLOOM | −0.054 | $< 10^{-4}$ |
| | | Tekken | 0.075 | $< 10^{-4}$ |
| Morphological (Non-EN) | GPT-4o | TokenMonster | −0.113 | $< 10^{-4}$ |
| | | Comma | −0.054 | $< 10^{-4}$ |
| | | Tekken | 0.042 | $< 10^{-4}$ |
| | | BLOOM | 0.052 | $< 10^{-4}$ |
| Noise (EN) | Llama-3.2 | TokenMonster | −0.045 | $< 10^{-4}$ |
| | | Comma | −0.034 | $< 10^{-4}$ |
| | | XGLM | −0.027 | $< 10^{-4}$ |
| | | mBERT | 0.030 | $< 10^{-4}$ |
| | | BLOOM | 0.034 | $< 10^{-4}$ |
| | | Aya | 0.041 | $< 10^{-4}$ |
| Noise (Non-EN) | Tekken | ByT5 | −0.031 | $< 10^{-4}$ |
| | | BLOOM | −0.027 | $< 10^{-4}$ |
| | | TokenMonster | −0.023 | $< 10^{-4}$ |
| | | Llama-3.2 | 0.027 | $< 10^{-4}$ |
| | | Gemma-2 | 0.034 | $< 10^{-4}$ |
| | | Aya | 0.040 | $< 10^{-4}$ |
| LaTeX | Comma | mBERT | −0.085 | $< 10^{-4}$ |
| | | Llama-3.2 | −0.056 | $< 10^{-4}$ |
| | | ByT5 | −0.052 | $< 10^{-4}$ |
| | | Tekken | 0.041 | $< 10^{-4}$ |
| | | XGLM | 0.066 | $< 10^{-4}$ |
| STEM (EN) | ByT5 | TokenMonster | −0.184 | $< 10^{-4}$ |
| | | BLOOM | −0.184 | $< 10^{-4}$ |
| | | Aya | 0.088 | $< 10^{-4}$ |
| | | Tekken | 0.145 | $< 10^{-4}$ |
| Unicode | ByT5 | XGLM | −0.418 | $< 10^{-4}$ |

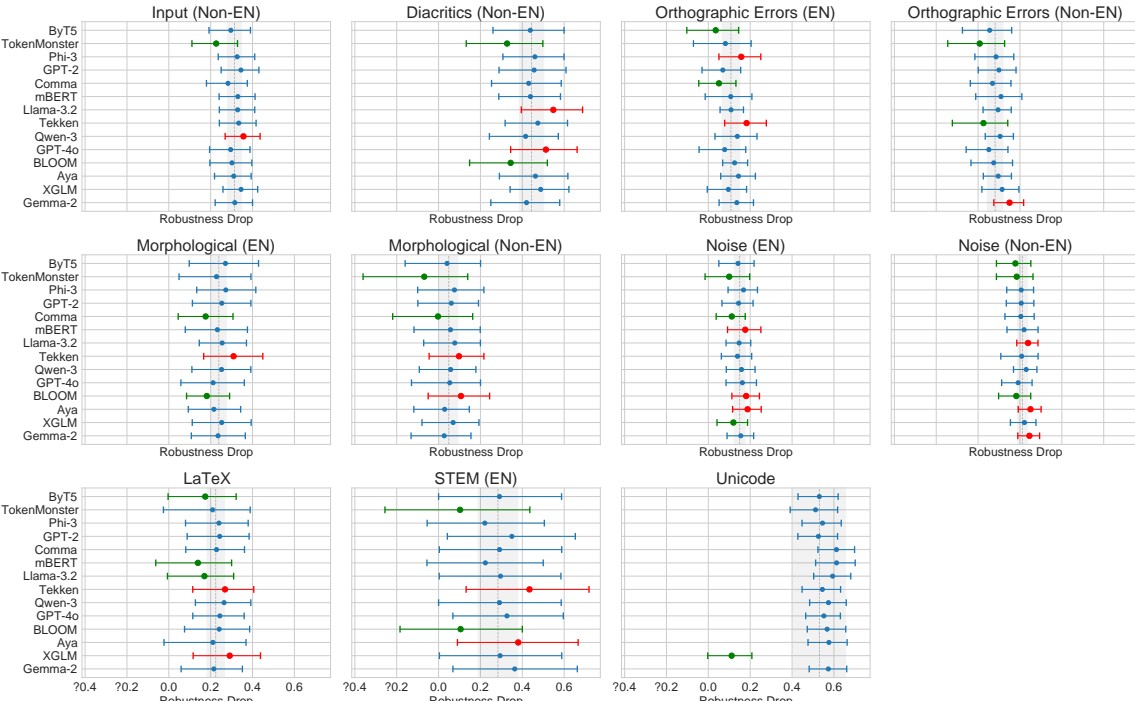

*Figure 10.* Distribution of tokenization robustness. Error bars represent the 2.5th to 97.5th percentile range across bootstrap samples. Models are ordered by their vocabulary size. The gray shaded region indicates ±1 standard deviation from the mean across all models for each perturbation type. Points are colored to highlight statistical significance: **green** indicates models that are significantly more robust (> 1 SD below mean), **red** indicates models that are significantly more fragile (> 1 SD above mean), and **blue** indicates performance within one standard deviation of the mean.

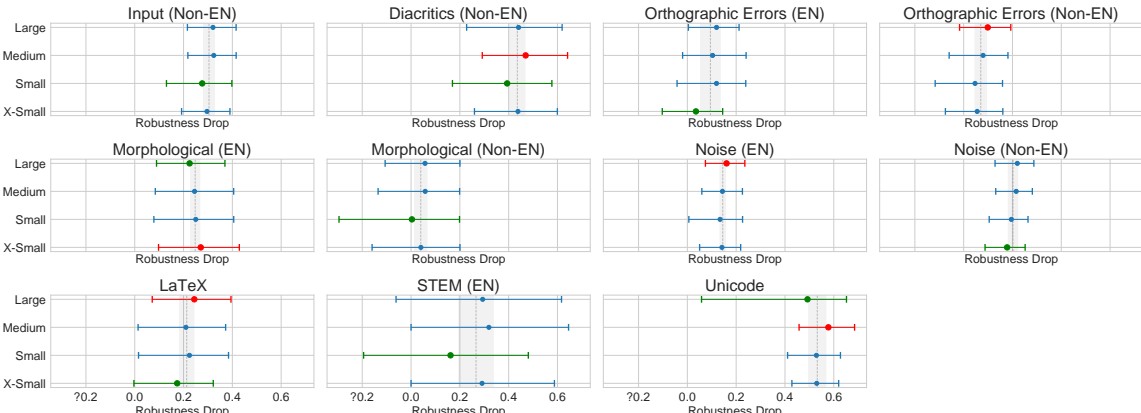

*Figure 11.* Same as Figure 10 but grouped by vocabulary buckets.

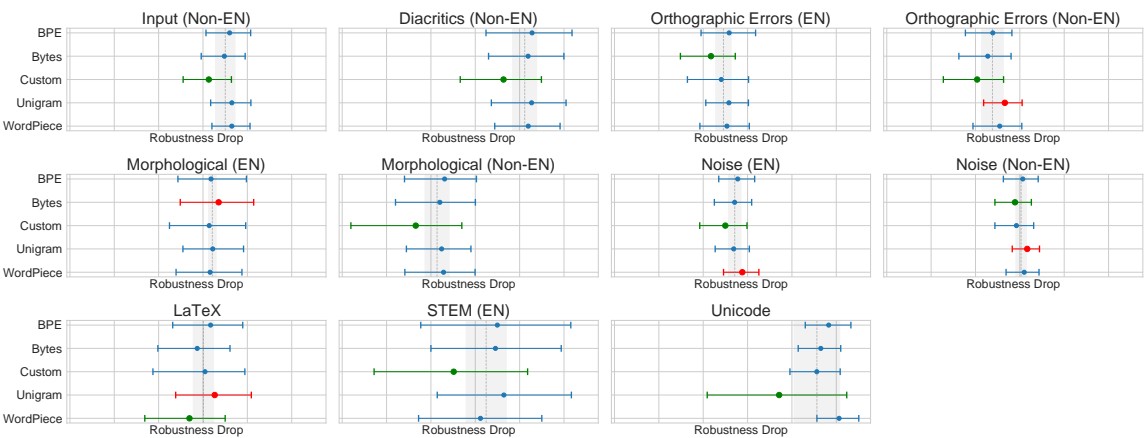

*Figure 12.* Same as Figure 10 but grouped by underlying algorithm.

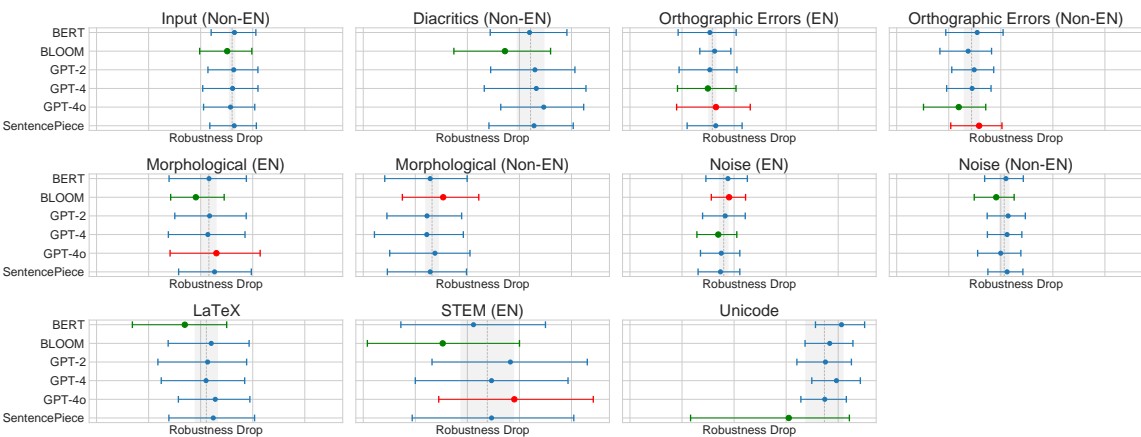

*Figure 13.* Same as Figure 10 but grouped by pre-tokenization splits (see Table 2 for details).

