# OpenReview forum: "TokSuite: Measuring the Impact of Tokenizer Choice on Language Model Behavior"
_ICML.cc/2026/Conference — ICML 2026 spotlight_

### Official Review · Reviewer_zMAa · 2026-03-10

**Soundness:** 3
**Presentation:** 2
**Significance:** 3
**Originality:** 2
**Overall Recommendation:** 4
**Confidence:** 5

**Summary:**

This paper introduces a benchmark for comparing tokenization differences.

**Compliance With Llm Reviewing Policy:**

Affirmed.

**Final Justification:**

My previous evaluation was based on my somewhat strong opinion that the tokenizer should be prepared for each corpus and model. But as the authors mentioned, unfortunately, many models reuse existing tokenizers regardless of compatibility. I reconsider that this paper could be a good alert for such a situation, and raised the overall score.

**Key Questions For Authors:**

Please see above

**Limitations:**

yes

**Strengths And Weaknesses:**

# Strength
- This paper compares multiple existing off-the-shelf tokenizers under as fair as possible conditions.
- The experimental results show the difference in performance caused by the difference in the tokenization (and the amount of training texts).
- This paper provides a dataset that includes both "canonical" and other variations with surface-level perturbations to check the robustness of the trained model.


# Weakness
- While I understand the importance of comparing the tokenization, my largest concern with this paper is that they compare existing off-the-shelf tokenizers bundled in different LLMs. This causes the following three problems, as the authors already noticed some of them in the paper:
  - Difficulty in the fairly controlled experimental setup: each tokenizer employs a different architecture, vocabulary size, normalization, and other hyperparameters. When comparing the tokenizers, what developers truly want to know is the difference caused by the architecture and other hyperparameter choices, instead of comparing existing tokenizers. Therefore, from the in-practice perspective, I consider the experimental setup in this manuscript is not aligned with what we truly want to know.
  - Problems with the amount of the training dataset: The different vocabulary size is also problematic, as the authors discussed. This causes a different amount of the training texts, and I consider this to be a significant effect on the performance. When one wants to compare the tokenization methods, they should use the same vocabulary size so that they can use the same amount of training data. While this paper argues that they compare the models with the same conditions but differ only in their tokenizers, I do not agree with this claim because they could not strictly exclude the effect of the difference in the training data.
  - Somewhat unrealistic setup: I do not agree with the experimental setup of employing the existing tokenizer for the training with a dataset that is not originally intended. Each tokenizer is developed for its training dataset by design (e.g., the LLaMA tokenizer is developed for the training data for LLaMA). Also, each tokenizer has its intended scope of languages, but the current experimental setup seems to break the alignment between the language coverage in the vocabulary and the new training dataset. I feel this experimental setup is not reasonable in practice.
- The size of TokSuite is too small to draw a comprehensive conclusion. To my understanding, the dataset contains only 40 examples and their surface-level perturbation. I am curious if only 40 questions can cover various linguistic phenomena occurring in the real texts. If the authors validated the diversity of the linguistic phenomena in the canonical texts, it is okay.

---

> ### Author Rebuttal · Authors · 2026-03-31
>
> Thank you for your review.
>
> ## Research Question
> > Difficulty in the fairly controlled experimental setup: each tokenizer employs a different architecture, vocabulary size, normalization, and other hyperparameters.
>
> We would like to clarify that all TokSuite models share the **exact same architecture (number of layers, heads, normalization, activation, etc.) and optimization hyper-parameters**, differing only in their tokenizer. The sole architectural consequence is the size of the embedding and unembedding layers. We further control for initialization such that only embeddings for non-shared tokens differ (Section 3.2). We consider this setup to provide appropriate controls for reliably measuring the impact of tokenizer choice on robustness.
>
> > …what developers truly want to know is the difference caused by the architecture and other hyperparameter choices, instead of comparing existing tokenizers.
>
> We believe this points to a complementary research question. In practice, tokenizer choice is often treated as a trivial decision, and off-the-shelf tokenizers are widely used without a clear understanding of their underlying design choices. TokSuite tokenizers were selected to represent diverse properties (and common practices) across all key dimensions: method, vocabulary size, pre-tokenization, training corpus, and normalization (Tables 2 and 3). A full factorial study jointly ablating all tokenizer design choices and training/model hyper-parameters would be computationally prohibitive and is beyond the scope of the current paper. Importantly, TokSuite uncovers statistically significant and meaningful differences in robustness across tokenizers.
>
> > from the in-practice perspective, I consider the experimental setup in this manuscript is not aligned with what we truly want to know.
>
> We believe understanding the isolated effect of the tokenizer choice is an important and understudied question—one that the other reviewers explicitly highlight as a valuable contribution, reflecting genuine community interest.
>
> ### Setup
> > Somewhat unrealistic setup…employing the existing tokenizer for the training with a dataset that is not originally intended…each tokenizer has its intended scope of languages
>
> We respectfully disagree with this framing. The scenario we study is not only realistic but increasingly common in practice: tokenizers are routinely reused across different datasets, domains, and languages. For example, GPT-Neo and GPT-J reuse GPT-2 tokenizer; SmolLM3-3B, TinyLlama, CodeLlama-34B, and Phi-3 all inherit from LLaMA tokenizers. Across model generations, Qwen-3 reuses Qwen's original tokenizer despite significantly expanded multilingual pretraining; Gemma-2 reuses Gemma-1's.
>
> ## Training Data & Performance
> > The different vocabulary size, effect on the performance.
>
> We address this in several ways:
> - When accounting for vocabulary size, all models attain similar [bits per byte (BPB)](https://anonymous.4open.science/r/toksuite-934F/loss-curves.md) through training.
> - While larger-vocabulary models tend to perform better on canonical questions (Table 6) and clean downstream tasks (Figure 6), we reduce this influence by selecting high-accuracy English canonical questions where all models perform very similarly (88%–100%), limiting capacity-driven variation on observed performance gaps. Crucially, we measure *robustness* rather than raw accuracy.
> - Ultimately, we find no clear correlation between vocabulary size and robustness (Figure 10), suggesting our benchmark is not confounded by vocabulary size.
>
> > Problems with the amount of the training dataset, different amount of the training texts
>
> More directly, we fix the training *text* for four models (LLaMA-3.2, Comma, Gemma-2, Qwen-3) in Appendix G.2. Robustness scores  fall within the bootstrapped confidence intervals from the paper—Comma: 0.22->0.24 despite 70% more tokens, Gemma: 0.26->0.24, Qwen: 0.26->0.25–suggesting robustness is primarily determined by tokenizer design rather than training duration.
>
> ## Benchmark Size
> > The size of TokSuite is too small to draw a comprehensive conclusion.
> We perform bootstrapping for the main results (Table 1) and only report statistically significant differences under paired Wilcoxon signed-rank tests.
> > …the dataset contains only 40 examples and their surface-level perturbation
> > if only 40 questions can cover various linguistic phenomena occurring in the real texts
>
> This is an excellent point we carefully considered during curation with iterative feedback from native speakers. TokSuite contains 40 canonical questions per language, plus 21 in MATH and 44 in STEM, each paired with multiple perturbation types—265 canonical questions and 4,676 perturbed variants, Table 7. We only included perturbation categories where sufficient high-quality examples could be found, prioritizing parallelism and statistical reliability over exhaustive coverage. Expanding TokSuite remains an exciting direction for future work.

---

> > ### Author Rebuttal · Reviewer_zMAa · 2026-04-03
> >
> > Thank you for the clarification and additional experimental results. I agree that many models reuse existing tokenizers; unfortunately, they should be retrained on the intended corpus. I changed my mind to value this work as an alert for such reuse and recommend the careful selection (or hopefully, recreation) of the tokenizers.

---

### Official Review · Reviewer_UHYH · 2026-03-12

**Soundness:** 4
**Presentation:** 3
**Significance:** 2
**Originality:** 3
**Overall Recommendation:** 5
**Confidence:** 4

**Summary:**

The authors pretrain language models using 14 different tokenizers to discern the impact of tokenizer choice on the robustness of the language model. Their results show that tokenization choice has significant measurable impact on the robustness of the models.

**Compliance With Llm Reviewing Policy:**

Affirmed.

**Final Justification:**

The authors addressed all my concerns. In particular they have explained adequately their tokenizer choices based on real-world design choices. Based on the rebuttal I believe this is an impactful paper. I keep my recommendation that this paper be accepted.

**Key Questions For Authors:**

1. Could you explain why TokenMonster has the best robustness?
2. Why did you choose relative performance drop as the metric?

**Limitations:**

yes

**Strengths And Weaknesses:**

# Soundness
## Strengths
The paper's methodology is strong in that it controls for all variable such as model architecture, number of parameters, token budget etc. Robustness is measured as relative performance drop which makes the different models comparable. The authors also performed statistical significance tests to make sure that their results are valid.
## Weakness
The tokenizers were chosen from some pretrained large language models instead of specific tokenizer properties. That makes it hard to draw broad conclusion about tokenizer properties from the experiments.

# Presentation
## Strengths
The paper is well presented with clear explanation of the methodologies.
## Weakness
The tables present all the non-English languages in aggregate. This makes it difficult to understand how tokenizer choice affects individual languages and how the characteristics of those languages affect the robustness of the language model. In addition the authors should spend more space discussing novel aspects of their paper such as cross-tokenizer vocabulary alignment and reduce discussions of background information.

# Significance
## Strengths
The paper shows some significant findings such as the TokenMonster tokenizer being the most robust even though it is English-only and that model scale does not significantly improve model robustness. Although the authors do not dive deep into the tokenizer properties to explain the results, the findings are still significant.
## Weakness
The main weakness is that the tokenizers were not chosen based on tokenizer properties. Thus the insights are not clear for future tokenizer design which limits it's significance

# Originality
## Strengths
The authors provide a controlled dataset for their evaluation. The dataset is multilingual about the perturbations for testing robustness were provided by human experts.
## Weakness
While the proposed methodology is sound it is not highly novel. Future, researchers may only find the results of this paper as interesting.

---

> ### Author Rebuttal · Authors · 2026-03-31
>
> Thank you for your constructive review and questions.
>
> ## Design Choices
> ### Tokenizer Choice
>
> > The tokenizers were chosen from some pretrained large language models instead of specific tokenizer properties. That makes it hard to draw broad conclusion about tokenizer properties from the experiments.
> > The main weakness is that the tokenizers were not chosen based on tokenizer properties.
>
> We would like to clarify that the tokenizers in TokSuite were selected specifically to represent diverse properties across all key dimensions of tokenizer design: (1) method (BPE, Unigram, byte-level, WordPiece); (2) vocabulary size (ranging from 32k to 256k); (3) pre-tokenization rules (whitespace handling, digit splitting, contraction splitting, Unicode normalization); (4) merge constraints and continuation-marker schemes (e.g., GPT-2 byte-fallback, WordPiece "##", SentencePiece "▁"); (5) training corpus used to learn merges; and (6) normalization and OOV strategies (byte-fallback vs. UNK).
>
>
> As a result, we consider our chosen tokenizers to cover a broad and realistic range of tokenizer properties. Additionally, by choosing existing tokenizers, we ensure that our study reflects common real-world design patterns, where off-the-shelf tokenizers are widely adopted without a clear understanding of how their underlying design choices affect model behavior. We acknowledge that this entangles design choices and makes it hard to pinpoint the effect of individual tokenizer properties. However,by studying existing tokenizers as deployed in practice, TokSuite sheds light on otherwise opaque design choices and reflects the real-world conditions under which tokenizers are actually selected and used.
>
>
>
> ### Robustness Metric
> > Why did you choose relative performance drop as the metric?
>
> Thank you for raising this question. Since we are not interested in the model's knowledge and capabilities but specifically in its sensitivity to variations that change tokenization while keeping the meaning intact, raw accuracy could conflate this analysis.  The perturbations in TokSuite are simple, realistic variations of text that **a native speaker would identify with perfect accuracy**. Ideally a good multilingual multi-purpose tokenizer/model pairing should be able to differentiate them too. The relative performance drop accounts for the influence of baseline performance differences across models.
>
>
> ## Results & Implications
> > Could you explain why TokenMonster has the best robustness?
>
> We speculate that the way TokenMonster handles tokenization might allow it to learn more robust representations. TokenMonster differs from standard BPE-based tokenizers in two key ways: its vocabulary is constructed through a distillation-based process that optimizes vocabulary and segmentation jointly rather than prioritizing compression alone, and its ungreedy algorithm combined with consistency constraints explicitly limits the number of distinct token sequences that can represent the same word.
>
> As an immediate consequence, surface-level variations such as capitalization or spacing are more likely to map to the same token sequence. For example, in many modern tokenizers "hello" and " hello" (with a leading space) are two distinct tokens, whereas TokenMonster's consistency constraints unify such variations—presenting the model with more consistent token sequences. This points to tokenization consistency as a promising design objective for future tokenizer research, particularly in multilingual settings. We will include this discussion in the camera-ready version of the manuscript.

---

> > ### Author Rebuttal · Reviewer_UHYH · 2026-03-31
> >
> > The authors addressed all my concerns.

---

### Official Review · Reviewer_r6y5 · 2026-03-13

**Soundness:** 3
**Presentation:** 4
**Significance:** 4
**Originality:** 3
**Overall Recommendation:** 5
**Confidence:** 5

**Summary:**

This paper examines the impact of tokenisation on LLM performance across 5 diverse languages and 14 tokenisers, by: (a) developing a test suite for each of the languages incorporating a wide range of "deviations" from standard text (e.g. orthographic errors, formatting inconsistencies, grammatical errors, structural elements, and markup in STEM/mathematical inputs); and (b) training and releasing 14 multilingual LLMs (one for each of the 14 tokenisers) trained under a fixed token budget in an attempt to make cross-model comparison as equitable as possible. Major findings include: (1) large disparities in performance across the LLMs attributable to the tokeniser choice, with some of the simplest tokenisers such as ByT5 being the most robust multilingually (at a cost in terms of inference efficiency); (2) "noise" of different types leading to a greater reduction in performance in non-English languages; (3) tokeniser choice also has a strong influence on performance for STEM/mathematical tasks with markup in the input; and (4) all tokenisers are vulnerable to unicode and character style transformations.

**Compliance With Llm Reviewing Policy:**

Affirmed.

**Final Justification:**

I confirm that I engaged with the authors in the rebuttal period. There were no major concerns with the paper to begin with, and while the responses/additional results further strength the paper, my overall assessment of Accept remains unchanged.

**Key Questions For Authors:**

- only one LLM is trained for each tokeniser, for only one token budget: how different are the loss curves across models, to get a sense of how well trained the respective models are?
- similarly, how robust are the results for a given tokeniser? e.g. if you had three separate (combined vocab) initialisations and trained three separate models for each each, how stable would the results be? I fully acknowledge that there are computational constraints in performing this analysis, but is there anything that can be inferred across the model checkpoints to suggest an answer here?
- given that the token budget is fixed, the larger the vocabulary, the less well trained each token is going to be on average (as much as it makes sense to talk about tokens in isolation); what is the correlation between vocab size and results across the different languages/tasks to get a sense of how much of an impact the vocab size has? alternatively, if you compare checkpoints such that the token budget for pre-training is normalised relative to vocab size, how much does this change the findings?
- for the scaling results for off-the-shelf models, is there a more systematic analysis that can be done relative to the 14 "toy" models in terms of models that use a similar or the same tokeniser? it seems that there is more that can be explored on this front
- for the EN vs. other lang results in Table 1, how different are the results across the other languages? for example, I would expect the results for IT to be closer to EN than for ZH, say, but looking in E2, this is sometimes the case and sometimes no, depending on the specific perturbation type/task. is there more analysis that can be done here/more than can be said?
- while I agree that latent tokenizer LMs are hard to train under the fixed token budget regimen to do an apples-for-apples comparison, are there results for pre-trained models over your dataset? it would be interesting to get a basic sense at least of how effective they are


== Not questions but because there is nowhere obvious in the review form to provide feedback on typos
[line 186, column 1] *and* tokenization algorithms
[line 186, column 2] use the AdamW optimizer
[line 366, column 2] improves, robustness

**Limitations:**

yes

**Strengths And Weaknesses:**

== strengths:

- (by far) the most systematic study of tokenisation on LLM performance that I am aware of, across a diverse range of commonly-used tokenisation strategies
- the multilingual benchmark dataset is expert crafted by native speakers of each of the 5 languages, and a huge contribution in itself
- the release of LLMs trained in a highly controlled manner to use in further analysis of tokenisation impacts on LLM performance in follow-up work

== weaknesses
- only one LLM is trained for each tokenisation strategy, with a fixed token budget in terms of pre-training, raising questions about the robustness of the results, and also the effects of under/over-training wrt vocabulary (size); this impacts on how credible/robust the results reported in the paper

---

> ### Author Rebuttal · Authors · 2026-03-31
>
> Thank you for your detailed review.
>
> ## Models
> ### Vocabulary Size and Performance
> > what is the correlation between vocab size and results across the different languages/tasks
>
> Models with larger-vocabulary tend to perform better on canonical questions (Table 6) or clean downstream tasks (Figure 6). To reduce this influence, we select high-accuracy English canonical questions where performance is very high across models (between 88% to 100%), reducing the influence of capacity-driven variation on observed performance gaps. Crucially, we measure *robustness* rather than raw accuracy and find no clear correlation (Figure 10) between vocabulary size and robustness on TokSuite, suggesting that our benchmark is not confounded by this simple variable.
>
> > is there anything that can be inferred across the model checkpoints to suggest an answer here?
>
> Unfortunately, we no longer have the intermediate checkpoints; hopefully multi-seed runs mentioned below will answer this question more rigorously. Relatedly, we include experiments where we control for the text budget for four tokenizers (Llama, Qwen, Gemma, and Comma) in Appendix G.2, ensuring they see the exact same data—and our findings mirror those from the main text.
>
> ### Training
> > how different are the loss curves across models, to get a sense of how well trained the respective models are?
>
> We did not observe any unusual behavior, loss spikes, or divergent losses during training. The only valid way to compare loss across tokenizers is bits per byte (BPB), to compute this we have to rematerialize the training data and rescale the loss accordingly. We computed BPB for 11/14 models at [https://anonymous.4open.science/r/toksuite-934F/loss-curves.md](https://anonymous.4open.science/r/toksuite-934F/loss-curves.md), where models perform comparably after accounting for vocabulary size. We will update the final manuscript with the complete curves.
>
> ### Randomness
> > how robust are the results for a given tokeniser?
>
> We launched two additional trainings with different initializations of the Llama tokenizer-model, with training data order fixed relative to the original model. One run (seed=888) has completed, and all values fall within the bootstrapped confidence intervals from the paper, suggesting that the originally reported uncertainty estimates are consistent with cross-seed performance variation.We will share the complete results if the second run finishes during the discussion period.
>
> **Table: Robustness of Llama model across two initializations**
> |Model|Input|Diacr.|Orth.\&Gram.||Morph||Noise||LaTeX|STEM|Unic|Avg|
> |-|:-:|:-:|:-:|-|:-:|-|:-:|-|:-:|:-:|:-:|:-:|
> ||NEN|NEN|EN|NEN|EN|NEN|EN|NEN|EN|EN|EN||
> |Llama-3.2-1B-seed_888|0.31|0.44|0.12|0.15|0.25|0.13|0.09|0.24|0.09|0.26|0.58|0.24|
> |Llama-3.2-1B-seed_42 (original) |0.31|0.56|0.11|0.11|0.25|0.10|0.09|0.26|0.13|0.29|0.60|0.26|
> ## Off-the-shelf models
>
> > is there a more systematic analysis that can be done relative to the 14 "toy" models in terms of models that use a similar or the same tokeniser?
>
> We agree this would be valuable. However, even within the same model family, confounding factors are difficult to control for—for example, some Qwen-3 models are reportedly distilled from larger Qwen-3 models, though the details are not fully disclosed. Off-the-shelf models sharing the same tokenizer still differ substantially in training data, scale, and optimization, making it difficult to attribute performance differences to the tokenizer specifically. We are happy to explore specific comparisons if the reviewer has concrete suggestions.
>
>
> ### Latent Tokenizer Language Models
> > while[...]latent tokenizer LMs are hard to train under fixed token budget regimen[...]are there results for pre-trained models over your dataset?
>
> We already evaluate Byte-Latent Transformer (BLT) (Table 27) and Bolmo (Table 28) on TokSuite, with the latter restricted to the English subset as it is English-only. While BLT shows strong robustness to English noise—similar to TokSuite-ByT5—this does not extend to Bolmo. Moreover, BLT performs substantially worse on the multilingual canonical questions (Table 29), likely because TokSuite models are explicitly trained on the benchmark languages, making direct comparisons difficult to interpret.
> ## Benchmark
> > how different are the results across the other languages?[...]I would expect the results for IT to be closer to EN than for ZH
>
> This is an interesting observation that we also noticed. The variation across languages does not follow a simple typological similarity as perturbations cause different tokenization fragmentations across languages–reflecting the interaction between the language-specific nature of each perturbation and tokenizer coverage of each script and morphology. Disentangling these effects in a principled way is non-trivial and we were unable to find a definitive signal. We are happy to explore specific comparisons if the reviewer has concrete analyses in mind.

---

> > ### Author Rebuttal · Reviewer_r6y5 · 2026-04-03
> >
> > In my view, the paper was comprehensive and covered the territory it needed to in its original form, and the responses to my questions further confirm my positive assessment of the work.

---

> > > ### Author Response · Authors · 2026-04-08
> > >
> > > We thank the reviewer for their continued positive view of our paper.
> > >
> > >
> > > We are happy to share that the third Llama variant also finished training. We see that all reported values fall in the confidence intervals reported in Figure 9 in the Appendix, confirming that the originally reported uncertainty estimates are consistent with cross-seed performance variation
> > >
> > > | Model | Input (Non-EN) | Diacritics (Non-EN) | Orthographic & Grammatical Errors (EN) | Orthographic & Grammatical Errors (Non-EN) | Morphological (EN) | Morphological (Non-EN) | Noise (EN) | Noise (Non-EN) | LaTeX | STEM (EN) | Unicode | Avg |
> > > | --- | --- | --- | --- | --- | --- | --- | --- | --- | --- | --- | --- | --- |
> > > | Llama-3.2-1B-seed_888| 0.31 | 0.44 | 0.12 | 0.15 | 0.25 | 0.13 | 0.09 | 0.24 | 0.09 | 0.26 | 0.58 | 0.24 |
> > > | Llama-3.2-1B-seed_42 (original) | 0.31 | 0.56 | 0.11 | 0.11 | 0.25 | 0.10 | 0.09 | 0.26 | 0.13 | 0.29 | 0.60 | 0.26 |
> > > | Llama-3.2-1B-seed_222 | 0.29 | 0.54 | 0.09 | 0.12 | 0.20 | 0.13 | 0.14 | 0.29 | 0.19 | 0.37 | 0.59 | 0.27 |

---

### Official Review · Reviewer_hgUr · 2026-03-13

**Soundness:** 4
**Presentation:** 4
**Significance:** 4
**Originality:** 4
**Overall Recommendation:** 6
**Confidence:** 4

**Summary:**

In this paper, they study the fundamental ingredient of all LLMs, tokenizers. Tokenizers' contribution to language model performance is understudied and hard because tokenization is so entangled with everything. Isolating it is difficult.

The paper contributes:
- Toksuite models: a collection of 14 language models (both monolingual in English and Multilingual)  that have identical init, architecture, and training data, and differ only in the tokenizer used (BPE, Wordpiece, Sentencepiece, byte-level approaches). The tokenizers also handle OOVs, Unicode normalization, whitespace handling, etc.
- All models use Lingua with the Llama-3.2-1B configuration, about 1B non-embedding parameters, trained for 100,000 steps on batches of 256 sequences of length 4096, with AdamW. The corpus is 100B tokens total: 40B English from FineWeb-Edu and 15B each of Chinese, Turkish, Italian, and Farsi from FineWeb-2 HQ.
- Toksuite benchmark: A benchmark of 5000 samples designed to target use cases where, essentially, even with identical semantic meaning, they are challenging for different tokenizers. The benchmark also includes languages other than English, such as Mandarin, Italian, Farsi, and Turkish. This includes a multilingual parallel dataset where 40 canonical multiple-choice texts are translated into other languages.
- Toksuite evaluation framework, which evaluates robustness, intrinsic tokenization efficiency, and statistical validation.


Their evaluation concludes the following:
- Tokenmonster and ByT5 are substantially more multilingually robust than subword methods despite smaller vocabularies. (Tokenmonster's 32k, which is 1/10th of Aya/XGLM).  Large multilingual vocabularies do not guarantee robustness.
- Noise-based perturbation methods affect non-English languages more
- All tokenizers struggle with LaTeX and STEM notation.
- Pinpoint universal challenges across tokenizers, like unicode styling and character transformation, that degrade performance across all models.

**Compliance With Llm Reviewing Policy:**

Affirmed.

**Final Justification:**

My recommendation for this work is to accept it, though some parts could be expanded, since it is a good, self-contained work. The paper has a strong, clear motivation, and the authors outline future work directions. This would be a good venue for bringing some light back to tokenization.

The rebuttal clearly addressed all my queries, and the authors will emphasize some details in the camera-ready version of this paper.

**Key Questions For Authors:**

- Given that embedding/output parameter counts vary with vocabulary size, could you estimate how much of the performance spread comes from extra capacity rather than tokenization quality?
- The benchmark questions [multiple choice] were chosen for high canonical accuracy. Was there a small subset for other tasks, like open-ended questions, and do you think these observations would hold?
- For practitioners, what objective should dominate tokenizer selection in multilingual systems? The conclusion of the paper is that TokenMonster is robust, but it has low cross-lingual parity/efficiency. [A suggestion that does not impact scores, a small practical takeaways section/or call on future work in specific directions would enhance the contribution of the paper.]

**Limitations:**

Limitations:
- The size of the models is still 1B (max), and it's 5 languages. [There could be languages where these observations don't hold]
- The benchmark questions are mostly multiple-choice-based questions, so we cannot use them to evaluate tokenizers for open-ended generation, long context, or retrieval.
- The study compares off-the-shelf tokenizers that vary simultaneously in algorithm, vocab size, tokenizer-training corpus, normalization, and pretokenization. So the methods are appropriate for measuring real tokenizer choice as practitioners face it, but less suitable for isolating any single tokenizer design factor causally.
- Total model size is not truly identical. Vocabulary size dramatically changes the embedding table and the final output layer. The paper carefully says “~1B non-embedding parameters,” for this, but this might be the best that can be done to isolate the variables. No good answer here.

**Strengths And Weaknesses:**

Strengths:
- The biggest strength of the paper is its contributions: the models, the ability to study vocabulary impact, and the analysis with 5 languages that are orthographically different.
- The benchmark is specifically constructed around perturbations that change tokenization. (Romanization, wrong keyboard layouts, zero-width characters, homoglyphs, Pinyin, Finglish, optional Farsi diacritics, and Unicode styling)
- They conclude that tokenizer quality is multi-dimensional: intrinsic efficiency, canonical task accuracy, and robustness to perturbation are not the same thing. [ XGLM achieves the parity scores, and mBERT shows the lowest average subword fertility)
- The experimental control to maintain everything else constant so as to isolate tokenization to study is one of the strongest features of the paper.
- High practical utility since future tokenization works can tokenize against this to show effectiveness against a variety of tokenizers.



Weaknesses:
- Factor attribution is observational, not factorial. The paper establishes that tokenization design matters. They show how each tokenizer differs on many axes at once: algorithm, normalization, OOV policy, tokenizer-training language mix, contractions, numbers, whitespace, etc. But do not isolate which part of tokenization it might be; the conclusion from the work is that "packaged tokenizer" matters a lot.

---

> ### Author Rebuttal · Authors · 2026-03-31
>
> Thank you for your substantial review and feedback. We are glad our work resonated with you.
>
> ## Models and Baseline Performance
> > Given that embedding/output parameter counts vary with vocabulary size, could you estimate how much of the performance spread comes from extra capacity rather than tokenization quality?
>
> Indeed, we observe that the models trained with tokenizers with a larger vocabulary tend to perform better on the canonical questions (Table 6) or clean downstream tasks (Figure 6). To reduce the influence of this factor, we select high-accuracy English canonical questions where performance for every model is very high (between 88% to 100%), reducing the influence of capacity-driven variation on observed performance gaps. Crucially, when measuring performance on TokSuite, we measure the *robustness*---i.e. the improvement or degradation in performance after perturbation–rather than raw accuracy. Ultimately, we find no clear correlation between vocabulary size and robustness on TokSuite (Figure 10), suggesting that our benchmark is not confounded by this simple variable.
>
>
> > The size of the models is still 1B (max)
>
> While most models in TokSuite have 1B non-embedding parameters, we ablate the model scale in Appendix G.1 with three models trained with the Llama tokenizer with 300M, 1B and 7B non-embedding parameters. While their canonical performance may vary, strikingly their robustness is very close to each other (0.22 for the 7B model and 0.26 for the 300M and 1B models) in Table 22, emphasizing that tokenizer choice rather than model scale is the determining factor for robustness.
>
>
> ## Benchmark
>
> > The benchmark questions [multiple choice] were chosen for high canonical accuracy. Was there a small subset for other tasks, like open-ended questions, and do you think these observations would hold?
> > …
> > The benchmark questions are mostly multiple-choice-based question
>
> We aimed for TokSuite to be able to be used with *base models*–i.e. models that have not undergone any supervised fine-tuning or other post-training. When designing TokSuite, we found it challenging to meaningfully measure base model performance on open-ended questions. Additionally, the primary goal of TokSuite is to measure *robustness* of different tokenizers, rather than raw capabilities. Hence, we concluded it was sufficient to use relatively easy multiple-choice questions for our "canonical" questions, and indeed, we were able to draw meaningful conclusions about the relative robustness of different tokenizers using our benchmark. Beyond this, we do agree that it would be valuable to design a similar benchmark targeting more powerful post-trained models and will emphasize this point for future work in our updated draft.
>
> > 5 languages
>
> We consider TokSuite a significant first step towards highlighting tokenization differences and language-specific failure modes. We limited the study to 5 languages in order to maintain a sufficient training budget for each language–ensuring enough base performance to meaningfully study robustness–and to work with native speakers for benchmark quality. Extending TokSuite to more languages, diverse domains, and generation-focused evaluations by leveraging the community is a natural and exciting next step.
>
>
> ## Impact
>
> > Factor attribution is observational, not factorial.
> > For practitioners, what objective should dominate tokenizer selection in multilingual systems?
> > So the methods are appropriate for measuring real tokenizer choice as practitioners face it, but less suitable for isolating any single tokenizer design factor causally.
>
> We agree with this characterization. This is a deliberate methodological choice that complements prior work such as [1] and [2], which train tokenizers from scratch to control specific properties in a more factorial setting. Together, these approaches address different but equally important questions.
>
> Interestingly, the generally close performance of most tokenizers on TokSuite suggests a convergence in how tokenization is performed across the field. In line with [1] and [2], we do not observe a clear correlation with intrinsic metrics such as parity, fertility, vocabulary size, and PCW—all of which prioritize compression. As a notable exception, TokenMonster, which differs fundamentally in its vocabulary construction and ungreedy tokenization algorithm, achieves strong robustness despite having one of the smallest vocabularies—suggesting that tokenization consistency may matter more than compression for robustness. We hope TokSuite provides a foundation from which more targeted tokenizer design research can be motivated and guided.
>
>
> [1] Ali, et al. Tokenizer Choice For LLM Training: Negligible or Crucial? https://arxiv.org/abs/2310.08754
>
> [2] Schmidt, et al. Tokenization Is More Than Compression. https://aclanthology.org/2024.emnlp-main.40/

---

> > ### Author Rebuttal · Reviewer_hgUr · 2026-04-01
> >
> > I thank the authors for their prompt response to my questions. I hope the authors will update their manuscript to highlight future work and clearly mention limitations wherever possible.
> >
> > Thank you and good luck!

---

### Decision · Program_Chairs · 2026-04-30

**Decision:**

Accept (spotlight)

**Comment:**

The paper studies the impact of tokenizer choice on language model behavior by training 14 otherwise identical models with different tokenizers, and releases a multilingual robustness benchmark spanning five diverse languages (English, Chinese, Farsi, Italian, and Turkish).

There was consensus among reviewers that this is an interesting and important contribution, with all four recommending acceptance. In the revised version, the authors are encouraged to address all reviewer comments by incorporating the additional arguments and results provided during the rebuttal. The most important point raised across reviews is that real-world tokenizers differ along multiple dimensions simultaneously (e.g., vocabulary size, algorithm, normalization, pre-tokenization, and training corpus), making it hard to attribute performance differences to any single design choice. The authors should bring the rebuttal discussion on this point, including the specific discussion related to TokenMonster into the main text to help guide future tokenizer design.